# MIND dataset for diet planning and dietary healthcare with machine learning: Dataset creation using combinatorial optimization and controllable generation with domain experts

Changhun Lee [* 1], Soohyeok Kim [* 1], Sehwa Jeong[1], Jayun Kim[2], Yeji Kim[2],
Chiehyeon Lim [† 1], Minyoung Jung [† 3]

[1]Ulsan National Institute of Science and Technology (UNIST)
{messy92, sooo, jsh0746, chlim}@unist.ac.kr
[2]Kosin University Gospel Hospital
{jydk6557, kimhana0419}@naver.com
[3]Kosin University College of Medicine
{my.jung}@kosin.ac.kr

## Abstract

Diet planning, a basic and regular human activity, is important to all individuals. Children, adults, the healthy, and the infirm all profit from diet planning. Many recent attempts have been made to develop machine learning (ML) applications related to diet planning. However, given the complexity and difficulty of implementing this task, no high-quality diet-level dataset exists at present. Professionals, particularly dietitians and physicians, would benefit greatly from such a dataset and ML application. In this work, we create and publish the Korean Menus–Ingredients–Nutrients–Diets (MIND) dataset for a ML application regarding diet planning and dietary health research. The nature of diet planning entails both explicit (nutrition) and implicit (composition) requirements. Thus, the MIND dataset was created by integrating input from experts who considered implicit data requirements for diet solution with the capabilities of an operations research (OR) model that specifies and applies explicit data requirements for diet solution and a controllable generative machine that automates the high-quality diet generation process. MIND consists of data from 1,500 South Korean daily diets, 3,238 menus, and 3,036 ingredients. MIND considers the daily recommended dietary intake of 14 major nutrients. MIND can be easily downloaded and analyzed using the Python package dietkit accessible via the package installer for Python. MIND is expected to contribute to the use of ML in solving medical, economic, and social problems associated with diet planning. Furthermore, our approach of integrating data from experts with OR and ML models is expected to promote the use of ML in other fields that require the generation of high-quality synthetic professional task data, especially since the use of ML to automate and support professional tasks has become a highly valuable service.

---

[*]Equal contribution.
[†]Corresponding author.

35th Conference on Neural Information Processing Systems (NeurIPS 2021) Track on Datasets and Benchmarks.

# 1 Introduction

Diet is "the sum of foods consumed by a person or other organism" [24], and diet planning is a regular human activity. The term "meal" implies consumed foods in general, and the term "diet" is used to indicate the combination of food menus planned for a specific purpose such as nutritional satisfaction, allergen avoidance, or weight control [8, 19]. Given that a diet is necessary for all individuals, diet planning has emerged as a core function of dietary healthcare research (DHR) in diverse disciplines that include food technology [21, 36, 37], nutrition management [5], clinical medicine [40], sports science [3, 15], and military nutrition [28, 12]. A single diet can be defined as a sequence of menus; diet planning involves the consideration of menus, ingredients, and nutrients (see Figure 1). A menu item is the complete product of cooked foods. For example, "a salad" is food and "ricotta cheese salad" is on the menu. Individuals usually consume end-products, not raw foods, and "menu" corresponds to the end product. "Ricotta cheese salad" consists of ingredients such as ricotta cheese, lettuce, and balsamic vinegar; and each ingredient contains several nutrients such as protein, fat, iron, sodium, etc. Therefore, any single diet can be hierarchically expressed with respect to menu-level, ingredient-level, or nutrient-level representations.

Diet planning is an advanced issue of the traditional "diet problem", the problem of optimizing quantities of foods and ingredients. The diet planning problem involves assessment of menus rather than foods. The solution to this problem is the optimization of the quantity of each menu with the simultaneous attainment of the optimal combination of menus (refer to Section 2 and Appendix A.1 for further details on the diet problem and diet planning). Recently in the healthcare field, researchers have attempted to define a health-related diet planning problem and to solve this problem using machine learning (ML). A major interest of medical DHR with ML is the design of a diet that counters disease-related factors [40, 20, 34, 1], and the ML studies of sports and military DHR focus on diets that strengthen physical abilities and metabolic controls [13, 6]. Despite the importance of ML application in academia and practice, studies in ML-based DHR are challenging because of the insufficiency of data. Figure 1 illustrates how DHR studies have been conducted based on the data of diet + X (e.g., menu, ingredient, or nutrition) configurations. Most of these previous studies have evaluated the physiological changes in subjects consuming different foods or have focused on recommending the consumption of specific foods based on perceived benefit. This indicates that diet data are the main source of information in those studies. However, a sufficiently large benchmark diet dataset that is accessible to the public does not yet exist. [7, 11, 30, 41]. This lack of a diet-level dataset may be the reason that most dietary studies have been based on operations research (OR) modeling instead of the ML approach that requires a dataset for training.

Several reasons exist for the lack of a diet-level dataset. From a data perspective, the diet can be defined as a set of menu items or food items arranged in a sequence, e.g., appetizer, main course, and dessert, for a specific purpose (see Figure 1). Obtaining a large quantity of diet data from current consumption practices may appear to be relatively simple. However, actual diet data have significant data quality issues. Our previous study provides evidence of this [17, 14]. While we were able to obtain an actual diet dataset that was created and used by public institutes and professional dietitians in South Korea, difficulty in use of this as a benchmark dataset arose for two reasons. First, the nutritional quality of each diet was inadequate. The first objective of dietary studies is to meet nutritional requirements according to age or other conditions, and necessary guidelines are clearly delineated by nutrition science. Surprisingly, many of the diets provided by public institutes did not meet these requirements. Many dietitians believe that this is an unavoidable reality because of the high complexity and difficulty of diet planning. Designing a diet plan is indeed complex and difficult because of its combinatorial optimization nature, which represents an NP-hard problem [39, 29]. For example, a breakfast plan with a combination of 100 menu items will consist of approximately $10^8$ options, supposing that a breakfast contains five menu items.

Second, the available datasets are insufficient in size. Usually, a unit of data in a diet dataset is one daily diet. Therefore, yearly data only contain approximately 300 examples, limiting the composition patterns of the diets. Additionally, diet planning involves substantial knowledge of food and nutrition. Understanding the context, e.g., religious beliefs and cultural

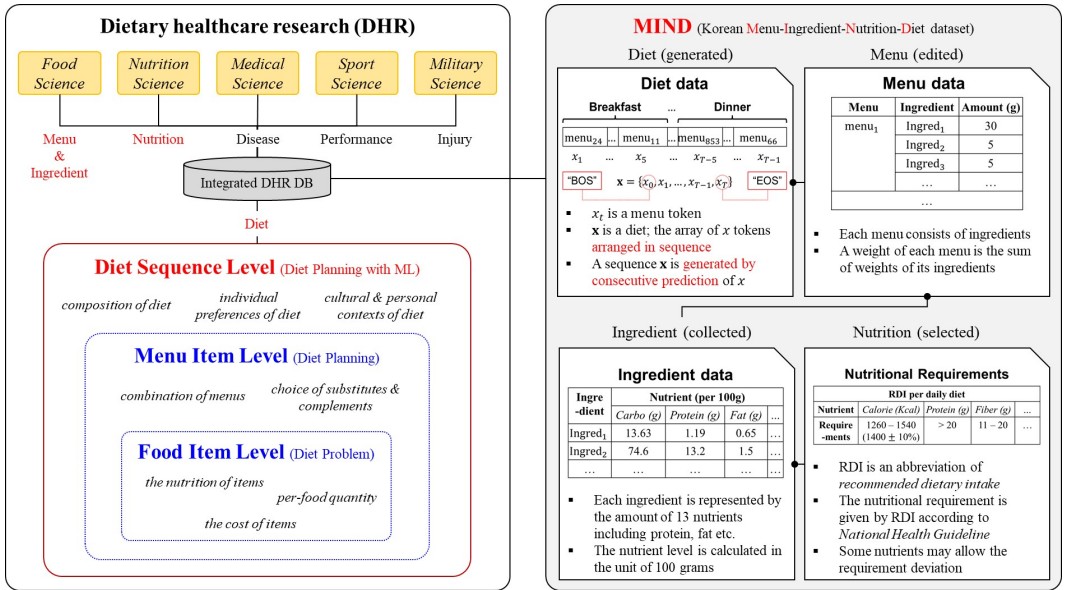

Figure 1: The scope of our study (left) and structure of the MIND dataset (right). The approaches in the blue boxes are used by most OR studies, which are based on the formulation of explicit requirements of diet planning; the approach in the red box is extended to learn implicit patterns in diets through ML. This figure shows the spectrum from existing works, primarily using an OR approach to confront the diet problem and diet planning to our ML-based approach to address these issues. In summary, all previous studies on diet planning consider ingredient and menu-level information, but diet-level planning should involve the compositional patterns of menus in diets. In addition, existing ML studies on dietary healthcare also consider only the ingredient and menu levels. The proposed MIND dataset is the first dataset that integrates all of the hierarchical relationships between diets-menus, menus-ingredients, and ingredients-nutrition.

orientation, and health and development issues, e.g., growth, aging, and the pathogenesis of chronic diseases, is also of prime importance [23, 25]. This knowledge must be treated as constraints when generating diets, but only some of these topics have an explicit guide for specifying nutritional and other dietary requirements. No guidelines exist for the remaining topics because the guidelines and topics are related to implicit requirements that include the composition of a diet. As a result, professional dietitians employed in government or daycare centers often copy and edit existing diets that are poorly crafted (see Section 4), and this emulation behavior adversely impacts the quality and size of available diet datasets. Similarly, although medical doctors and dietitians in large hospitals should design specialized diet plans for inpatients, few inpatients receive these services. Last, diet planning in the home is usually unsystematic, contributing to the low quality and insufficient size of the available benchmark dataset. Therefore, the focus of our study is data augmentation using synthetic diets of high quality to construct a benchmark dataset for ML-based diet planning applications and DHR.

To generate synthetic diets of high quality, we initially performed the task of diet generation by redefining the traditional OR diet planning problem as an ML one, a controllable generation problem as described in Section 2. Accordingly, we devised an OR–Xperts–ML (ORxML) framework that integrates input from experts with the capabilities of OR and ML modules (see Section 3). Each OR, Expert, and ML module is responsible for the initialization, evaluation, adjustment, and control of diet generation. The specific process involves the formulation of a combinatorial optimization OR model to generate synthetic diets as a means of satisfying explicit nutrient requirements. Next, we recruited experts, professional dietitians, to evaluate and adjust the initial data in terms of implicit requirements. These implicit requirements are criteria that cannot be specified in the combinatorial optimization model. An example of these requirements is the essential dietician task of assessing the

composition of a diet based on its implicit and contextual nature. This is critical to make the diet recipients accept and enjoy menus with high nutritional quality. See Appendix A.4 for further details on the compositional quality of diets. Without this consideration, feasible solutions for diet planning cannot be provided in practice. Last, we developed a controllable diet generation machine to: (a) ensure composition compliance by learning the data patterns constructed by the OR model and experts, (b) enhance nutrition by approximating an optimal policy to maximize the nutrient rewards, and (c) automatically augment the data by executing an optimal policy and generating synthetic diets.

With the diets generated by the ORxML framework, we created the *Menu–Ingredient–Nutrient–Diet* (MIND) dataset for diet planning and DHR with ML and introduce this dataset in this study. Figure 1 shows the MIND dataset that consists of 1,500 daily diets, 3,238 menus, and 3,036 ingredients. Satisfaction of the nutritional intake requirements for 14 major nutrients was a significant consideration. The original sources of the menu items, ingredients, and nutrient information are the public databases of South Korean government organizations that are responsible for ensuring the country's nutrition standards, and the diet data were created by the authors from the beginning using the ORxML framework. The quality of the diets was validated by dietitians and physicians, and we received approval from the government organizations responsible for determining nutrition quality in South Korea (e.g., the Ministry of Food and Drug Safety and the Rural Development Administration) to distribute the MIND dataset. The MIND dataset can be downloaded and subsequently analyzed easily using the Python package called *dietKit*, which is accessible via the package installer for Python.

This work is original research with academic merit and practical implications as illustrated in Figure 1. Diet planning is an important problem that should be solved with ML but could not be addressed in this way due to the lack of datasets for this data-driven approach. To the best of our knowledge, this work is the first to create and publish a large-scale and high-quality diet-level dataset for diet planning and DHR using ML. Section 2 explains the methodological background more thoroughly. In addition, this work represents a first attempt to develop a framework for generating high-quality synthetic data for professional tasks. Section 3 explains the ORxML framework in detail. In Section 4, we discuss how the quality of the MIND dataset was evaluated via a series of experiments to demonstrate the significance of the three modules, the OR model, the knowledge and experience of experts, and the ML model. The final outcome of the MIND dataset is described in Section 5. Our work has already started to create an impact. In Section 6, we discuss ML applications of our dataset as a means of assisting dietitians, medical doctors, and the public in their diet planning and related healthcare tasks. In Section 7, we discuss how the ORxML framework can be applied to constructing high-quality synthetic data involving professional tasks in other domains.

## 2  Background and Literature Review

The academic concepts and definitions necessary to understand our research are briefly discussed in this section. Each of the two subsections defines the diet planning problem and its recent paradigm with the support of ML.

**Diet planning problem**  The concept of the *diet problem*, highlighted by Dantzig [4], was motivated by the United States Army's desire to meet the nutritional requirements of military personnel in the field while minimizing the cost of implementing the endeavor [2]. The prototype study of the diet problem was published in 1945 when George Stigler, who later received the Nobel Prize, presented an economical diet model [35]. Stigler regarded the diet problem as a scenario involving continuous optimization to identify optimal quantities of food items; thus, a linear programming approach was adopted. However, Stigler's approach was later criticized as impractical by subsequent economists and operation researchers. Most criticisms centered on the optimization units. Smith [33] and Smith [32] explained that the linear programming solution, i.e., using an optimal set of food items, was "unpalatable" because the linear models exemplified "one-dish meals" similar to animal feed blends rather than those fit for a "daily human diet." Similarly, Peryam [27] and Eckstein [9] also disapproved

of the linear programming approaches. Their contention was that a diet is optimized at the level of food items or ingredients rather than at the level of menu items or recipes; the solutions of the linear programming were viewed as raw materials and not as end-products. This view is critical, as humans do not consume a specific quantity of each food unit but rather the end-product as a whole unit. Subsequent to this wave of criticism of Stigler's approach, a new type of diet problem, i.e., diet planning, has emerged. The diet problem in this case has been formulated as a combinatorial optimization problem. Consequently, most researchers have applied mixed-integer programming (MIP) to solve the diet problem. Diet planning research has since focused on formulating the diet problem in the MIP form [31, 18, 10] as follows:

$$\max_{\mathbf{X}} \ \text{total nutrition} = \sum_{t=1}^{T} \left( \mathbf{A}^{\mathrm{T}} \mathbf{X} \right)_{nt} \tag{1}$$

subject to

$$\sum_{t=1}^{T} \left( \mathbf{C}^{\mathrm{T}} \mathbf{X} \right)_{t} \leq C \tag{2}$$

$$\sum_{t=1}^{T} \left( \mathbf{S}^{\mathrm{T}} \mathbf{X} \right)_{kt} = T \qquad \text{for} \quad k = 1, 2, 3, ..., K \tag{3}$$

...

$\mathbf{X} = [\mathbf{x}_1, ..., \mathbf{x}_T] \in \{0, 1\}^{M \times T}$ is the matrix representation of a single diet consisting of $T$ menus out of l $M$ available menus. $\mathbf{x}_t \in \{0, 1\}^M$ is the menu representation of $t$th menu in a single diet. This is the one-hot vector of size $M$, the $j$th element of which is marked as one if the $j$th element is the $t$th menu in a diet. The remaining elements are assigned zeros. $\mathbf{A} \in \mathbb{R}^{M \times N}$ is the menu-nutrient matrix in which the value of each element is an amount of nutrient contained in one unit of the menu. $N$ is the number of nutrients. $\mathbf{C} \in \mathbb{R}^M$ is the menu-cost vector in which the value is the unit cost of each menu. $C$ is the upper bound of the total cost available for spending. $\mathbf{S} \in \{0, 1\}^{M \times K}$ is the menu-category indicator matrix in which $s_{mk}$ is set to one if the $m$th menu belongs to the category $k$ and zero otherwise. Note that the subscript of the matrix from equation (1) to (3) indicates an indexing after inner-product operations. For emphasis, the equations are not the absolute forms of formulating the diet planning problem; rather, a dual form can be designed. One example has the objective of minimizing the total cost, given the constraint in which some amounts of the total nutrition can be achieved. Figure 1 in Appendix A.1 illustrates the concept of diet planning.

**Diet planning with ML** In the previous section, we described diet planning as a combinatorial optimization problem; this is the main approach used in OR communities. However, using OR approaches solely is insufficient when a problem is dynamic or comprises latent elements, but ML is an emerging approach that can help overcome the limitations of OR (see Appendix A.1). In this work, we assert that a ML-supported OR approach is the best approach to diet planning. We offer two reasons to support this view. First, diet planning is a task that requires expertise. We discovered from our interviews with diet experts that rules or practices of planning are tightly adhered to. Second, some elements are difficult to define, even among diet experts. Elements such as individual preference that include the color, flavor, or texture of menus and context that includes mealtime, food culture, or composition of the diet, are implicit information such that all possible scenarios cannot be explicitly defined. In summary, the nature of diet planning is essentially static, and various latent elements are highly implicit and difficult to describe explicitly. More specifically, we found in our previous study [17] that dietitians consider the chemistry of the menus, i.e., the composition of a diet, to be as important as meeting the nutritional requirements. We, therefore, propose a diet planning framework that addresses both explicit (nutrition) and implicit (composition) requirements. See Appendix A.4 for further details on these explicit and implicit requirements of diets. The framework consists of OR and ML parts,

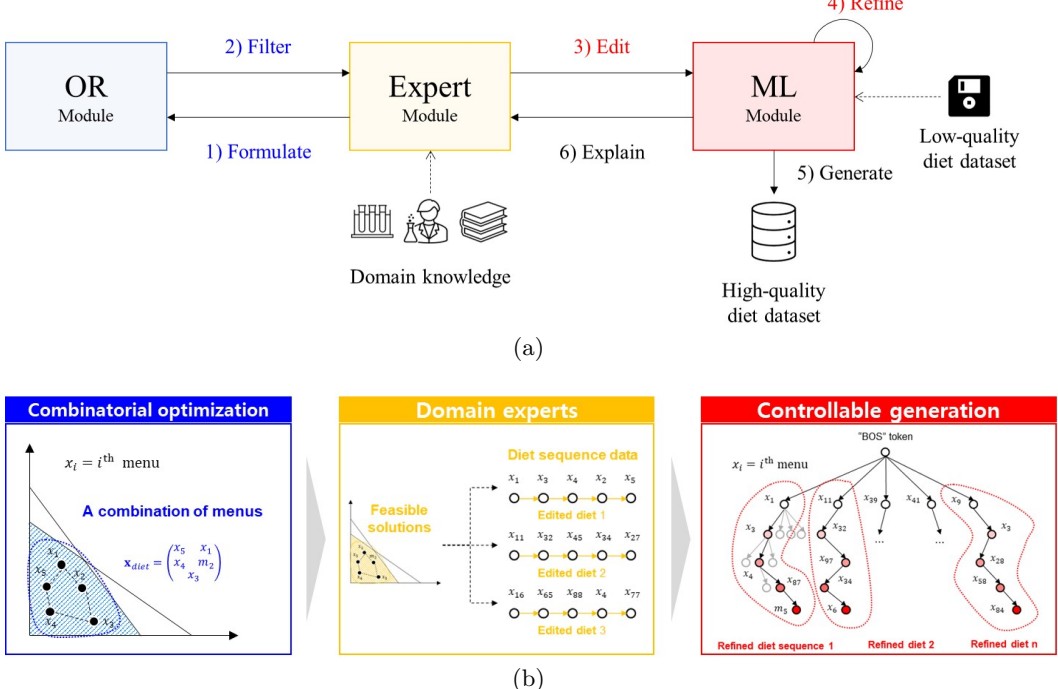

(a)

(b)

Figure 2: OR-Xpert-ML framework (ORxML). Figure (a) shows that the framework consists of three modules and six steps. Steps in blue are related to explicit requirements, and steps in red are related to implicit requirements. Note that this framework is designed to have a cyclical structure that is intended to reflect the continuous interaction between experts and machines. Figure (b) describes the task of each module, e.g., combinatorial optimization and the form of the corresponding diet, e.g., a combination of menus.

each responsible for nutrition and composition; and the outputs are high-quality diet plans addressing both requirements. We elaborate on the details in the next section.

## 3 Methodology

To generate a high-quality diet-level dataset, we propose the ORxML framework. This framework integrates the input of experts with the capabilities of OR and ML. In particular, this section presents the overall framework and then describes each of the three modules in detail.

**The overall framework**   As illustrated in Figure 2(a), the proposed framework consists of three modules and has a cyclic structure of six repeated steps. Each of the three modules has a slightly different view of diet planning. In the OR module, diet planning is a combinatorial optimization problem to find a feasible set of menus that achieves nutritional requirements. In the expert module, the experts define diet planning as a problem that includes composition requirements and occasionally sacrifices meeting nutritional requirements to achieve desirable composition. In the ML module, diet planning is defined as the midpoint between the previous two modules. A machine is trained to control diet generation, allowing this generation to proceed as intended. We intend the algorithm to refine the diets into high-quality ones that recover the nutritional requirements obtained in the OR module while maintaining the compositional requirements provided by experts.

**Six steps for the three modules**   The six steps specify the tasks of each module. In the first step, we **formulate** an OR model to define a searching space. The search space is the space of feasible diets defined as a set of menus that satisfy the nutritional requirements. In

the second step, we **filter** the optimal solutions. An optimal solution is a combination of menus in the feasible space. The aim of this step is to filter the diets that satisfy nutrient requirements and to include these diets as candidates for the initial settings of the diet dataset. The OR module is guaranteed to find existing optimal solutions.

In the third step, the candidate diets are given to experts, e.g., dietitians, who **edit** these diets to ensure acceptability. Note that the experts are guided to edit the diets and arrange the edits into a sequential format. Such a guide is necessary because the following module, the ML module, is designed to learn the sequential patterns of menu compositions in diets. The goal of this step is to create a diet dataset with desirable compositions using the leverage of human experiences. However, manual editing is labor-intensive with the risk of biased editing or mistakes. To automate the editing task and to prevent the risk of human error, we added an ML module in the fourth step to **refine** the input diets. The goal is the compliance of compositional patterns in generated diets and the enhancement of the nutritional rewards. We designed the reward functions based on the nutritional requirements in the National Health Guideline provided by the Ministry of Health and Welfare of the Korean government. Then, based on Lee et al. [17], we developed a sequence generation model that maximizes the rewards using policy-based reinforcement learning with the REINFORCE algorithm [38]. Since reinforcement learning is built on the Markov assumption, the generative model could be based on any type of neural networks that belong to the family of RNNs, e.g., GRU and LSTM. This is powerful in learning sequential patterns and the reason that the experts were guided to edit diets in the sequence form. The ML module is trained by learning the edited diet data from the Expert module and the nutritional knowledge. Through this step, we control the candidate generated diets to be excellent in both compositional and nutritional quality.

The first two steps, i.e., formulation and filtering, are devised to consider explicit requirements; the middle two steps, editing and refinement, are introduced to consider implicit requirements. We can **generate** as many diets as is necessary in the fifth step. With a trained ML model, the process of diet generation is totally automated. In the sixth step, the learned parameters, e.g., the coefficients of the model and the attention map, of the ML model provide an **explanation** of the latent elements, implicit requirements that are unobservable to experts. Finally, these six steps can be repeated as many times as necessary such that the outcomes of the generation and explanation steps may motivate the experts to reformulate the OR model and to improve the diet editing process. See Appendix A.2 for further details of each module.

## 4   Evaluation

In this section, we describe the experiment settings, including the models and algorithms implemented in the OR and ML modules. In addition, we introduce three measures to evaluate the usefulness of the ORxML framework and its outcome, the diet dataset. Finally, we discuss the evaluation results.

**Experiment settings**   For the OR module, we formulated the problem of combinatorial optimization. Then, we solved the problem using the branch and cut method [16, 22], a popular optimization algorithm in OR communities, and found optimal MIP solutions (see Appendix A.3). For the ML module, we define diet planning as a task of neural machine translation (NMT) that maps the source diets, i.e., edited diets from the expert module, into the refined target diets. Furthermore, we applied reinforcement learning (RL) to control the generative translation process such that the translated diet becomes more nutritious than the source diet [17]. All of the details of this approach are provided in Appendix A.2.

We evaluate the quality of diets generated by the modules of the ORxML framework with three measures. First, we count the number of nutrients that satisfy the nutritional standards using the Diet-Nutrition data in MIND (see Figure 6 in Supplementary material C). For the nutritional standards, we referred to the recommended dietary intake (RDI) provided by the Ministry of Health and Welfare of the Korean government (see Table 9 in the Supplementary material). We applied 15 nutritional evaluation criteria and assigned one point each time the diet satisfied a nutritional criterion. Therefore, the perfect score was 15. We named this

measure the RDI score. Note that we also applied the RDI score for the constraint design and reward shaping in the OR and ML modules, respectively. Second, we calculate the ratio of mispositioned menus to evaluate the compositional quality of diets. A menu is considered mispositioned when placed as a side dish but located in the position of the main dish or vice versa. In this study, the diet has a sequence length of $T = 19$ in which each token $x_t$ represents a menu served as a $t$th dish according to a rule of the dietitians' table (refer to Table 15 in the Supplementary material). This means that each menu occupies a feasible position in the diet sequence, and a diet consisting of mispositioned menus is not acceptable. Third, we performed a $\mathcal{X}^2$ homogeneity test in terms of ingredient usages. (According to the diet planning policy developed by professional dietitians, an ingredient-based diet evaluation is important. See Supplementary material B.3 for further details). $\mathcal{X}^2$ evaluates how similar the pattern of ingredient usage is between the generated diets and actual diets. Specifically, the pattern was defined based on the type and frequency of ingredients used for each meal, breakfast, lunch, and dinner. Here, we computed the co-occurrence frequency of ingredients over meals and regarded this as a homogeneity measure between the generated diets and actual diets. If the ingredients usage of each meal in a generated diet is similar to that of an actual diet, then their $\mathcal{X}^2$ value decreases. Given that usage of ingredients represents implicit compositional patterns of the flavors, colors, and textures, we compare the diets generated by each module to the actual diets using the following measure. $\mathcal{X}^2$ measures whether generated diets have the same population in terms of implicit patterns as actual ones.

**Results**   Table 1 shows the experimental result of the diet data generated by the ORxML framework. This table shows the quality of the generated diets compared with the actual diets over the RDI score, % Mispos, and $\mathcal{X}^2$. The RDI score represents nutritional excellence of diets, and % Mispos, mispositioned menu items, and $\mathcal{X}^2$ indicate the compositional compliance with respect to the dishes and ingredients in diets.[3] The results verify that the ORxML framework succeeds in increasing diet quality. As shown in Table 1, the OR module generates diets of perfect nutrition as expected; we provide the average nutrition of generated diets and report the achievement ratio of the nutritional standards in the bracket. This is self-evident considering the characteristics of the MIP model and algorithm we used that guarantees optimal solutions unless there is no feasible one within the constraints.

However, as the % Mispos and $\mathcal{X}^2$ values denote, the compositional qualities of diets generated by the OR module are low. Note that the composition-related criteria shown in Table 1 can cover few aspects of the compositional quality of diets, and designing a metric to measure all aspects of the compositional quality is impossible, especially regarding the implicit requirements of diet planning (see Appendix A.4). Thus, we conducted a survey of 51 professional dietitians to further evaluate the compositional quality in a relatively qualitative way. The survey participants rated the compositional quality of diets generated by the OR module as low. For the expert module, the compositional quality of diets from the OR module could be enhanced by editing the diets into a more realistic form. However, the average nutritional quality declined due to the limited capability of experts to consider nutrition. The ML module recovered the nutritional quality sacrificed by the expert module; this is encouraging and as expected. In addition, the ML module outperformed the composition-related measures. That the ML module can further increase the RDI score after a sufficient training time is significant; we trained our ML model for only 40 hours[4] Additionally, as mentioned in Section 3, our framework is able to provide the experts with explanations of the compositional patterns (implicit requirements) using the attention mechanism. See Appendix A.5 for further details on the explanation of the ML module. Furthermore, in Appendix A.6, we explain the evaluation survey completed by 51 professional dietitians in detail, which is "the human evaluation of our dataset". In summary, the quality of the generated diets was validated by the experts. Furthermore, this qualitative evaluation of the ORxML framework and MIND dataset also showed the necessity of this work in creating and publishing the MIND dataset that incorporates the expertise of domain experts to perform combinatorial optimization and controllable generation.

---

[3]Note that the perfect RDI score is 15.

[4]with an Nvidia Quadro RTX 5000 GPU and Intel(R) Xeon(R) Gold 6136 CPU.

Table 1: Evaluation results of the diet data generated by the ORxML framework

|  | real diets | | OR | | Expert | | ML | |
|---|---|---|---|---|---|---|---|---|
| RDI score ($\uparrow$) | 11.63 | | **15.00** | | 12.26 | | 13.19 | |
| % Mispos ($\downarrow$) | – | | 0.43 | | 0.06 | | **0.05** | |
| $\mathcal{X}^2$ ($\downarrow$) | – | | 6.32 | | 5.70 | | **3.61** | |
| *Energy* | 1359.5 | (68%) | 1383.5 | (100%) | 1314.4 | (62%) | 1321.97 | (72%) |
| *Protein* | 56.16 | (100%) | 53.45 | (100%) | 54.72 | (100%) | 55.66 | (100%) |
| *% Carbo* | 0.61 | (87%) | 0.62 | (100%) | 0.61 | (77%) | 0.61 | (81%) |
| *% Protein* | 0.17 | (100%) | 0.15 | (100%) | 0.17 | (98%) | 0.17 | (100%) |
| *% Fat* | 0.21 | (97%) | 0.22 | (100%) | 0.22 | (94%) | 0.22 | (98%) |
| *Dietary Fiber* | 9.84 | (21%) | 17.52 | (100%) | 12.92 | (74%) | 13.08 | (73%) |
| *Calcium* | 592.6 | (97%) | 612.3 | (100%) | 538.8 | (57%) | 601.25 | (94%) |
| *Iron* | 9.26 | (100%) | 10.74 | (100%) | 9.47 | (100%) | 9.78 | (94%) |
| *Sodium* | 1978.5 | (11%) | 1517.4 | (100%) | 1663.7 | (44%) | 1620.81 | (100%) |
| *Vitamin A* | 445.3 | (87%) | 345.7 | (100%) | 349.7 | (88%) | 374.93 | (100%) |
| *Vitamin B1* | 1.15 | (100%) | 0.97 | (100%) | 0.96 | (100%) | 0.94 | (78%) |
| *Vitamin B2* | 1.32 | (100%) | 1.29 | (100%) | 1.19 | (100%) | 1.27 | (100%) |
| *Vitamin C* | 56.9 | (69%) | 55.56 | (100%) | 61.28 | (87%) | 71.93 | (53%) |
| *Linolenic* | 6210.9 | (82%) | 7407.3 | (100%) | 6965.5 | (82%) | 6796.78 | (90%) |
| *α-Linoleic* | 886.2 | (44%) | 869.3 | (100%) | 938.0 | (61%) | 925.70 | (73%) |
| Time required | – | | 30 min | | 3 weeks $\leq$ | | 40 hours | |
| # of diets | 62 | | 500 | | 500 | | 500 | |

## 5 MIND dataset

MIND is a dataset related to the diet and its constituent elements. We created this dataset based on the ORxML framework. Dietkit is the Python package that provides tools for MIND. MIND is distributed with the dietkit package and can be loaded and manipulated through dietkit.[5] There are four elements that comprise the MIND dataset: Diets, Menus, Ingredients, and Nutrition. Nutrition is the substances absorbed by our bodies through food consumption and includes protein and vitamins. Ingredients are the materials that are used to cook food menus. Menus are the end-products of foods cooked with ingredients. Diets are sequences of menus organized in terms of nutritional and compositional requirements. The detailed relationship between these elements is described in Figure 5 in the Supplementary material B. The MIND dataset is currently available in two languages, Korean and English. The contents of the dataset in each language version are the same. The ingredient data of MIND were extracted from the Standard Food Components provided by the Rural Development Administration of the Korean government. For each ingredient, the data consist of the name, its category, and the quantity of nutrients per 100 g. The dataset consists of a total of 3,036 ingredients; these are classified into 20 categories. Menu data were collected from food service management centers under the Ministry of Food and Drug Safety of the Korean government. A total of 3,238 menus were collected. We prepared the final data after processing the collected data. For instance, the names of the ingredients in the original data were unified, along with the names in the food ingredients data. A total of 527 food ingredients were used in the menu data. The menus were classified into 22 categories according to a policy that can be found in the Supplementary material B.4. Additional information on each menu can be found in the "note" field. This information includes whether the menu is Korean or Western and whether the menu item is a main dish or side dish. Currently, we provide three kinds of diet data: "OR-generated diets", "Experts-generated diets" and "ML-generated diets." Each of the diet datasets is made up of 500 different diets. Each diet dataset consists of 19 menus: five breakfast menus, two morning snacks, five lunch menus, two afternoon snacks, and five dinner menus. Some diets have a slightly smaller number of menus. In this case, we included a dummy menu called "empty" to fit the data form. Detailed information about MIND and dietkit can be found in Appendix A.8 and in the Supplementary material.

---

[5]github.com/pki663/dietkit

# 6    Application of the MIND dataset

Many interesting applications of ML can be developed based on the MIND dataset. While we discuss various uses of the MIND dataset in detail in Appendix A.7, in this section we summarize some of the diet planning and DHR tasks that can be improved with ML using the MIND dataset. First, new interesting embeddings can be created with the MIND dataset for diet planning. For example, a Menu2Vec embedding can be created to represent the compositional patterns of menus in diets; similarly, an attention map can be extracted as exemplified in Appendix A.5. The "identification of alternative or complementary menus" is one of the frequent tasks that dietitians conduct in diet planning; this task is particularly important for those with food allergies. A Menu2Vec embedding extracted from the MIND dataset can be used to develop a system to recommend alternative menus to dietitians, considering their contributions to the nutritional and compositional quality of diets as well as their alternative or complementary relationships in diets. The authors are conducting such a clinical study for children with atopic dermatitis (AD) and food allergy (FA) who must restrict allergenic foods that could lead to fatal anaphylaxis. Precise diet planning and dietary healthcare are necessary to manage the growth and health of these children. In fact, this work on creating a novel high-quality diet dataset was initiated for our clinical study of children with AD and FA; actual diet data in practice were not of sufficient quality to be used to train a machine for the AI service. See Appendix A.7 for further details on these use cases. Note that a high-quality, large-scale diet dataset, such as the MIND dataset, is the basis for these use cases of diet planning and dietary healthcare that could not be conducted without ML.

The MIND dataset has limitations that involve future research issues. First, the diets in MIND are healthy "reference diets" for an unspecified majority of the population. Therefore, there is no guarantee that a user will prefer the provided diets for their meals. Second, the dataset should be extended to multinational and multicultural contexts. See the Supplementary E.2 for further details on our dataset maintenance plan. Third, integrating our MIND dataset with existing databases covering molecules and compounds is necessary [26]. This attempt will allow "precision diet for healthcare" (see Appendix A.7). Finally, a tree or graph structure could be adopted as a diet data structure, and the ML module used to synthesize high-quality diet data could be designed to learn diet data in a graph or tree structure. Nonetheless, the MIND dataset is the first high-quality resource for this research direction, and the ORxML framework can further encourage this research. We hope that future studies will take different approaches and expand the research field of "diet planning and dietary healthcare with ML." We believe the proposed MIND dataset and the ORxML framework will contribute for such work.

# 7    Concluding remarks

Data construction efforts are essential for training machines that can subsequently assist in a variety of human tasks. This work creates a validated dataset to support diet-related human tasks with ML. In fact, our work has already created an impact in solving medical, economic, and social problems associated with diet planning and dietary healthcare. For example, the MIND dataset has been used by the Center for Children's Food Service Management in South Korea beginning in the fall of 2021. This government organization is responsible for the support of daycare centers and kindergartens in Korea that cannot hire professional dietitians due to economic constraints. See Appendix A.7 for further details of these use cases. Furthermore, we did not construct the MIND dataset with manual effort only. We devised a systematic framework that integrates the capabilities of experts and machines to scientifically and efficiently create high-quality data of the complex professional diet planning task. We hope our ORxML framework can be used to inspire and promote dataset preparation methodologies and ML applications for other professional tasks. See Appendix A.7 for a detailed discussion of the value of our ORxML framework.

---

The authors bear all responsibility in the case of a violation of rights.

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
