# Supplementary materials for the MIND dataset

## A    Appendix

### A.1    Relationship between OR and ML

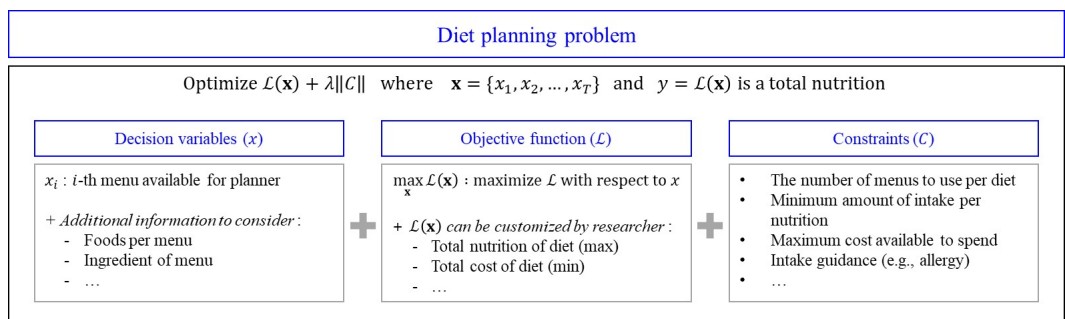

Figure 1: Concept of diet planning

**Limitations of OR**    As mentioned in Section 2, mathematical programming has emerged as the mainstream approach to solving diet planning problems (see Figure 1). Therefore, diet planning has been attempted exclusively using OR in academia. OR is a problem-solving approach that seeks to identify an optimal solution (or a set of optimal solutions) by directly formulating a given problem in mathematical form and applying optimization techniques such as the simplex method [12] or the cutting plane method [23]. After the OR approach became popular, researchers were able to define a problem and formulate the corresponding objective function based on their own interpretation. However, Ackoff [1] argued that OR is inherently limited. OR approaches are valid when the system of the problem is static and all of the systemic elements are tangible. However, in actual situations, the predictions, i.e., the obtained optimized solutions, typically intervene in the system dynamics. Many latent elements are unobservable. As a result, a tendency of researchers is to oversimplify actual situations using obvious, case-dependent elements to allow the application of OR approaches.

**ML as a breakthrough**    ML has been utilized in the past few decades to overcome the limitations of the previous OR approaches and has emerged recently as a promising complementary approach. While OR seeks to optimize variables, ML focuses on parameter optimization. The following is an example of a linear optimization problem.

$$
\begin{aligned}
\text{minimize} \quad & \mathcal{L}_\theta(\mathbf{x}) \\
\text{subject to} \quad & f_\theta(\mathbf{x}) \geq 0
\end{aligned}
\tag{1}
$$

where $\mathbf{x} \in \mathbb{R}^n$ is the real-valued variable of $n$-dimensions and the objective function $\mathcal{L}_\theta(\mathbf{x})$ is a mean-squared-error (i.e., $||y - f_\theta(\mathbf{x})||_2$). Note that $y$ is a target scalar variable and $f_\theta$ is a linear function that maps $\mathbf{x}$ into $y$ using a set of coefficients parameterized by $\theta$ (i.e., $f : \mathbb{R}^n \xrightarrow{\theta} \mathbb{R}$). In the OR approach, the problem can be described as an optimal variable $\mathbf{x}^*$ that minimizes $\mathcal{L}$. The value must be obtained directly in the feasible variable space $\mathcal{X} = \{\mathbf{x}|f_\theta(\mathbf{x}) \geq 0\} \subset \mathbb{R}$ because the parameter is fixed ($\theta = \bar{\theta}$). In contrast, ML takes different approaches. In other words, the implementation of ML is somewhat indirect, but the outputs are generalizable, as depicted by the following:

$$
\theta_{t+1} \leftarrow \theta_t + \frac{\partial \mathcal{L}_{\theta_t}(\mathbf{x})}{\partial \theta_t} \qquad \text{for} \quad ||\theta_{t+1} - \theta_t|| \to 0
\tag{2}
$$

where the parameter $\theta$ is optimized (i.e., the machine is trained) until it converges. This is called the *steepest ascent method*. In terms of convergence, the parameter is assumed to be optimal ($\theta = \theta^*$), and the approximated optimization can be achieved with respect

to (1). This is a benefit of using ML. Moreover, whereas OR seeks to directly obtain the optimal solution $\mathbf{x}^*$ from the original problem, ML seeks to obtain an approximator $f_{\theta^*}$ and generalize the problem domain from the variable space to the parameter space. This manner of generalization allows researchers to address any problem with similar settings, [1] and the data-driven optimization over the parameter space enables them to explore the unobservable elements that constitute latent problems [8].

**ML-supported OR**  ML is complementary to OR, and various endeavors have been pursued on the basis of this combined OR–ML perspective. These endeavors can be categorized into three classes. The first class is called a *two-stage approach* [17]. In the first stage of this approach, ML functions as a parameter estimator. In the second stage, the estimated parameters are used in OR to model the objective function and determine the optimal solutions. The second class is the *end-to-end approach* [27]. This class can be subdivided into two further types of approaches. The first type focuses on the implementation of OR methods within the ML framework by avoiding loss in the end-to-end structure. The second type attempts to develop an end-to-end learning algorithm for discovering the optimal policies that are generalizable to any OR-related problems. The third class represents approaches that fall between the first and second classes. The methods in this class focus on replacing some of the OR components, e.g., initialization of the solutions and search strategies, with ML components; then the components simultaneously interact. Thus, the third class is called the *interactive approach.* The common goal of these three classes is to solve OR-related problems with the support of ML to achieve significantly better performance than pure OR approaches can obtain. Our work belongs to the third class of interactive approaches, and we refer readers to [6, 50, 8] for the details of the OR–ML relationship.

## A.2  Three modules of the ORxML framework

**OR module**  The OR module generates initial diets for use as diet data in the MIND dataset. In this module, we formulate a MIP model and optimize this model to obtain feasible solutions. The solutions are candidate diets arranged as a set, not a sequence, of menus. We formulated the model so that its objective is to maximize the number of menus to be used in each diet under the constraints of explicit requirements, such as nutrition levels, quantity limits, and substitutes or complementary relationships. The formulation method is:

$$\max_{\mathbf{X}} \sum_{m=1}^{M} \sum_{t=1}^{T} \mathbf{X}_{mt} \tag{3}$$

subject to

$$N_i^{(L)} \leq \sum_{t=1}^{T} (\mathbf{A}^{\mathrm{T}} \mathbf{X})_{it} \leq N_i^{(U)} \ , \quad \text{for} \quad i = 1, ..., 12 \tag{4}$$

$$Q_k^{(L)} \leq \sum_{t=1}^{T} (\mathbf{S}^{\mathrm{T}} \mathbf{X})_{kt} \leq Q_i^{(U)} \ , \quad \text{for} \quad k = 1, ..., 12 \tag{5}$$

$$N_{i'}^{(L)} \leq \sum_{t=1}^{T} (\mathbf{A}'^{\mathrm{T}} \mathbf{X})_{i't} \leq N_{i'}^{(U)} \ , \quad \text{for} \quad i' = 1, ..., 6 \tag{6}$$

$$Q_{k'}^{(L)} \leq \sum_{t=1}^{T} (\mathbf{S}'^{\mathrm{T}} \mathbf{X})_{k't} \leq Q_{k'}^{(U)} \ , \quad \text{for} \quad k' = 1, 2 \tag{7}$$

$$\sum_{m=1}^{M} \sum_{t=1}^{T} \mathbf{X}_{mt} = T \tag{8}$$

$$\mathbf{X}^{(p+1)} \sim \{\mathbf{x}^{(p+1)} | \mathbf{x} \notin \mathbf{X}^{(p)}\} \quad \text{for} \quad p = 1, 2, ..., P \tag{9}$$

---

[1]Whereas OR performs a case-dependent modeling, ML is case-free and instead performs data-dependent modeling.

where $\mathbf{X} = [\mathbf{x}_1, ..., \mathbf{x}_T] \in \{0, 1\}^M$ is a set of $T$ menus available out of total $M$ menus ($T$ = 19 and $M$ = 3228); $N^{(L)}, N^{(U)}$ and $Q^{(L)}, Q^{(U)}$ are the lower and upper bounds of the constraints related to nutrition and quantity respectively; $\mathbf{A} \in \mathbb{R}^{3228 \times 14}$ and $\mathbf{S} \in \mathbb{R}^{3228 \times 12}$ are respectively the nutrient-menu and category-menu matrices, consisting of 14 nutrients, 3228 menus, and 12 categories.[2] Note that each category represents the group of menu items (e.g., yogurt belongs to the snack group).

We formulated two types of constraints. The equations from (4) to (5) indicate 24 main constraints, while the equations from (6) to (7) denote the eight sub-constraints. The *main constraint* is a nutritional or quantitative requirement of a single nutrient or category. The *sub-constraint* is a linear combination of main constraints; assume $\mathcal{A}$ is a feasible space constrained by each single nutrient, $u$ and $v$ (i.e., $A_{u\cdot}, A_{v\cdot} \in \mathcal{A}$). $\tilde{\mathcal{A}}$ is a feasible space constrained by multiple nutrients simultaneously (e.g., $A_{u'\cdot} = A_{u\cdot} + A_{v\cdot} \in \tilde{\mathcal{A}}$). Then, $\tilde{\mathcal{A}}$ is a subspace of $\mathcal{A}$ ($\tilde{\mathcal{A}} \subset \mathcal{A}$). We introduced the concept of a sub-constraint to consider a substitute or complementary effect between menus from the nutrient or category perspective. For example, equation (4) indicates that we only consider 12 nutrition constraints for 14 nutrients given that the remaining two nutritional requirements are achieved by other nutrients (see the Appendix A.3 for further details). Equations (8) and (9) are introduced to facilitate the generative process. Equation (8) sets a diet to be generated with a fixed length of $T$, and equation (9) functions to create a diet in bulk.[3] Diet generation is executed by running an optimization algorithm, such as the branch and cut method [29, 38]. We implemented this algorithm using a Julia-based on the Cbc–solver, an open-source program for MIP [16] This MIP model filters the diets that satisfy the constraints from (4) to (9).

**Expert module**   After filtering the initial diets generated by the OR module, we recruited several professional dietitians to evaluate and adjust the initial data in the expert module. The experts edited the diets to be more acceptable in terms of implicit requirements, e.g., the composition of diets. The edit step consists of two tasks: arrangement and replacement. In the first task, experts arrange a set of menus in sequence. Such a task is quite natural considering that diets are generally perceived as a practice or rule of food consumption that follows a certain order. Furthermore, this approach is important in that the approach separates from the traditional nutrient-only view of the diet problem. This assists in overcoming the limitations that arise with OR approaches (see Appendix A.1). Moreover, this approach provides us with an opportunity to utilize a family of ML techniques for sequence data (e.g., seq2seq [49]). In the second task, experts replace some menus with their alternatives. The intent is to develop more aesthetically desirable diets by deleting inappropriate menus and inserting alternative menus. An edit was carried out rigorously according to the five standards introduced by the experts. Table 1 shows an overview of the considerations in determining whether to edit a diet. See the Supplementary material (Refer to B.3. for planning and diet data generation and the details on the considerations that the experts used in the diet editing process.

Table 1: Five standards for editing

| Number | Reason to edit | Decision |
|--------|----------------|----------|
| 1 | Improper use of menus | |
| 2 | Weight correction required | Edit |
| 3 | Recipes need changes | |
| 4 | Duplicated ingredients | |
| 5 | No reason to edit | No edit |

---

[2] Each $k$th category is labelled as follows: $k = 1$ for *rice with soup*, 2 for *rice*, 3 for *soup*, 4 for *side dish*, 5 for *main side dish*, 6 for *snack*, 7 for *empty*, and so on. The *empty* category contains an *empty* menu only and is introduced to diversify the combinations of menus.

[3] During a diet generation with a total of $P$ epochs, the menus of every $p$-th diet $\mathbf{X}^{(p)}$ are not involved in the $(p+1)$-th diet $\mathbf{X}^{(p+1)}$. This prevents a single unique diet from being established and creates multiple diets instead.

**ML module**  With the expert module, we can treat a diet as a sequence of menus $\tau = [x_1, x_2, ..., x_T]$. In this study, the diet has a sequence length of $T = 19$, and each token $x_t$ represents a menu served as a $t$th dish.[4] The 19 tokens represent a schedule of servings, and the schedule consists of three meals, e.g., breakfast, lunch, and dinner, and two snacks, e.g., a morning snack and an afternoon snack. The diet sequence of 19 or less menu items is the standardized form of daily diets for children that have been used by the Center for Children's Food Service Management in South Korea and the Ministry of Food and Drug Safety. Given that our research objective is to create a "standard" high-quality daily diet dataset, we applied this standardized form. In meal and snack servings, the menus are listed according to a writing rule of the diet table. That is, each diet sequence is defined as an array of menus that imitates a diet table aligned by a serving schedule. However, the status of diets resulting from manual editing is vulnerable to human error such as bias or inertia. As evidence, we found that the nutrition requirements guaranteed by the OR module are damaged by the editorial process (see Section 4). This implies that the presence of implicit requirements is difficult to capture and often sacrifices satisfaction of explicit requirements. Therefore, we considered the necessity of the ML module to alleviate such a risk and refine the diets into high-quality ones.

The focus of the ML module is obvious. The first function is the necessity to recover the nutritional level of the diet sacrificed in the editorial process. Second, we are required to implement a machine that controls the recovery of explicit nutritional requirement while maintaining compliance with the implicit requirements. Third, the machine should be trained using edited diets. The edited diets already achieved both a desirable nutrition level and composition to some degree. In this context, we defined the task of the ML module as a controllable sequence generation with an objective function as follows:

$$
\begin{aligned}
\max_{\pi_\theta} J(\theta) &= \mathop{\mathbb{E}}_{x \sim \pi_\theta} \left[ \sum_{t=1}^{T} \gamma^t r(x_t, x_{t+1}) \right] \\
&\approx \sum_{\substack{\tau \sim \mathcal{H} \\ \hat{\tau} \sim \pi_\theta(\tau)}} \left[ \sum_{t=1}^{T} \pi_\theta(x_{t+1}|x_{0:t}) r(\hat{\tau}|\tau) \right]
\end{aligned}
\tag{10}
$$

where $x_t$ is a $t$th menu token; $\pi_\theta$ is a diet generator parameterized by $\theta$. In this study, we defined the diet generator as a deep neural network having a seq2seq framework [49] built on the gated recurrent unit (GRU) [10] updated by the REINFORCE algorithm [57]; the generative process is defined as consecutive predictions of $\hat{x}_t$ for $t = 1, 2, ..., T$ to generate the refinements $\hat{\tau}$; $\tau$ is an edited diet suggested from human resources $\mathcal{H}$; $\hat{\tau}$ is a refined diet generated by $\pi_\theta$; $r(\cdot)$ is a reward function that returns a reward, a numerical value which measures the RDI score of diet. The RDI is shorthand for "recommended dietary intake" that is an explicit nutrition guide to follow; Note, that a reward is returned only once when the end token $x_T$ is observed, because we can obtain an RDI in terms of a complete diet not a part of diet. Meanwhile, we defined a reward function in tricky way:

$$
r(\hat{\tau}|\tau) = \frac{r(\hat{\tau})r(\tau|\hat{\tau})}{r(\tau)} = r(\hat{\tau}) \times \frac{r(\tau, \hat{\tau})}{r(\tau)r(\hat{\tau})} \approx r(\hat{\tau}) \times \rho(\tau, \hat{\tau})
$$

to increase the sample diversity. In controllable sequence generation, the sample diversity means that the machine has more examples to use for training, and it enables a generative process that is more controllable with rich representations (and thereby enables a policy generalization.). As equation (10) shows, there are two independent sampling processes, of which one is for edited diets, $\tau \sim \mathcal{H}$, and the other is for refined diets, $\hat{\tau} \sim \pi_\theta(\tau)$. Then, it is obvious that we can make diverse generations with a joint sampling space such as $\mathcal{H} \bigcap \pi_\theta$. Since $\pi_\theta$, i.e., a sampling distribution or policy, only changes according to rewards, we modified a reward function to force $\pi_\theta$ near to $\mathcal{H} \bigcap \pi_\theta$. Beginning from a conditional reward $r(\hat{\tau}|\tau)$, we derived the reward of refinements $r(\hat{\tau})$, multiplied by the reward correlation between edited and refined diets $\rho(\tau, \hat{\tau}) = \frac{r(\tau, \hat{\tau})}{r(\tau)r(\hat{\tau})}$. We defined $\rho(\tau, \hat{\tau})$ to be the ratio of menus that overlap between an edited diet and its corresponding refined diet. This forces

---

[4]Note that we did not count the indicative tokens, i.e., "BOS" and "EOS". These designate the begin and end of a sequence, respectively.

the refined diets to become as similar as possible to the edited diets. Thereby, it helps in maintaining the implicit requirements introduced by the editorial process of humans. In summary, Equation (10) indicates that the goal of the objective function is to approximate $\pi_\theta$ which maximizes $J(\theta)$, and $\pi_\theta$ is controlled to generate $\hat{\tau}$, with a reward that is greater if it complies with a composition judged by experts. Then, the controllable diet generator is optimized using the Adam optimizer [26]:

$$\theta \leftarrow \theta - \alpha\left(-\nabla_\theta J(\theta)\right) = \theta + \alpha \nabla_\theta J(\theta) \tag{11}$$

where

$$
\begin{aligned}
\nabla_\theta J(\theta) &= \sum_{\substack{\tau \sim \mathcal{H} \\ \hat{\tau} \sim \pi_\theta(\tau)}} \left[\sum_{t=1}^{T} \nabla_\theta \log \pi_\theta(x_{t+1}|x_{0:t}) r(\hat{\tau}|\tau)\right] \\
&= \mathbb{E}_{\substack{\tau \sim \mathcal{H} \\ \hat{\tau} \sim \pi_\theta(\tau)}} \left[r(\hat{\tau}|\tau)\nabla_\theta \log \pi_\theta(\tau)\right].
\end{aligned}
\tag{12}
$$

To implement and execute equation (12), we applied the *teacher-forced REINFORCE* algorithm (TFR) suggested by Lee et al. [30], which is one of the most recent papers that address the diet planning problem with ML leverage.

## A.3 Details of MIP formulation in the OR module

In this section, we introduce some techniques and more details concerning the OR module. For a typical OR problem, a unique solution is found with an objective function that minimizes or maximizes values such as cost. However, unlike the usual problem, we needed to generate different diets satisfying the conditions and used a slightly different method to accomplish this. As shown in Equation (3), we used an objective function that maximizes the number of menus used in the diet, but we have fixed the number of menus in the diet to 19 as a constraint. We prevent using the menu of the previous diet, as shown in Equation (9), to avoid the repeated usage of a specific menu. Specifically, we calculated the appearance frequency of menus in generated diets and selected multiple menus to remove at the next generative process. The number of the menus to remove can be customized, and we removed ten menus in every epoch. In this way, we could continuously generate a new diet that satisfies nutritional and basic composition constraints. A more detailed description of the two equations (6) and (7) follows. As shown in Table 9 in the Supplementary material, there are three requirements for the percentage of carbohydrates, protein, and fat. The reason for this is that the proportion of each of these nutrients is affected by the number of calories provided by each of these nutrients. Thus, in order to make a linear inequality with a predefined value, we reorganized our formula below into a different form:

$$0.55 \leq \frac{4\sum_{t=1}^{T}\left(carbohydrate^{\mathrm{T}}\mathbf{X}\right)_t}{\sum_{t=1}^{T}\left(calorie^{\mathrm{T}}\mathbf{X}\right)_t} \leq 0.65$$

$$0.07 \leq \frac{4\sum_{t=1}^{T}\left(protein^{\mathrm{T}}\mathbf{X}\right)_t}{\sum_{t=1}^{T}\left(calorie^{\mathrm{T}}\mathbf{X}\right)_t} \leq 0.20$$

$$0.15 \leq \frac{9\sum_{t=1}^{T}\left(fat^{\mathrm{T}}\mathbf{X}\right)_t}{\sum_{t=1}^{T}\left(calorie^{\mathrm{T}}\mathbf{X}\right)_t} \leq 0.30$$

By rearranging the equations, we converted three inequalities to six inequalities, as shown below:

$$0 \leq \sum_{t=1}^{T} \left( (4carbohydrate - 0.55calorie)^{\mathrm{T}} \mathbf{X} \right)_t$$

$$\sum_{t=1}^{T} \left( (4carbohydrate - 0.65calorie)^{\mathrm{T}} \mathbf{X} \right)_t \leq 0$$

$$0 \leq \sum_{t=1}^{T} \left( (4protein - 0.07calorie)^{\mathrm{T}} \mathbf{X} \right)_t$$

$$\sum_{t=1}^{T} \left( (4protein - 0.20calorie)^{\mathrm{T}} \mathbf{X} \right)_t \leq 0$$

$$0 \leq \sum_{t=1}^{T} \left( (9fat - 0.15calorie)^{\mathrm{T}} \mathbf{X} \right)_t$$

$$\sum_{t=1}^{T} \left( (9fat - 0.30calorie)^{\mathrm{T}} \mathbf{X} \right)_t \leq 0$$

where a coefficient matrix $\mathbf{A}'_{i'j} \in \mathbb{R}^{3228 \times 6}$ is the subspace of matrix $\mathbf{A}$, and the six columns of $\mathbf{A}'$ is the stack of coefficient vectors in the six inequality constraints above. Note that the coefficients denoted as the name of each nutrient is a vector of length $M$ whose each element represents a value of the corresponding nutrient.

Since there is a combo menu that can substitutes for rice and soup at once, we made an equation (13). In this case, the rice and soup should always be used at the same time so that we added constraints (14) to fix the number of rice and soup equal to each other within a single diet. These two equations are reflected in Equation (7) in the form of $\mathbf{S}'_{k'j} \in \mathbb{R}^{3228 \times 2}$, the subspace of matrix $\mathbf{S}$.

$$\sum_{t=1}^{T} \left( (2ricewithsoup + rice + soup)^{\mathrm{T}} \mathbf{X} \right)_t = 6 \tag{13}$$

$$\sum_{t=1}^{T} \left( (rice - soup)^{\mathrm{T}} \mathbf{X} \right)_t = 0 \tag{14}$$

## A.4 Details of the requirements and the difficulties of diet planning

**Explicit and implicit criteria of diet planning**  Many countries define nutritional standards that are tailored to the characteristics of their own culture. Such standards are established based on studies of dietary patterns and citizen health status. The goal of establishing such standards is to prevent excesses or deficiencies of any dietary nutrient. This is accomplished by encouraging the intake of nutrients that are generally consumed in insufficient quantities. Equally, the standards are designed to strictly limit the intake of those that are consumed in excess of nutritional requirements, ultimately helping the citizens lead healthier lives. Thus, one of the responsibilities and missions of dietitians is to plan diets that comply with both cultural and nutritional standards. However, in practice, difficulty arises in planning diets based on these nutritional standards. The reason for this is that individuals do not just intake individual nutrients; our foods contain a variety of nutrients that must be balanced as a whole in our diets. Meeting nutritional requirements would be relatively easy if individuals consumed nutritional supplements with limited quantities of food. However, consumption patterns do not allow this; the combinations of nutrient quantities in individual diets are complex when considering the variety of foods consumed. Because of the difficulty in balancing nutritional intake, dietitians plan diets using a method called a "food guide." The food guide classifies types of foods into groups based on their nutritional characteristics; those having similar nutrients are classified in the same group.

Based on these food groups, grains, meat/fish/eggs/beans, vegetables, fruits, milk/dairy products, and fats/oils, dietitians generate diets. These diets are designed to meet nutritional standards by determining the frequency of consumption of items from each food group. This was the method used in developing the 2019 guideline for diet planning provided by the Center for Children's Food Service Management.

However, diets generated in this manner only roughly meet nutritional standards. Upon close examination, diets planned this way were found to result in a deficiency or excess of specific nutrients when compared to the Korean Dietary Reference Intakes. Furthermore, diets used in schools or food services for childcare centers only manage the children's nutrient intake during their stay at the institutions [3]. Children that do not have access to such food services provided by the institutions, for example, those homebound due to COVID-19-induced difficulties in face-to-face education or those with food allergies, are without these planned diets. While parents generally attempt to serve their children balanced diets, the difficulties faced by professional dieticians in balancing diets are compounded for parents [22].

The diet composition, the harmony among menus, is the most important factor for dietitians in designing and planning diets [46]. In diets, menus are considered to be in harmony if various textures, colors, and food groups are represented without significant overlap [54]. The foods provided in a diet should complement or enhance one another in taste, smell, texture, and nutritional value. The ability to determine the nutritional component is solely based on the professional background knowledge of the dietitian planning the diet, and dietitians plan diets based on their estimation of the nutritional content of different foods. Dieticians' personal dietary patterns, preferences, and tendencies may be reflected in their diet planning. Knowledge of the exact nutritional value of every ingredient is not possible. This is the primary reason that "diversity" is used and emphasized in maintaining harmony when planning diets[25]. The use of ingredients from various food groups and colors minimizes the risk of an excess or deficiency of any one nutrient, and consumption of a variety of foods results in intake of a variety of nutrients.

However, composition satisfaction can limit the provision of diets with high nutritional qualities as the goals of satisfying the nutritional and compositional standards pose restrictions on each other. Efforts to satisfy more standards limit the use of various ingredients; consequently, a pattern may arise in which a certain nutritional standard can only be satisfied by including a specific food that contains a large amount of the corresponding nutrient. Adding variety to the diet composition, therefore, increases the difficulty in satisfying certain standards. While the harmony found in menus composed by dietitians can provide variety and consumer satisfaction, these diets, planned based on estimated nutrient content, may be limited in their ability to satisfy nutritional standards.

**Values and implications of the MIND dataset for nutrition and healthcare** Given the requirements and difficulties of diet planning, dietitians and their planned diets are more affected by the composition of the diet and the preferences of the recipients than by the nutritional standards. Although dietitians are aware that the diets should satisfy nutritional standards, there are often no options but to plan diets based on these other factors, even if the resulting diets do not satisfy the standards. Dieticians know this, and artificial intelligence (AI) may be able to solve this. AI offers the possibility of diet provision that satisfies nutritional and compositional standards. In additional to providing a nutritionally balanced diet, these diets can introduce a variety of nutrient-supplying foods, avoid overlaps of recipes or ingredients, maintain harmony in the tastes/colors/forms of foods, and fulfill consumer preference requirements. Dieticians would be able to redirect their efforts and focus on other job requirements such as hygiene or nutritional education. The positive effect that AI-generated diets would have on the growth and health of infants is especially important as parents would be able to provide higher-quality diets that fit the nutritional standards every day.

The MIND dataset is significant in that the dataset was planned through the coordination of AI systems and human dietitians with the goal of providing diets that satisfy both nutritional standards and harmonious composition requirements. With an optimization model, the draft data were developed to fulfill most of the nutritional requirements for children aged 3–5

years. Attempting to satisfy all of the nutritional standards generally limits the range of ingredients or foods that can be used. However, despite such difficulties, this MIND dataset work demonstrated the ability to create harmonious compositions while incorporating food variety. The MIND dataset will help many dietitians plan diets that maintain quality in both nutritional and aesthetic composition.

## A.5 Explanations of the ML module in the ORxML framework

As mentioned in A.2, we developed an ML model based on the seq2seq framework with GRU units. GRU is in the family of recurrent neural networks (RNNs) that are widely used to address sequential information using backpropagation through time (BPTT) [56]. However, RNN-based models with BPTT have a chronic problem of an information bottleneck that causes a memory loss, and an attention mechanism [4] was proposed to overcome this problem. Attention is a technique that makes a model recognize the parts of the sequence on which to concentrate, and the attention map allows visualization of the distribution of focuses of attention.

Figure A.5 shows the attention maps extracted from the experiment in Section 3. As shown, these maps explain the menus on which to focus for the improvement of nutrition. Here, the x-axis is the input, an edited diet, and the y-axis is a refined diet, the output. The highlighted cells provide explanations regarding the menus that should be considered as candidates for replacement. Note that the highlight implies that the replacement of menus in that cell will recover the nutrition of the diet. Using this method, the experts can expand and advance their knowledge required to improve the diets being designed and to reformulate the OR module if necessary.

## A.6 Details on the evaluation of the MIND and the ORxML framework

**Expert evaluation**   As of August 2021, the evaluation experiment in Section 4 is the latest experiment, and the MIND dataset published in this work involves the outcomes from this experiment. These outcomes are the diet data from the OR module, Expert module, and ML module. Additionally, we conducted previous rounds of the evaluation experiment of the MIND dataset development process. This included the conduct of a survey of 51 professional dietitians with an average job experience of 4.85 years (range of 1 to 12 years). Although the nature of an expert survey could be qualitative in nature, the purpose of our survey was to evaluate the compositional quality of the diets synthesized by the ML module. This was required because the design of a metric to measure all aspects of the compositional quality of diets, the implicit requirements of diet planning, is impossible (see Appendix A.4). Note that the composition-related criteria shown in Table 1 in Section 4 covers few aspects of the compositional quality.

The 51 professional dietitians evaluated the 60 diets designed by the OR module, OR+Expert modules, and OR+Expert+ML modules. Table 3 shows results regarding the evaluation criteria described in Table 2 [7]. The results suggest three issues. First, experts may be biased primarily toward composition satisfaction in dietary evaluation and then apply this bias when evaluating other aspects such as the nutritional content and overall satisfaction. For example, the experts did not give a high nutrition score to the nutritionally perfect diets generated by the OR module. This was perplexing. Second, following this example, most of the experts were not capable of precisely evaluating the nutritional quality of diets. As shown in Section 4, the OR module and the OR+Expert+ML modules should be superior to the OR+Expert modules in nutritional excellence. However, Table 3 shows that the experts could not evaluate the nutrition of diets; this may well be due to limited calculation abilities and the biases mentioned above. Finally, although the OR+Expert+ML generated the diets to be compositionally less adequate than the human-designed diets, this result is natural because any machine-generated diet was new and unfamiliar to the experts. Also, the experts were biased by the human-designed diets publicly disclosed as a reference database by the government. The important question is whether the composition of machine-generated diets is acceptable for actual food provision. As shown with the reliability score in Table 3, we received feedback from the survey participants that the diets generated by the OR+Expert+ML modules can be used in practice. In addition, as shown in Section

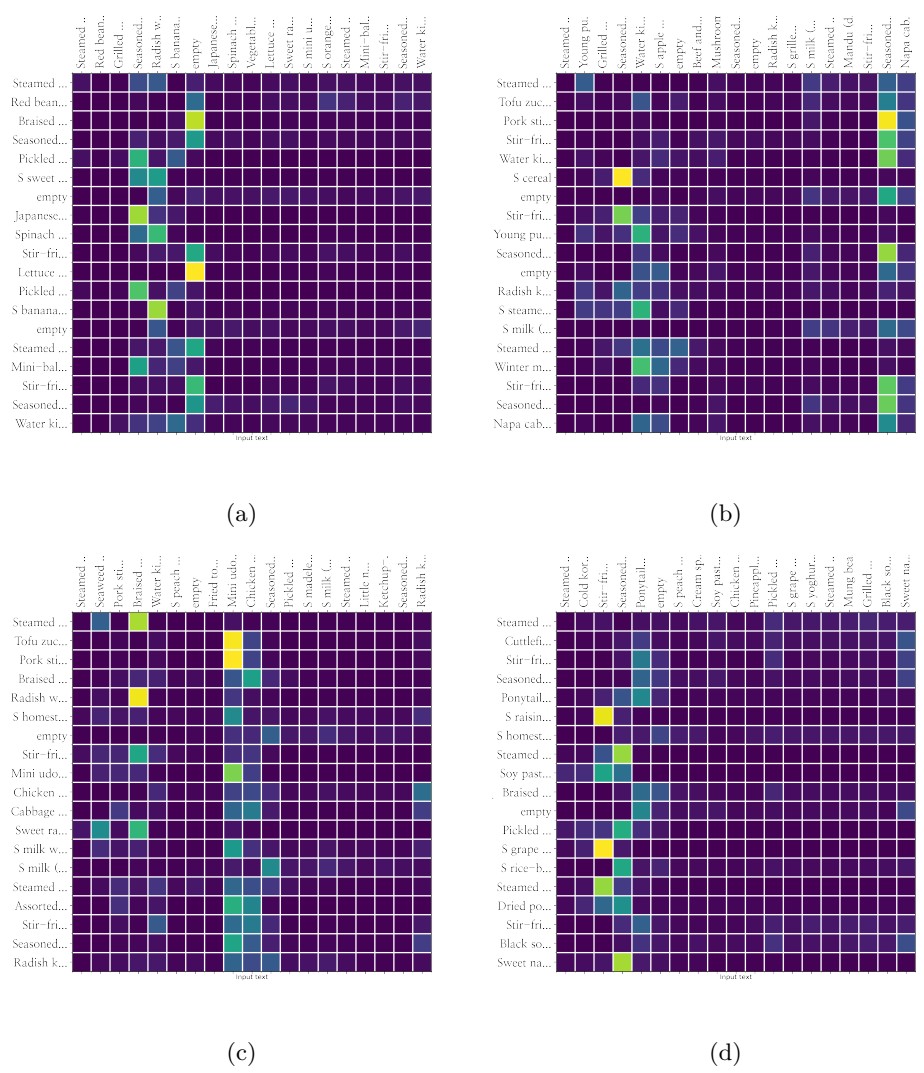

(a)    (b)

(c)    (d)

Figure 2: Attention maps of the ML module: An edited diet sequence (i.e., an input) provided by the expert module is marked on the x-axis while the y-axis denotes its counterpart, a refined diet sequence (i.e., an output), generated by the ML module.

4, the OR+Expert+ML modules demonstrate better performance than experts in terms of ensuring the nutritional quality of diets.

In summary, the experts considered composition compliance as the most important factor in evaluating diets in addition to being incapable of accurately evaluating the nutritional quality of diets. This lack of capability confirms the motivation of this work, to create and publish the MIND dataset using combinatorial optimization and controllable generation as no high-quality diet-level dataset exists at present due to the extreme complexity of diet planning. The survey findings confirm the necessity of the OR and ML modules in our ORxML framework, and the importance of composition compliance confirms the irreplaceable role of domain experts, professional dietitians. These experts are required for the consideration of the implicit requirements of diet planning and data generation, thereby confirming the necessity of the expert module in our ORxML framework.

Table 2: Form of survey

| Section | No. | Evaluation criteria | Scoring |
|---------|-----|---------------------|---------|
| Nutrition | 1.1 | Does this diet satisfy the nutrition standards in terms of calories? | |
| | 1.2 | Does this diet satisfy the nutrition standards in terms of a balance of carbohydrate, protein, and fat? | |
| | 1.3 | Does this diet satisfy the nutrition standards in terms of vitamins and minerals? | Integer between 1 and 5 |
| Composition | 2.1 | Is this diet balanced in terms of the food groups? | |
| | 2.2 | Does this diet harmonize in terms of the menus in meals? | |
| | 2.3 | Is this diet balanced in terms of the cooking methods? | |
| | 2.4 | Does this diet harmonize in terms of the menus in snacks? | |
| Overall reliability | 3.1 | Do you think this diet is suitable for a real service? | Yes (1) or No (0) |
| | 3.2 | Do you think this diet was planned by a professional dietitian? (Turing test) | |

Table 3: Result of survey

| Questions | Score of the evaluation criteria | | | | | | | | |
|-----------|------|------|------|------|------|------|------|------|------|
| | 1.1 | 1.2 | 1.3 | 2.1 | 2.2 | 2.3 | 2.4 | 3.1 | 3.2 |
| *Real* | 4.38 | 4.15 | 3.97 | 3.96 | 3.87 | 3.85 | 3.70 | 3.62 | 0.67 |
| *OR* | 3.75 | 3.12 | 3.52 | 3.25 | 2.56 | 3.17 | 2.38 | 2.19 | 0.15 |
| *Expert* | 4.30 | 4.01 | 4.03 | 4.04 | 3.81 | 3.93 | 3.83 | 3.61 | 0.68 |
| *ML* | 4.26 | 3.92 | 3.80 | 3.80 | 3.61 | 3.82 | 3.39 | 3.29 | 0.55 |

**A.7  Use of the MIND dataset and the implications of the ORxML framework**

**MIND for ML applications in dietary healthcare**  As mentioned in the introduction, a balanced diet is important for all. This includes children and senior citizens, the well and the sick . Thus, diet planning has emerged as a core part of healthcare research in a variety of fields including food technology [36], nutrition management [13], clinical medicine [61], sports science [5, 28], and military nutrition [40, 20]. The MIND dataset can be used in all of these fields.

Using dietary data in the MIND dataset, ML can design diets that counter disease-related factors [61, 33, 47, 2] or to identify dietary factors that contribute to the strengthening of physical abilities and improvement of metabolic controls that are important in sports and military contexts [21, 14]. MIND, a large-volume dataset accessible to the public, is the first benchmark dataset for ML-based dietary healthcare studies.

Our work has already created an impact. The Center for Children's Food Service Management in South Korea will use our outcomes beginning in the fall of 2021. In addition, a startup company in South Korea is using this dataset for the development of a gut microbiome-personalized diet recommendation AI system for children with atopic diseases and food allergies. This service will be distributed under the support from the Ministry of Science and ICT. The authors have also started a government project to use this dataset for the development of a precision diet service system in which the users, e.g., dietitians and physicians, can work with the application interactively for precise diet planning for recipients. This service will be distributed under the support from the Ministry of Food and Drug Safety and the National Research Foundation of Korea. The following paragraphs more specifically illustrate MIND dataset utilization for ML-based dietary healthcare studies and applications.

**Task 1: Complementary healthy menu recommendation**  Typically, people spend much of their day away from home and consume more than half of their daily energy intake

away from home. Therefore, identifying complementary in-home and out-of-home menus is important. This is particularly important for growing children. Because of the increase in the number of working mothers in developed countries, the number of children attending daycare centers is significantly large. The participation rate in Korean daycare centers increased from 51.7% - 69.2% in 2010 to 53.8% - 88.8% in 2017. Therefore, parents need to be able to produce home menus for their children that nutritionally complement the diets provided in daycare centers. However, as mentioned in the introduction and Appendix A.4, this task is difficult because of the required nutrient and growth knowledge and the high complexity of design associated with large numbers of food items.

Using the MIND dataset, an application can be trained to generate healthy menu recommendations inside the home, outputs from inference, complementary to the menus consumed outside the home, inputs for inference. For example, an incomplete sequence of morning snacks, lunch meals, and afternoon snacks can be input to the algorithm, and the outputs will be a complete daily diet sequence that adds the complementary healthy menus for breakfast and dinner. We will distribute the MIND dataset to parents in South Korea through the Center for Children's Food Service Management in South Korea beginning in the fall of 2021. In addition, beginning in June 2021, we started a government project to use this dataset for the development of a precision diet service system in which the users, e.g., parents and pediatricians, can interactively work with the application for precise diet planning for children, including complementary healthy menu identification. This service will be distributed under the support from the Ministry of Food and Drug Safety and the National Research Foundation Korea.

**Task 2: Allergy-free safe diet planning**  Vegetarians and food-allergic patients need to prepare diets customized to their unique needs [45]. Recently, there is an increase in the number of children with food allergies [32], and one of the most difficult tasks in daycare centers is to provide special care to these children. For these children, dietitians consider food allergen elimination in the meal and prepare alternative food menus tailored to meet nutritional needs in the absence of potential allergens. In addition to menu considerations, care needs to be taken to avoid cross-contamination during the cooking process. In the case of vegetarian children and adolescents, some vegetarian diets may be low in specific nutrients, such as calcium and vitamin B12, and their inclusion is needed to ensure proper growth and development [34]. AI is expected to reduce the time- and resource-consuming processes related to diet design and clinical nutrition.

Using the MIND dataset, AI can be trained for safe, allergen-free diet planning. A controllable generation algorithm that is similar to the ML module in the ORxML framework can be designed using reward-shaping for explicit requirements to reflect the needs of food-allergic patients. This algorithm would give negative rewards for restricted ingredients and nutrition and give positive rewards for recommended ingredients and nutrition. A startup company in South Korea has already begun to develop such an algorithm for a gut microbiome-personalized diet recommendation AI system for children with atopic diseases and food allergies. In this service, parents request a gut microbiome examination for their children, and the service identifies the type and conditions of the child's microbiome. Then, the controllable generation algorithms that are pre-trained for specific types of children's microbiome conditions are used to generate safe, allergen-free diets for the children in a customized manner. This service will be distributed under the support of the Ministry of Science and ICT. As pioneered by Zeevi et al. (2015) [61], this kind of precision diet service with ML will have considerable value in the prevention and management of chronic diseases.

**Task 3: Menu2Vec embedding for menu recommendation in diet planning and dietary healthcare**  Figure 3 shows that new interesting embeddings can be created with the MIND dataset for diet planning and DHR. In Figure 3, the relative position among menus in the T-SNE map changes according to different embeddings. The Menu2Vec embedding represents the compositional patterns of menus in diets, and the Menu2Vec embedding is concatenated with the nutrition features. To obtain the Menu2Vec embedding, we applied the Word2Vec [35] to the generated diets. Each point in the T-SNE map is a menu, and the color indicates the cluster to which the menu belongs. Cluster assignment was determined based on the affinity propagation [18] with the Euclidean distance in terms of nutrition. As

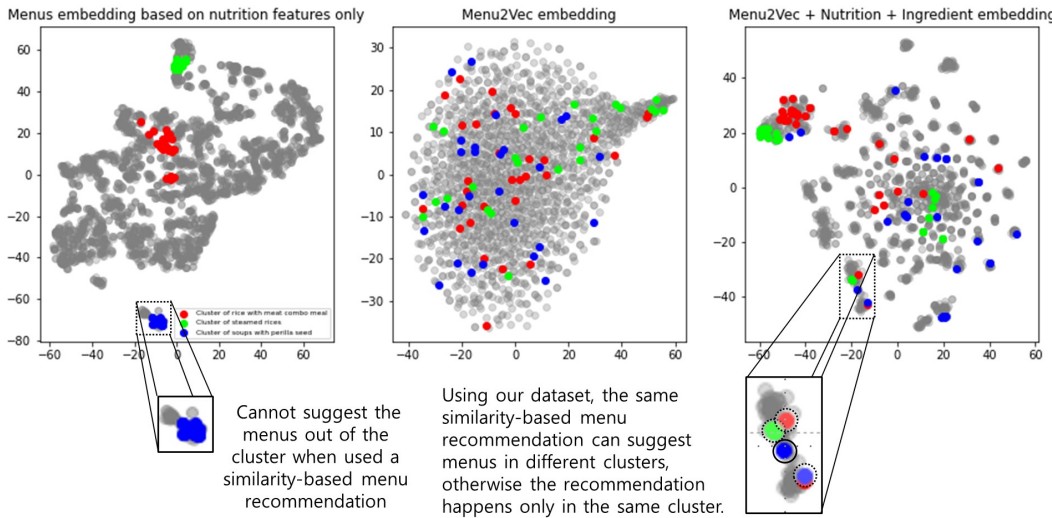

Figure 3: Menu embedding using the MIND dataset and the visualization of T-SNE map.

such, each of the three figures illustrates a cluster distribution of menus that are identified with the original nutrition features only.

The left figure shows that menus in the same cluster are distributed near one another. This implies that the menus of the same cluster are positioned so close in the embedding space that the choice of alternative menus, i.e., the choice of menus in different clusters, is strictly limited. However, the central figure shows that the menus of generated diets are randomly distributed regardless of the nutrition cluster. The generated diets always guarantee high nutritional quality (see Table 1 in main paper), but menu embedding is not solely dependent on nutritional criteria. Therefore, simply replacing a menu with its nutritional counterpart, i.e., replacing with substitutes in the same cluster, the usual practice of dietitians, does not always provide a nutritionally better diet. Last, the right figure shows a visual demonstration of possible scenarios in which the dietitians use the MIND dataset for actual applications. For example, some dietitians design diets for patients suffering from "*perilla seed*" allergy, a common allergy usually discovered when the patient first consumes a perilla oil. These dietitians can slightly modify a reference diet from the MIND dataset, replacing a menu that contains perilla seeds with its similar, perilla-free substitutes. To support dietitians, we can develop a menu recommender system that suggests perilla-free menus by calculating the similarity between all menus. The similarity-based menu recommendation has been addressed in many previous studies [60, 52, 43, 53]. However, menus with nutritional value are likely to have similar ingredients. Therefore, a similarity-based recommendation may fail to suggest a perilla-free substitute that maintains a similar nutritional level. In this context, the MIND dataset is potentially valuable. Specifically, if a menu recommender system is trained on our dataset and calculates a similarity based on a Menu2Vec embedding, then the system can successfully suggest perilla-free menus by simply executing a similarity-based recommendation that maintains the same quality of nutrition.

Table 4 shows the result of a similarity-based recommendation for perilla-free menus. The *perilla seeds* were assumed to be an allergy-causing ingredient and *seasoned dried radish leaves and perilla seed* to be a target menu. The similarity was computed based on the embedding vector of nutritional feature and the concatenated embedding vector of Menu2Vec + Nutrition + Ingredient. Note that we purposely concatenated the ingredient feature to degenerate a performance of the recommender system; if the ingredient feature is concatenated, then the top-ranked recommendations will always include perilla-related ingredients. The similarity-based recommendation will be likely to suggest a menu that still contains the allergen. Concatenated embedding would obviously have a better recommendation. As shown in the table, the recommendation made on the basis of nutrition features alone completely failed; not only did the suggested menus still contain perilla seeds, but the menu category also

changed. (The target menu belongs to the side dish category, but one of the suggested menus belongs to the soup category.) To force a recommender system to not suggest the menus with the allergen, we added a filtering step prior to similarity calculation. In the filtering step, the menus having allergens were explicitly removed. Nonetheless, the suggested menus were not satisfactory in that the menus did not belong to the same category of target menu. In contrast, the recommendation based on the MIND dataset achieved the exclusion of the allergen perfectly, and most of the suggested menus were selected from the same category of target menu, i.e., side dish. This is evidence that the menu recommender system naturally considers nutrition- and composition-related factors simultaneously when using the MIND dataset. Therefore, the MIND dataset has a variety of potential uses in healthcare applications.

Table 4: Allergeny-free menu recommendation based on MIND dataset

| Embedding | Nutrition feature only | Menu2Vec + Nutrition + Ingredient |
|---|---|---|
| **Allergen ingredient** | *Perilla seed* | |
| **Target menu** | *Seasoned dried radish leaves and perilla seed* (side-dish) | |
| **Suggested menu** | Recommendation only | |
| | *Seasoned sweet potato stem and perilla seed* | *Stir-fried dried radish leaves* |
| | *Seasoned salad with perilla seeds and cucumber* | *Seasoned salad with dried radish leaves in soy paste* |
| | *Mushroom perilla seed soup* | *Bean powder dried radish leaves soup* |
| | Filtering and Recommendation | |
| | *Seasoned daikon* | (None) |
| | *Frozen and dried pollack soup* | (None) |
| | *Green onion daikon soup* | (None) |

**Task 4: Clinical studies with the MIND dataset**   The MIND dataset is being used for a clinical study of children with atopic dermatitis (AD) and food allergy (FA). Restriction of allergenic foods that may lead to fatal anaphylaxis is required for these children. The parents are burdened with having to limit the children's participation in social activities, e.g., camps and parties, and exposure to restaurants due to the potential of accidental ingestion of the allergenic food [48]. The children need to consume only foods prepared at home. Thus, precise diet planning and dietary healthcare are necessary to ensure the growth, development, and health of these children. The authors are working to develop an AI service application for this purpose. This work on creating a novel high-quality diet dataset was initiated for our clinical study of children with AD and FA (IRB number: KUGH IRB No. 2021-09-019). The available, practical dietary data were of insufficient quality to be used in the training of a machine for the AI service application.

To the best of our knowledge, this is the first clinical study to use and test the utility of an AI application for diet-level planning and healthcare for patients. In this clinical study, the quantifiable health measures include: (1) the Food Allergy Quality of Life-Parental Burden Questionnaire, (2) the Food Allergy Quality Of Life Questionnaire-Parent Form, (3) Food Allergy Independent Measure, (4) nutrient assessment using 24-hour dietary recall, (5) food frequency questionnaires, (6) growth status assessments, and (7) other life satisfaction indices. These measures are validated measures in clinical studies of patients requiring food restrictions[11][15][31]. Using these measures, our clinical study will evaluate the utility of our machine for diet-level planning and healthcare in the AI service user group and the control group. In addition, we have collected gut microbiome data from children with AD and healthy children to analyze the diets-microbiota association (IRB number: KUGH IRB No. 2020-11-025-016). The gut microbiome is rapidly becoming an important factor for precision medicine in cancer [9], metabolic diseases [41], autoimmune or inflammatory diseases, and allergic diseases [44]. Diets-microbiota data are essential for precision medicine, and machine learning contributes to this approach for medical recommendations [9]. In our

ongoing clinical study, diets are precisely recommended to patients with AD after analyzing their diets-microbiota associations.

Note that a high-quality large-scale diet dataset is the basis for such clinical studies on diet planning and dietary healthcare with ML. Previous to our MIND dataset, these could not be conducted. Our MIND dataset is dedicated to diet planning and dietary healthcare for allergic diseases and for chronic diseases such as diabetes mellitus, hypertension, obesity, celiac disease, gastrointestinal cancer, liver cirrhosis, and chronic kidney failure. Although there exist digital healthcare services for obesity, existing services do not tailor support to individual needs. For diabetes mellitus patients, the proportions of carbohydrate, protein, and fat must be considered carefully in diet planning, but compositional compliance is still important. For patients with chronic kidney diseases, diets should be composed of menus with decreased potassium, phosphorus, and calcium. The needs and requirements of diet planning and dietary healthcare for chronic diseases are all different, and design of quality diets that are acceptable to patients is always important. The MIND dataset is the first high-quality resource for this research direction, and the ORxML framework can further encourage this research. Physicians and dietitians have a nutrition- and menu-level dietary management education. Our work can contribute to expanding patient dietary management to the diet level.

**Toward precision diet for healthcare** Previous studies have also created and used representations of food-related datasets for different purposes. For example, the FlavorGraph is used to create food representations and food pairings recommendations [39], and food knowledge graphs can be used for personalized dietary recommendations and food production [37]. These studies analyzed representations at the menu, ingredient, and nutrition levels; our MIND dataset can be used to analyze representations at the diet, menu, ingredient, and nutrition levels. The addition of the diet level representation is important because of its usefulness in considering the compositional patterns of menus in diets, the identification of feasible menus, diet harmony, and the complementary nature of diets with existing food consumption habits. Our work should be connected and integrated with existing datasets with the goal of establishing a healthcare "precision diet". For example, food bioactive small molecule databases (FBSMDs) [59] provide valuable information at the levels of molecular behavior and molecular nutrition and can be used for drugs and health products. Note that precision diet is a diet-level approach for precision nutrition, a new branch of precision medicine, which aims to understand the health effects of the complex interplay among genetics, the microbiome, antibiotic and probiotic use, metabolism, food environment, and physical activity. Economic, social, and other behavioral characteristics are also included in this analysis [51, 42]. The integration of FBSMDs and our MIND dataset can be used for diet-level planning and healthcare that considers all information on nutrition, ingredients, their compounds and flavors, menus, and diets. Molecular consideration is essential for healthcare purposes as exemplified in existing studies on anti-cancer food identification [62, 55]. The information on anti-cancer foods can be integrated with our MIND dataset to design diet recommendations for cancer patients. Recent studies have identified the extent of the gut microbiome's importance to health and for precision diet development [9, 61]. We described our related clinical study in Task 4 "Clinical studies with the MIND dataset". Foods are critical to the prevention, management, and treatment of diseases. Our work will contribute by extending existing food science to diet-level planning and healthcare with machine learning to include recommending scientifically healthy and contextually attractive menus to patients resulting from consideration of the molecular level to the diet level.

**ORxML framework for the ML application to support professional tasks** Despite its dramatic success in many fields from academia to industry, ML still has critical shortcomings. In particular, a huge amount of data needs to be used when training the ML model so that the model can subsequently attain an acceptable level of performance. Given this necessity of creating high-quality datasets for ML research and development (R and D), government bodies and large companies have invested extensive financial resources for dataset construction projects involving ML. In 2021, the South Korean government allocated 260 million USD to construct high-quality datasets for transportation, healthcare, agriculture, manufacturing, and other domains in which the types of data include sensor

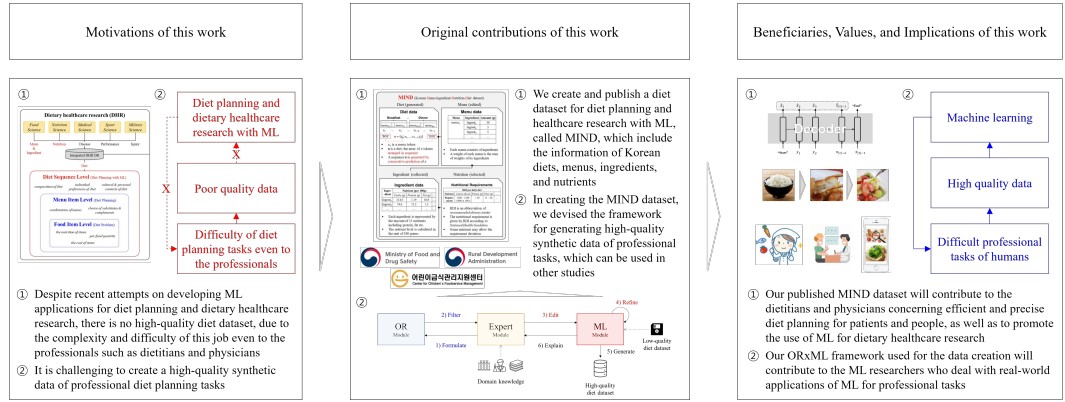

Figure 4: Summary and contributions of this work

values, text, sound, images, and video. These datasets will be released to the public via the AI Hub ($https://aihub.or.kr/$). Subsequently, R and D projects will explore AI as a means of assisting specific human tasks such as driving, diagnosis, sports, and machine control. Our diet-level data synthesis case demonstrated in this study has been motivated by the necessity of assisting the diet design tasks of dietitians and physicians.

Meanwhile, unlike the other data construction projects that require manual effort, our case is unique in its use of a systematic approach integrating the different capabilities of input from experts with OR and ML. To avoid training ML models with readily available but poor-quality records of human tasks related to diet planning and DHR, we devised the ORxML framework to generate high-quality synthetic data for such tasks (Section 3). This framework can be used for other cases requiring the generation of high-quality synthetic data. In particular, the framework can be generally applied to address the data insufficiency problem of many "professional tasks" that are difficult to perform or describe, even among experts. The quality of some professional tasks, such as diet planning and drug discovery, cannot be easily trusted because even experts, including scientists and physicians, are not skilled enough to optimally execute these tasks. This challenge is attributed to the inherent complexity of a task that involves both explicit and implicit criteria. Given this challenge, ML models trained on available but poor-quality records of the task cannot be fully utilized because the model will not perform well. In addition, the collection of data on specific professional tasks, such as the acquisition of disease diagnostic records of medical doctors and product inspection records of manufacturing engineers, is not always economically viable. Moreover, the development of a simulation environment to generate realistic data of these tasks can be nearly impossible because of task complexity. The value of learning and automating professional tasks is high, given the difficulty and importance of these tasks, and requires assistance from machines. As described in Section 3, our proposed ORxML framework addresses this problem by integrating three modules with a "human-in-the-loop": 1) the OR module to generate initial data satisfying the explicit criteria of a professional diet planning task; 2) professional dietitians who evaluate and adjust the initial data considering the implicit criteria of the task, i.e., criteria that cannot be modeled; and 3) the ML module to control the diet generation and refine the quality and completeness of the data.

**Summary of this work on the MIND dataset and the ORxML framework** In conclusion, the summary and contributions of this work are shown in Figure 4. (1) Despite recent attempts to develop ML applications for diet planning and DHR, there is no high-quality diet dataset due to the complexity and difficulty of this endeavor even for professionals. (2) The creation of high-quality synthetic data of professional diet planning tasks is challenging. (1) Thus, we have developed and published a diet dataset for diet planning and healthcare research with ML, called MIND, which includes information on Korean diets, menus, ingredients, and nutrients. (2) In creating the MIND dataset, we devised the ORxML framework for generating high-quality synthetic data regarding professional tasks, a method that can be used in other fields as well. (1) Our published MIND dataset will contribute

to the work of dietitians and physicians concerning efficient and precise diet planning for patients and individuals and will promote the use of ML for DHR. (2) Finally, our ORxML framework used for data creation will contribute to ML researchers working with professional task applications of ML.

The contribution point (1) is significant and meaningful because diet planning has been mainly considered an academic operations research (OR) problem, particularly a combinatorial optimization problem, one that mainly aims to fulfill the nutrient requirements of diets. However, this approach has been ineffective in actual practice due to its limited capability to consider the implicit patterns in diets. These implicit patterns are considered most important by professional dietitians (see Section 1 and Appendix A.4). In our paper, we originally clarified that the diet planning problem should actually be considered a sequence generation problem in order to accommodate the implicit patterns in diet sequences, i.e., to fulfill the "compositional" requirements of diet planning and to effectively address the issue of having no high-quality dataset to promote "diet planning with ML."

The history of diet planning research is quite long. Around 1945, diet planning was first defined as a linear programming problem to identify optimal quantities of food items. However, humans do not consume a specific quantity of each food unit, a combination of cooked ingredients, but rather the end-product, the food "menu," as a whole unit. Thus, around 1990, the problem was expanded to a mixed-integer programming problem to identify optimal combinations of menu items. All previous studies considered ingredient- and menu-level information, but diet-level planning should involve the compositional patterns of menus in diets. The compositional patterns are implicit depending upon the contexts and should be addressed by data-driven machine learning (ML) approaches. This work appears to be the only study on diet-level planning, and the main reason for this is the lack of a high-quality diet dataset with which to investigate data-driven ML approaches. To the best of our knowledge, existing studies on the application of ML to dietary healthcare only considered the ingredient and menu levels.

In summary, diet planning is an important problem that should be solved with ML but could not be addressed in this way due to the lack of datasets for this data-driven approach. Our dataset is the first high-quality resource to promote the research field of "diet planning with ML." Through research and development of the use cases exemplified in Appendix A.7, our goal is that the proposed dataset will be widely disseminated for diet planning and dietary healthcare with ML.

Finally, our dataset creation framework (ORxML) is unique and relevant to the dataset creation efforts for ML research. As such, our contribution point (2) is to devise the ORxML framework to systematically create a large-scale high-quality synthetic diet dataset. Given the high complexity of diet planning (see Section 1 and Appendix A.4), practical, high-quality diet data are lacking even though previous data were generated by professional dietitians. Thus, we created a novel high-quality dataset and devised an OR–Xperts–ML (ORxML) framework for diet planning and dietary healthcare with ML. This framework integrates the capabilities of OR, Expert, and ML modules. The OR module, a combinatorial optimization model, generates synthetic diets to satisfy explicit nutrient requirements. The Expert module evaluates and adjusts the initial data in terms of implicit composition requirements, the criteria that cannot be specified in the combinatorial optimization model; and the ML module automatically augments the data that ensure composition compliance with nutrition enhancement. A series of experiments demonstrate the significance of the three modules and validate the quality of our dataset (see Section 4 and Appendix A.6). Note that this framework can be used in any other contexts of creating diet data and those of difficult professional tasks as well.

## A.8  Datasheets for Datasets

The following questions and answers are from the datasheets for the datasets framework [19].

1. **Motivation**
   (a) **For what purpose was the dataset created? Was there a specific task in mind? Was there a specific gap that needed to be filled? Please**

**provide a description.**
A diet is important to all people from children to seniors and from healthy people to patients. As described in the introduction and literature review sections, the necessity of creating a benchmark dataset for DHR with ML has increased rapidly for food technology, nutrition management, clinical medicine, sports science, and military nutrition, among others. In addition, another important motivation of our work was as follows: In South Korea, most daycare centers rely on the local government's Center for Children's Food Service Management for diet planning. However, the dietitians employed in the government centers or daycare centers are burdened with designing diet plans because of the complexity of diet design. In addition, they have other duties, such as monitoring the cooking and hygiene status, as well as budget management. Our work was initiated to solve this problem and help dietitians efficiently design high-quality diets for children.

As such, under the support from the Korean government, we aimed to develop AI that could automatically generate diets for diet planning and DHR in South Korea. However, to train the AI, we needed information on the ingredients of the menu and the nutrition corresponding to those ingredients. Moreover, the sources that publish the food and ingredient data are different, leading to inconsistencies in the data. As a result, it was difficult to have sufficiently organized data for training. Thus, datasets had to be developed before developing the diet-related AI, and so we worked with dietitians from hospitals to unify the ingredient data and menu data from different sources. We believe that generating diets based on the unified data that has been validated by dietitians will be a valuable asset for AI developers working on diet-related projects, in addition to our own research. Therefore, we decided to generate and distribute this data.

(b) **Who created the dataset (e.g., which team, research group) and on behalf of which entity (e.g., company, institution, organization)?**
Authors from the Ulsan National Institute of Science and Technology created prototypes of the unified datasets using an operations research model for diet planning. Authors from the Kosin University College of Medicine validated and modified the prototype data from the nutritional and clinical perspectives through a collaboration with external experts. Then, based on the diet data refined by the experts, we trained a controllable generation machine to generate diet data. Further details on the generation process are shown in Section 3 in the main body and Appendix A.2.

(c) **Who funded the creation of the dataset? If there is an associated grant, please provide the grant name and number.**
This work was supported by the Institute of Information & communications Technology Planning & Evaluation (IITP) grant funded by the Korean government (MSIT) (No.2020-0-02135: Development of a gut microbiome-personalized diet recommendation AI system for the children with atopic diseases, No.2020-0-01336: Artificial Intelligence Graduate School Program - UNIST). This work was supported by the Bio & Medical Technology Development Program of the National Research Foundation (NRF) funded by the Ministry of Science and ICT (grant number 2019M3E5D1A02070867) and by the Ministry of Education (grant number NRF-2021R1I1A4A01049121). This work was supported by the 2021 Research Fund (1.210050) of UNIST.

2. **Composition**

(a) **What do the instances that comprise the dataset represent (e.g., documents, photos, people, countries)? Are there multiple types of instances (e.g., movies, users, and ratings; people and interactions between them; nodes and edges)?**
Our dataset includes instances of food ingredients, menus, and diet. See the Supplementary Material B and C for more detailed information.

(b) **How many instances are there in total (of each type, if appropriate)?**
Our dataset includes 3,036 ingredient instances, 3,238 menu instances, and 1,500 daily diet instances.

(c) **Does the dataset contain all possible instances or is it a sample (not necessarily random) of instances from a larger set?**
Yes. Our dataset contains all possible instances.

(d) **What data does each instance consist of? "Raw" data (e.g., unprocessed text or images) or features? In either case, please provide a description.**
The ingredient data involve the category and nutrition information for each ingredient item. The menu data involve the category, note, and ingredient information for each menu item. The diet data involve the identifier and its menu composition. See the Supplementary material B and C for further details.

(e) **Is there a label or target associated with each instance? If so, please provide a description.**
Yes. For example, menus are labeled with their category (e.g., rice, soup). See the Supplementary material B for further details.

(f) **Is any information missing from individual instances? If so, please provide a description, explaining why this information is missing (e.g., because it was unavailable). This does not include intentionally removed information, but might include, e.g., redacted text.**
Yes. From the original source of the government database, various types of nutrients have been removed due to the incompletion issue (e.g., Amount unknown for specific ingredients). We included the most prominent types of nutrients based on the literature of nutrition.

(g) **Are relationships between individual instances made explicit (e.g., users' movie ratings, social network links)? If so, please describe how these relationships are made explicit.**
Yes. For example, menus are related in diets, while ingredients are related in menus. See the Supplementary material B anc C for further details.

(h) **Are there any errors, sources of noise, or redundancies in the dataset? If so, please provide a description.**
Although we tried hard to minimize any errors, sources of noise, or redundancies in the dataset, there could be such cases. Meanwhile, two or more recipes may exist for one kind of menu. Because there is a nutritionally significant difference between the two recipes, we did not select one recipe only. We included both cases by differentiating the minor part of the menu name (e.g. From 'Pork fried rice made with oyster sauce' to 'Pork tenderloin fried rice made with oyster sauce').

(i) **Is the dataset self-contained, or does it link to or otherwise relyon external resources (e.g., websites, tweets, other datasets)? If it links to or relies on external resources.**
The MIND dataset is self-contained.

(j) **Does the dataset contain data that might be considered confidential (e.g., data that is protected by legal privilege or by doctorpatient confidentiality, data that includes the content of individuals' non-public communications)? If so, please provide a description.**
No.

(k) **Does the dataset contain data that, if viewed directly, might be offensive, insulting, threatening, or might otherwise cause anxiety? If so, please describe why.**
No.

3. **Collection Process**

(a) **How was the data associated with each instance acquired? Was the data directly observable (e.g., raw text, movie ratings), reported by subjects (e.g., survey responses), or indirectly inferred/derived from other data (e.g., part-of-speech tags, model-based guesses for age or**

**language)?**

The ingredient data had been collected and created from the Rural Development Administration of the Korean government through validated experiments. The menu data was developed by the Children's food service management center under the Ministry of Food and Drug Safety of the Korean government based on the expertise of the affiliated professional dietitians. The diet data were generated in this work. See the main body of this article for details.

(b) **What mechanisms or procedures were used to collect the data (e.g., hardware apparatus or sensor, manual human curation, software program, software API)? How were these mechanisms or procedures validated?**

The ingredient and menu data could be downloaded from the Web pages of the government organizations. As aforementioned, we created high-quality synthetic diet data based on the proposed method in this work.

(c) **Who was involved in the data collection process (e.g., students, crowdworkers, contractors) and how were they compensated (e.g., how much were crowdworkers paid)?**

The ingredient and menu data were collected by the authors. They created the diet data.

(d) **Over what timeframe was the data collected? Does this timeframe match the creation timeframe of the data associated with the instances (e.g., recent crawl of old news articles)? If not, please describe the timeframe in which the data associated with the instances was created.**

The data collection and creation jobs have been conducted from June 2020 to August 2021.

(e) **Were any ethical review processes conducted (e.g., by an institutional review board)? If so, please provide a description of these review processes, including the outcomes, as well as a link or other access point to any supporting documentation.**

The corresponding authors (one affiliated with an engineering school and the other with a medical school) confirm that our data do not need an ethical review process. The aforementioned grants do not require an ethical review process regarding our data.

(f) **Does the dataset relate to people? If not, you may skip the remainder of the questions in this section.**

No.

4. **Preprocessing/cleaning/labeling**

(a) **Was any preprocessing/cleaning/labeling of the data done (e.g., discretization or bucketing, tokenization, part-of-speech tagging, SIFT feature extraction, removal of instances, processing of missing values)? If so, please provide a description. If not, you may skip the remainder of the questions in this section.**

We conducted preprocessing and cleaning jobs for the ingredient and menu data sourced from the government organizations. See the Supplementary material B for further details.

(b) **Was the "raw" data saved in addition to the preprocessed/cleaned/labeled data (e.g., to support unanticipated future uses)? If so, please provide a link or other access point to the "raw" data.**

No.

(c) **Is the software used to preprocess/clean/label the instances available? If so, please provide a link or other access point.**

No.

5. **Uses**

(a) **Has the dataset been used for any tasks already? If so, please provide a description.**

Yes, our work has already started to create a real impact. As the synthesized

high-quality diet-level dataset MIND is intended to be used in the development of a machine for professional dietitians and pediatricians, the Center for Children's Food Service Management in South Korea have indicated that they will use our outcomes from the fall of 2021. In addition, a startup company in South Korea will use this dataset for the development of a gut microbiome-personalized diet recommendation AI system for children with atopic diseases and food allergies. See Appendix A.7 for further details.

(b) **Is there a repository that links to any or all papers or systems that use the dataset? If so, please provide a link or other access point.**
No. We published our MIND dataset in August 2021 for the first time.

(c) **What (other) tasks could the dataset be used for?**
Our data and its management tools are useful for many food-related tasks that require the verification of ingredients and nutrients. For example, data can be used to filter menus containing allergen ingredients from a diet for allergic patients. In addition, a diet generation application to support diet planning for allergic patients can be trained based on our MIND dataset. See the Introduction and Appendix A.7 for further details.

(d) **Is there anything about the composition of the dataset or the way it was collected and preprocessed/cleaned/labeled that might impact future uses? For example, is there anything that a future user might need to know to avoid uses that could result in unfair treatment of individuals or groups (e.g., stereotyping, quality of service issues) or other undesirable harms (e.g., financial harms, legal risks) If so, please provide a description. Is there anything a future user could do to mitigate these undesirable harms?**
The diets provided by MIND are healthy "reference diets" for an unspecified majority of the population. Therefore, there is no guarantee that a user will have a positive response to the provided diets for their meals. Although our diets have been evaluated and confirmed by the related government organization and the affiliated professional dietitians, there is a further need to consider health and preference issues, such as food allergies and low-salt diet preference, when serving the provided diets. We do not consider these user-specific factors because our dataset was not created for a particular target group for a customized purpose. Therefore, in order to use the diets provided in this MIND, it is recommended to inspect the ingredients and nutrition using the analysis functions provided together in the dietkit package.

(e) **Are there tasks for which the dataset should not be used? If so, please provide a description.**
We cannot imagine such tasks yet. There should be no problem, given that the original source data from the government are all certified, while the synthesized data were created by machines and confirmed by professional dietitians.

6. **Distribution**

(a) **Will the dataset be distributed to third parties outside of the entity (e.g., company, institution, organization) on behalf of which the dataset was created? If so, please provide a description.**
No. We distribute the data to the public. See the last part (license and rights) of the Supplementary material.

(b) **How will the dataset will be distributed (e.g., tarball on website, API, GitHub)? Does the dataset have a digital object identifier (DOI)?**
Our dataset and its management package have DOI: 10.5281/zenodo.5302044 Also, our dataset is distributed with the management package though GitHub: github.com/pki663/dietkit and PyPI(The Python Package Index): pypi.org/project/dietkit

(c) **When will the dataset be distributed?**
It is currently available.

(d) **Will the dataset be distributed under a copyright or other intellectual property (IP) license, and/or under applicable terms of use (ToU)?**

**If so, please describe this license and/or ToU, and provide a link or other access point to, or otherwise reproduce, any relevant licensing terms or ToU, as well as any fees associated with these restrictions.**
The MIND dataset has three sub-datasets with different license information. See the last part (license and rights) of the Supplementary material.

(e) **Have any third parties imposed IP-based or other restrictions on the data associated with the instances?**
No.

(f) **Do any export controls or other regulatory restrictions apply to the dataset or to individual instances?**
Regulatory restrictions can be applied to the data depending on the case and the type of instance. See the last part (license and rights) of the Supplementary material.

7. **Maintenance**

(a) **Who is supporting/hosting/maintaining the dataset?**
The MIND dataset is hosted by the two channels: GitHub and PyPI (The Python Package Index). See the Supplementary material E for the data maintenance plan.

(b) **How can the owner/curator/manager of the dataset be contacted (e.g., email address)?**
By email: sooo@unist.ac.kr, chlim@unist.ac.kr, or my.jung@kosin.ac.kr

(c) **Is there an erratum? If so, please provide a link or other access point.**
No.

(d) **Will the dataset be updated (e.g., to correct labeling errors, add new instances, delete instances)? If so, please describe how often, by whom, and how updates will be communicated to users (e.g., mailing list, GitHub)?**
Yes. We will update the data approximately every year as relevant data sources are updated. For example, the National Standard Food Components, which is the original source of the ingredient data, is updated every year by the Rural Development Administration of the Korean government.

(e) **If the dataset relates to people, are there applicable limits on the retention of the data associated with the instances.**
The MIND dataset does not relate to people. Therefore, there is no limit on the retention of the data associated with the instances.

(f) **Will older versions of the dataset continue to be supported/hosted/maintained? If so, please describe how. If not, please describe how its obsolescence will be communicated to users.**
Yes. Older versions are automatically preserved and provided by the GitHub host. Users can access previous versions at any time.

(g) **If others want to extend/augment/build on/contribute to the dataset, is there a mechanism for them to do so? If so, please provide a description. Will these contributions be validated/verified? If so, please describe how. If not, why not? Is there a process for communicating/distributing these contributions to other users? If so, please provide a description.**
First, see the last part (license and rights) of the Supplementary material for the potential extension of our dataset. Second, we will accommodate the suggestions and contributions for our dataset from the users through GitHub, and all amendments will be explicitly recorded as commit commands. Users can upload or connect their data to our dataset through GitHub. Meanwhile, we would like to maintain and control the structure of our dataset (although we will accommodate suggestions), such that the data and contents are organized and managed consistently and coherently.

## B  MIND dataset development process

### B.1  Original sources of menu and ingredient data

The original data sources of the MIND dataset include the data from Korean food-related institutions. Ingredient data were extracted from the 9th revision of the National Standard Food Components provided by the Rural Development Administration of the Korean government. The data consist of the name of an ingredient, the food category to which the ingredient belongs, and the weight of nutrients per 100 g of that ingredient. The database consists of 3,036 ingredients that are classified into 20 categories.

Menu data were collected from the Center for Children's Food Service Management under the Ministry of Food and Drug Safety of the Korean government. The total number of menus collected was 3,238. The raw data consist of the name of the menu and the quantity of ingredients used in each single-serve item on the menu. The diet data were generated based on the menu data. During data generation, authors from the Ulsan National Institute of Science and Technology created prototypes of the unified datasets using an operations research model for diet planning. Authors from the Kosin University College of Medicine nutritionally validated and modified the prototype data through a collaboration with external experts. Then, based on the diet data refined by the experts, we trained a controllable generation machine to generate diet data. Further details regarding the generation process are shown in Section 3 in the main article.

### B.2  Preprocessing of menu and ingredient data

The original ingredient data contained a large amount of content, but some content was removed from the final data for ease of deployment and utilization. In particular, for nutritional information, the raw data included information on a total of 129 different nutrients. However, owing to problems such as unknown or unclear contents for some ingredients, only 14 of the nutrients deemed the most important by the professional dietitians were included in the final data.

The final menu data were obtained through some processing of the raw data. The names of the ingredients in the original data were unified with the ingredients data mentioned in the previous paragraph. A total of 527 kinds of food ingredients were delineated in the menu data. The categories of menus were reclassified according to specific policies. First, the menus were classified as rice, soup, side dishes, kimchi, and snacks according to the characteristics of a typical Korean meal. Second, rice dishes were subdivided as: i) served as a staple menu and consumed with other side dishes and ii) served both as a main meal and side dish (menu "combo meal"). Side dishes were subdivided according to the recipe as: tempura, braised, grilled, steamed, stir-fried, and pancake. Snacks were subdivided into fruits, salads, milk, drinks, soup, nuts, grains, cereal, and combo meal. In addition, substantially duplicated menus were removed when there were only minor differences in parts of their names, such as "stir-fried kimchi and pork" and "stir-fried pork and kimchi."

### B.3  Diet planning and diet data generation

Professional dietitians conducted the diet planning task under policies based on the references in nutrition literature. As described in the main article of this work (Section 3), the dietitians edited the initial diets (combinations of menus) generated from a combinatorial optimization model for diet planning[24][58]. The applicable policies were:

- Determine the main dish based on the energy distribution across meals (calorie distribution: 25

- Avoid more than one "combo meal" per daily diet.

- If possible, combo meals should be served for lunch when there may be more activities. If used for breakfast, relatively lighter combo meals, e.g., riceballs, fried rice, and gimbap should be served. Items such as greasy fries should be avoided at breakfast.

- If possible, use protein-supplying foods.

- Avoid overlaps in ingredients across meals.
- Avoid overlaps between ingredients used in breakfast, lunch, and dinner so that various types of protein-supplying foods (meat/fish/eggs/beans) can be served.
- Select the ingredients and type of soup so that these complement the main dish and main side dish (e.g., curry with rice – clear soybean soup, boiled rice – spicy beef soup, black bean paste with rice – soybean soup with green onions).
- Use ingredients and recipes that may not be present in the main dish, main side dish, or soup.
- As protein-supplying foods are primarily used as the main side dish, make use of vegetables for the supplementary side dish.
- Breakfast, lunch, and dinner should include a kimchi menu. This is an important feature of the Korean diet.
- Select the type of kimchi so that there is no overlap of ingredients (e.g., if radish soup is served, exclude kimchi that contains radish).
- Include dairy products in one or more snacks.
- If the regular meals did not contain sufficient fruits or vegetables, make use of fruits or vegetables in the snacks so that the snacks complement the meals.
- For easier consumption by infants, snacks should take the form of liquid drink + solid snack (e.g., cheese – steamed potato → may be hard to eat due to the absence of a drink).
- The diet should consist of various shapes (e.g., rice with peas – ball-shaped fishcake soup – braised meatballs – cucumber salad – diced radish kimchi → round overall, sliced kelp soup – stir-fried potato strips – dried squid and dried radish salad → long and thin overall).
- The diet should consist of various textures (e.g., spicy stir-fried squid with rice – spicy beef soup with bean sprouts – lotus roots seasoned with mayonnaise – diced radish kimchi → hard and tough overall).
- Consider the colors of the foods so that the diet consists of various colors (e.g., braised black beans – lotus roots and black sesame salad – young summer radish kimchi → black overall, soft bean curd with seasoning – mung bean sprout salad – water kimchi → white overall).
- The diet should consist of various flavors (e.g., soybean paste stew – steamed spareribs – braised potatoes and fishcakes – young summer radish seasoned with soybean paste – cabbage kimchi → salty overall with the flavor of soy sauce).

The following members of professional dietitians conducted the evaluation and consultation concerning the diet dataset. Considerations used in the experts' evaluation and consultation are:

- Does the main dish suit its corresponding meal (breakfast, lunch, dinner)?
- Are there fewer than two combo meals?
- Is the serving of the main dish appropriate?
- Does the main dish allow the daily intake of nutrients (specifically, energy)?
- Are various protein-supplying foods used?
- Is there no overlap in protein-supplying foods?
- Does the recipe used suit the corresponding meal?
- Is there no overlap in recipes between meals?
- Does the main side dish allow for the daily intake of nutrients (specifically, protein)?
- Is there no overlap between the ingredients of the soup and those of the main dish and main side dish?
- If the main dish and main side dish are heavy in spices, is a soup with relatively fewer spices (e.g., clear soup, soybean soup) used?

- Is the soup balanced with the main side dish?
- Does the soup allow for the daily intake of nutrients (specifically, energy, protein, vitamins, and minerals)?
- Are the ingredients of the supplementary side dish balanced with the main side dish and soup?
- Is there no overlap in ingredients or recipes between meals?
- Does the supplementary side dish use lighter ingredients than the main side dish?
- Does the supplementary side dish allow for the daily intake of nutrients (specifically, vitamins and minerals)?
- Is there no overlap between the ingredients of kimchi and those of the other components?
- Are water kimchi and general kimchi each used an appropriate number of times?
- Is the kimchi appropriate for the daily intake of nutrients (specifically, avoiding an excessive intake of sodium)?
- Does each meal contain various colors of foods?
- Are various shapes of foods present?
- Are various flavors present?
- Are various textures present?
- Is milk served at least once a day?
- Are drinks appropriately used in snacks?
- Is there no snack that interferes with the regular meals?
- Are morning and afternoon snacks well-planned? (Because the interval between breakfast and lunch is shorter than the interval between lunch and dinner, high-calorie foods should be avoided as morning snacks)
- Do the snacks allow for the daily intake of nutrients (specifically, vitamins and minerals)?

The information of experts who participated in the development and evaluation of the diet dataset is shown in Table 5. For the last item of the checklist in the main article, the compensation to the participants was estimated based on their contracts. The experts A and B were recruited for this work under full-time researcher contracts; thus, a monthly wage was provided to each expert depending upon the contract detail. The experts C to G were recruited for this work under temporary consultation contracts; thus, an honorarium was provided to each expert depending upon the participation time.

Table 5: Qualifications of the participants of the diet verification process

| Expert ID | Educational background | Work experience |
|---|---|---|
| A | Bachelor's degree | Health center nutrition management (7 years) |
| | | Food/Diet DB Establishment for AI (2 years) |
| B | Bachelor's degree | Cooking instructor for childern's meal (1 year) |
| | | Food/Diet DB Establishment for AI (1 year) |
| C | Master's degree | Foodservice in a corporation (7 years) |
| | | Gov (4 years) |
| D | Master's degree | Clinical dietitian |
| | | Gov (6 years) |
| E | Bachelor's degree | Gov (6 years) |
| F | Master's degree | Gov (6 years) |
| G | Master's degree | Gov (4 years) |

**Gov** in the work experience table denotes Government organization of food service management

## B.4 Composition of the MIND dataset

MIND dataset consists of four elements: Diets, Menus, Ingredients, and Nutrition. Diets, Menus, and Ingredients are the data with a hierarchical structure, and Nutrition is a characteristic that can be commonly included in all of the other elements. Figure 5 shows their relationships. For example, Menus have an ingredient list as their characteristic, and diets likewise have a Menu list. Nutrition of the higher-level elements is determined through the sum of the nutrition of the lower-level elements.

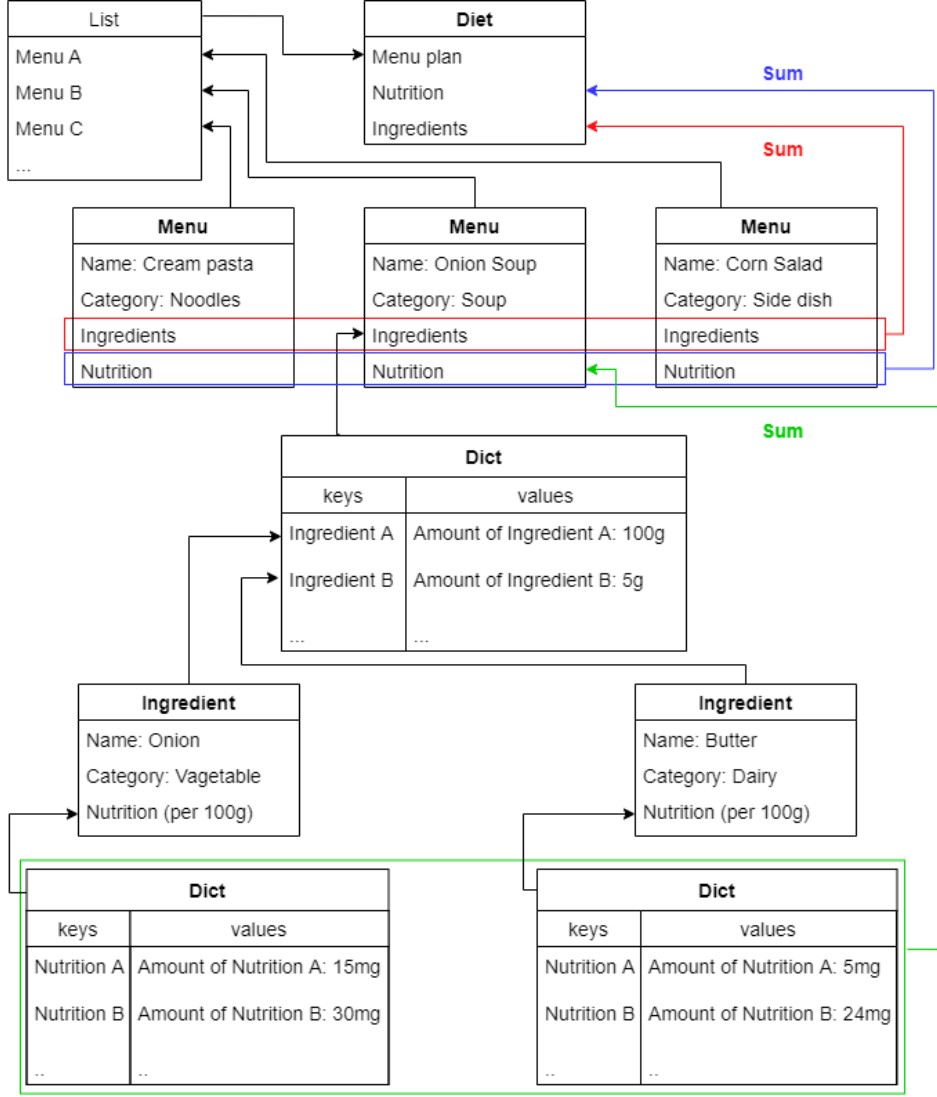

Figure 5: Relationship of elements of dataset

### B.4.1 Ingredient data

The name of the food is in the form of "(comprehensive name), (partial or serving form), (cooking form)," e.g., "pork meat, shoulder rib, raw." However, this may be omitted if there was no corresponding item for some ingredients such as "Worcestershire sauce." There were 20 categories of food ingredients. Table 6 shows the list of categories and the number of ingredients included in each category. Data include the content of 14 nutrients in the food ingredients. A list of nutrients and their representative values are shown in Table 7.

Table 6: Composition of the ingredient data

| Category | Number of ingredients | |
|---|---|---|
| Grains and grain products | 291 | (9.58%) |
| Potatoes and starch | 69 | (2.27%) |
| Sugars | 50 | (1.65%) |
| Beans | 64 | (2.11%) |
| Nuts and fruits | 87 | (2.87%) |
| Vegetables | 622 | (20.49%) |
| Mushrooms | 75 | (2.47%) |
| Fruits | 241 | (7.94%) |
| Meats | 328 | (10.80%) |
| Eggs | 22 | (0.72%) |
| Fish and shellfish and other seafood | 674 | (22.20%) |
| Seaweeds | 57 | (1.88%) |
| Milk and dairies | 55 | (1.81%) |
| Fats and oils | 33 | (1.09%) |
| Teas | 56 | (1.84%) |
| Other drinks | 25 | (0.82%) |
| Alcoholic drinks | 25 | (0.82%) |
| Condiments | 104 | (3.43%) |
| Cooked processed food | 128 | (4.22%) |
| Other | 30 | (0.99%) |

Table 7: Representative value of ingredients' nutrition

| | Energy (kcal) | Protein (g) | Fat (g) | Carbohydrate (g) | Total Dietary (g) |
|---|---|---|---|---|---|
| mean | 169.77 | 10.86 | 6.06 | 19.18 | 2.12 |
| Q1 | 51.0 | 1.85 | 0.2 | 0.6 | 0.0 |
| median | 118.0 | 7.15 | 1.1 | 7.3 | 0.0 |
| Q3 | 261.0 | 18.1 | 5.24 | 27.02 | 2.0 |

| | Calcium (mg) | Iron (mg) | Sodium (mg) | Vitamin A ($\mu$g RAE) | Vitamin B1 (mg) |
|---|---|---|---|---|---|
| mean | 415.77 | 115.44 | 0.15 | 0.24 | 9.53 |
| Q1 | 1.0 | 0.0 | 0.02 | 0.04 | 0.0 |
| median | 14.0 | 3.0 | 0.07 | 0.11 | 0.37 |
| Q3 | 128.0 | 25.6 | 0.16 | 0.23 | 5.3 |

| | Vitamin B2 (mg) | Vitamin C (mg) | Linoleic Acid (mg) | $\alpha$-Linolenic Acid (mg) |
|---|---|---|---|---|
| mean | 624.29 | 135.88 | 110.05 | 3.6 |
| Q1 | 0.0 | 0.0 | 10.0 | 0.5 |
| median | 0.0 | 0.0 | 28.0 | 1.2 |
| Q3 | 75.44 | 19.95 | 82.0 | 2.98 |

### B.4.2 Menu data

There are 22 categories in the menu. The number of menus included in each category is shown in Table 8. The note field states additional information provided by the menu, including whether the menu is Korean or Western and whether the menu is a main dish or side dish. There were 2,635 (81.63%) menus that were noted as Korean meals and 549 (16.95%) menus that were noted as Western meals. The number of main dishes and side dishes were 694 (21.50%) and 638 (19.76%), respectively.

Table 8: Composition of the menu data

| Category | Number of menus | |
|---|---|---|
| Nuts (snack) | 4 | (0.12%) |
| Grains (snack) | 98 | (3.03%) |
| Fruits (snack) | 47 | (1.45%) |
| Grilled | 116 | (3.58%) |
| Soup | 583 | (18.00%) |
| Others | 1 | (0.03%) |
| Kimchi | 21 | (0.65%) |
| Noodles | 60 | (1.85%) |
| Rice | 78 | (2.41%) |
| Stir-fried | 473 | (14.60%) |
| Salad | 466 | (14.39%) |
| Salad (snack) | 23 | (0.71%) |
| Soup (snack) | 226 | (6.98%) |
| Cereal (snack) | 4 | (0.12%) |
| Milk (snack) | 17 | (0.52%) |
| Drink (snack) | 32 | (0.99%) |
| Combo meal (snack) | 134 | (4.14%) |
| Combo meal | 226 | (6.98%) |
| Pancake | 123 | (3.80%) |
| Braised | 267 | (8.24%) |
| Steamed | 135 | (4.17%) |
| Tempura | 105 | (3.24%) |

### B.4.3 Diet data

Currently, we provide three kinds of diet data: "OR-generated diets," "Expert-generated diets," and "ML-generated diets." Each of the Expert and ML diet datasets comprises 500 different diets. The OR diet data consists of 140 diets. Each diet dataset consists of 19 menus: five breakfast menus, two morning snacks, five lunch menus, two afternoon snacks, and five dinner menus. Some diets have a slightly smaller number of menus. In this case, we included a dummy menu called "empty" to fit the data form. This dummy menu item was included in the menu data in the section B.4.2. Note that the OR-generated diets completely satisfy the criteria in Table 9. These criteria were based on the dietary reference intakes presented by the Ministry of Health and Welfare of the Korean government. The Expert- and ML-generated diets, in contrast, do not adhere to the nutritional criteria but reflect compositional requirements such as the distribution of ingredients preferred by dietitians based on their experience. The representative nutritional values of each diet are shown in Tables 10, 11 and 12.

Table 9: Nutrition criterion of the OR diet data

| Nutrient | Required Condition | Unit |
|---|---|---|
| Energy | $1260 \sim 1540$ | kcal |
| Protein | $20 \sim$ | g |
| Percentage of carbohydrates in total energy | $55 \sim 65$ | % |
| Percentage of protein in total energy | $7 \sim 20$ | % |
| Percentage of fat in total energy | $15 \sim 30$ | % |
| Total Dietary Fiber | $11 \sim 20$ | g |
| Calcium | $500 \sim 2500$ | mg |
| Iron | $5 \sim 40$ | mg |
| Sodium | $\sim 1600$ | mg |
| Vitamin A | $230 \sim 750$ | $\mu$g RAE |
| Vitamin B1 (Thiamine) | $0.4 \sim$ | mg |
| Vitamin B2 (Rivoflavin) | $0.5 \sim$ | mg |
| Vitamin C | $35 \sim 510$ | mg |
| Linoleic Acid | $4.6 \sim 9.1$ | g |
| Alpha Linoleic Acid | $0.60 \sim 1.17$ | g |

Table 10: Representative nutritional values of the expert diet data

| | Energy (kcal) | Protein (g) | Fat (g) | Carbohydrate (g) | Total Dietary (g) |
|---|---|---|---|---|---|
| mean | 1314.38 | 54.72 | 31.55 | 200.56 | 12.92 |
| Q1 | 1230.44 | 50.49 | 26.46 | 186.18 | 11.05 |
| median | 1316.21 | 54.55 | 31.00 | 200.07 | 12.75 |
| Q3 | 1408.34 | 58.88 | 36.07 | 214.94 | 14.55 |

| | Calcium (mg) | Iron (mg) | Sodium (mg) | Vitamin A ($\mu$g RAE) | Vitamin B1 (mg) |
|---|---|---|---|---|---|
| mean | 538.81 | 9.47 | 1663.67 | 349.72 | 0.96 |
| Q1 | 452.15 | 8.19 | 1469.67 | 272.39 | 0.84 |
| median | 515.56 | 9.15 | 1639.33 | 329.84 | 0.94 |
| Q3 | 617.56 | 10.34 | 1839.86 | 397.43 | 1.05 |

| | Vitamin B2 (mg) | Vitamin C (mg) | Linoleic Acid (mg) | $\alpha$-Linolenic Acid (mg) |
|---|---|---|---|---|
| mean | 1.19 | 61.28 | 6965.51 | 937.96 |
| Q1 | 1.03 | 40.46 | 5668.14 | 647.26 |
| median | 1.17 | 54.18 | 6932.71 | 834.31 |
| Q3 | 1.30 | 78.52 | 8215.07 | 1105.07 |

Table 11: Representative nutritional values of the OR diet data

|        | Energy (kcal) | Protein (g) | Fat (g) | Carbohydrate (g) | Total Dietary (g) |
|--------|--------|--------|--------|--------|--------|
| mean   | 1356.78 | 53.25 | 32.08 | 212.65 | 17.38 |
| Q1     | 1294.86 | 48.26 | 28.31 | 204.01 | 16.42 |
| median | 1349.06 | 53.71 | 32.07 | 211.24 | 17.22 |
| Q3     | 1423.69 | 56.90 | 35.64 | 220.81 | 18.09 |

|        | Calcium (mg) | Iron (mg) | Sodium (mg) | Vitamin A ($\mu$g RAE) | Vitamin B1 (mg) |
|--------|--------|--------|--------|--------|--------|
| mean   | 608.92 | 10.72 | 1508.72 | 338.96 | 0.99 |
| Q1     | 537.82 | 9.15 | 1454.34 | 263.87 | 0.86 |
| median | 579.83 | 10.62 | 1528.16 | 300.78 | 0.97 |
| Q3     | 652.96 | 11.72 | 1567.55 | 377.42 | 1.09 |

|        | Vitamin B2 (mg) | Vitamin C (mg) | Linoleic Acid (mg) | $\alpha$-Linolenic Acid (mg) |
|--------|--------|--------|--------|--------|
| mean   | 1.28 | 54.01 | 6521.25 | 764.49 |
| Q1     | 1.14 | 40.04 | 5702.19 | 638.27 |
| median | 1.28 | 45.90 | 6541.34 | 724.45 |
| Q3     | 1.39 | 60.70 | 7249.95 | 864.54 |

Table 12: Representative nutritional values of the ML diet data

|        | Energy (kcal) | Protein (g) | Fat (g) | Carbohydrate (g) | Total Dietary (g) |
|--------|--------|--------|--------|--------|--------|
| mean   | 1350.47 | 56.35 | 33.13 | 204.63 | 13.75 |
| Q1     | 1269.75 | 51.35 | 28.01 | 190.31 | 11.94 |
| median | 1351.62 | 56.18 | 32.62 | 204.29 | 13.57 |
| Q3     | 1436.06 | 60.80 | 38.21 | 218.98 | 15.50 |

|        | Calcium (mg) | Iron (mg) | Sodium (mg) | Vitamin A ($\mu$g RAE) | Vitamin B1 (mg) |
|--------|--------|--------|--------|--------|--------|
| mean   | 603.37 | 9.55 | 1621.37 | 387.18 | 0.98 |
| Q1     | 514.63 | 8.35 | 1438.77 | 304.20 | 0.86 |
| median | 590.96 | 9.16 | 1591.49 | 367.85 | 0.96 |
| Q3     | 692.85 | 10.49 | 1773.10 | 448.95 | 1.07 |

|        | Vitamin B2 (mg) | Vitamin C (mg) | Linoleic Acid (mg) | $\alpha$-Linolenic Acid (mg) |
|--------|--------|--------|--------|--------|
| mean   | 1.32 | 68.04 | 6904.28 | 914.62 |
| Q1     | 1.15 | 45.31 | 5777.78 | 667.72 |
| median | 1.29 | 62.26 | 6844.38 | 799.76 |
| Q3     | 1.46 | 83.47 | 7903.52 | 1029.58 |

## C  Dietkit package information

Dietkit is the Python package that provides tools for managing and analyzing diets and their components. The MIND dataset is distributed with dietkit as sample data. Our package includes three classes representing the diet and its elements: Diets, Menus, and Ingredients. The functions of dietkit are to handle data in the form of these classes. We provide data evaluation and visualization functions in dietkit so that the users can import, analyze, and review the dataset and its components in an automatic manner. The loader functions import sample or user data into our classes. In the import process, data, e.g., nutrient information is automatically calculated. Then, the imported data are analyzed by other functions. Evaluator functions assess the imported data using nutritional or ingredient criteria. Visualizer functions support the understanding of the characteristics of data by generating graphs and visual materials. The MIND dataset in the dietkit package will be maintained and updated continuously (see section E.2). The users can evaluate and visualize various characteristics of the latest version of the dataset for customized use. Finally, the users of dietkit can manually edit the dataset, e.g., by adding new diet data.

MIND consists of several datasets within the dietkit: "Ingredient-Nutrition data," "Menu-Ingredient Data," and "Diet-Menu data". Dietkit can generate additional data by processing these three datasets. For example, the Menu-Ingredient data generate Menu-Nutrition data along with Ingredient-Nutrition data. This generation process can be performed automatically when the data are loaded without input or commands from the user. Figure 6 shows this data flow in the dietkit. Dietkit essentially loads a sample dataset, but, if the user has data tailored to a given format in Section C.2.1, data also can be loaded.

### C.1  Embedded classes

The package includes three classes representing the diet: Ingredient, Menu, and Diet. The Ingredient class represents the grocery ingredients. Each ingredient instance includes nutritional information. The Menu class corresponds to the dishes served. Each menu instance comprises its own ingredients. The Diet class represents the diet plan. Here, diet does not refer to a single diet but a combination of several diets. Each diet instance consists of a pair of identifiers and the diet content. Additionally, the Diet provides the Criteria, a class for nutritional evaluation. The diet-related datasets represented by these classes are subsequently evaluated and visualized using the aforementioned functions.

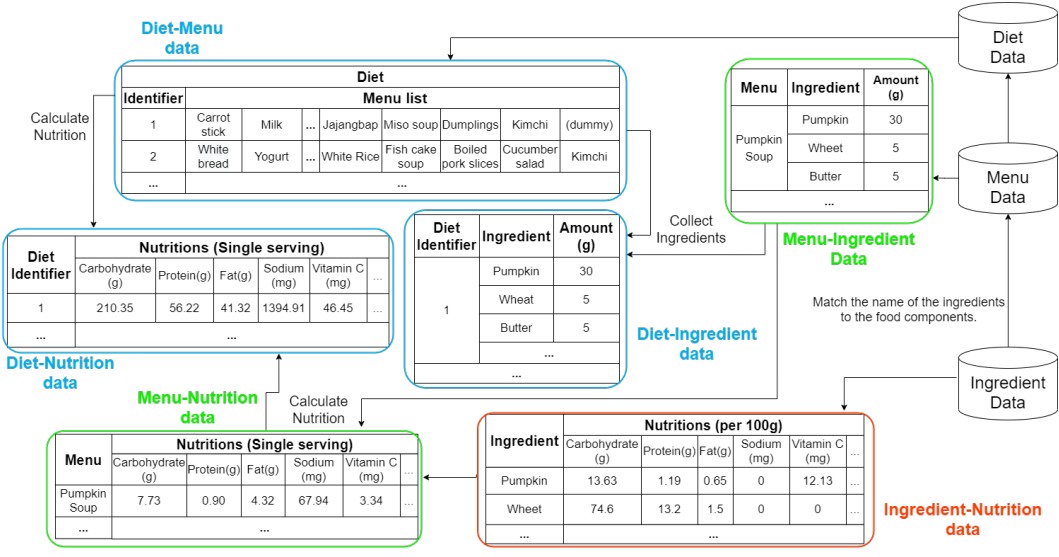

Figure 6: Schema of the MIND dataset and dietkit

### C.1.1 Ingredient

The Ingredient class represents the grocery ingredient(s) that comprise(s) the dishes. This class consists of name, category, and nutrition. Each Ingredient object includes name, category, and nutrition items. Nutrition refers to the information about the nutrients contained in 100 g of the ingredient, and this value is used to calculate the nutrients in the Menu or Diet class.

### C.1.2 Menu

The Menu class represents the dishes that are considered in the diet. This class consists of name, category, ingredients, and nutrition. The Ingredients object corresponds to the ingredients included in a single serving of a menu. Nutrition is automatically calculated based on the ingredients without any user input..

### C.1.3 Diet

The Diet class represents the plan of diets. This class consists of plan, ingredients, and nutrition. Plan corresponds to the pairs of identifiers and the list of menus in the diet. Ingredients and nutrition are automatically calculated from the menu list of the diet plan. The Diet class comprises the "includes" methods for exporting information from the diet.

- menu_category
  The user can export the table with the category data representing the diet's menu. The index is the identifier of the diet.

- menu_note
  The user can export the table with the note data representing the diet's menu. The index is the identifier of the diet.

### C.1.4 Criterion

The Criterion class represents the nutritional criterion. This class is used as an input to the evaluation method for assessing the nutrition of the menu or diet. Criteria includes three pieces of information. The first concerns the nutrients to which this criterion is related. The second is the baseline, and the last is the passing condition, i.e., information on whether the nutrient values are greater or less than the baseline.

## C.2 Embedded functions

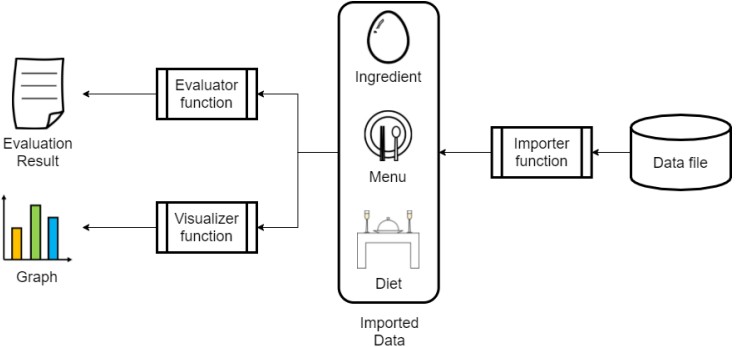

Figure 7: An overview of the dietkit functions

Dietkit includes functions that can be divided into three main types: Loader, Evaluator, and Visualizer. The Loader function loads the ingredient-, menu-, or diet-level data. If a specific file path is not passed, then our sample data are automatically loaded. The Evaluator function evaluates the menu or the diet with respect to the ingredients and nutrition based on the user's criteria. The Visualizer function graphically visualizes the diet's information or the evaluation results. Figure 7 shows an overview of the functions.

### C.2.1 Loader

The Loader function imports data from a file. If a specific file path is not passed, then our sample data is imported.

Table 13: Ingredient data file form

| Name | Category | Energy | Protein | Fat | ... |
|------|----------|--------|---------|-----|-----|
| Barley | Grains | 352 | 10 | 1 | ... |
| Oatmeal | Grains | 382 | 13.2 | 8.2 | ... |
| Broccoli | Vegetables | 32 | 3.08 | 0.2 | ... |
| | | ... | | | |

- load_ingredient
  Load the ingredient data to the input file path. The data file should be saved in *.csv format. The Ingredients data file should follow the format of Table 13. The first column contains the names of the ingredients. The second column represents the category of each ingredient such as meat, grain, or fruit. The third column corresponds to the amount of the nutrients contained in 100 g of the ingredient.

Table 14: Menu data file form

| Name | Ingredient | Weight | Category | Note |
|------|-----------|--------|----------|------|
| Fruit salad | Apple | 15 | Salad | Western |
| | Pear | 10 | | |
| | Orange | 25 | | |
| | Mayonnaise | 5 | | |
| Bulgogi | Beef | 30 | Stir-fried | Korean |
| | Soy sauce | 2 | | |
| | Sesame oil | 0.2 | | |
| | Paprika | 5 | | |
| | Garlic | 0.2 | | |
| | Pepper | 0.1 | | |
| | Onion | 15 | | |
| | Carrot | 5 | | |
| | | ... | | |

- load_menu
  Load the ingredient data to the input file path. The data file should be saved in *.csv format. The menu data file should follow the format of Table 14. The first column represents the names of menus. The second column shows the ingredients of each menu. The third column represents the quantity of the ingredients listed in the second column in grams $S(g)$. The column names are the names of the nutrients. The fourth column shows the category to which the menu belongs, and the fifth column shows notes about the menu. The contents of the fourth and fifth columns can be left blank, but the column must exist. Note that, in each menu from the second ingredient on, we leave the "Name" in the row blank.

Table 15: Diet data file form

| Identifier | Morning_1 | Morning_2 | Lunch_1 | Lunch_2 | Lunch_3 | ... |
|-----------|-----------|-----------|---------|---------|---------|-----|
| 0 | Carrot stick | Milk | Jajangbap | Miso soup | Dumplings | ... |
| 1 | White bread | Yogurt | White rice | Fish cake soup | Kimchi | ... |
| | | | ... | | | |

- load_diet
  Load the ingredient data to the input file path. The data file should be saved in *.csv format. The diet data file should follow the format of Table 15. The first column is the identifier for each diet. This can be a number, a string such as a date, or a mixture of several types; but the contents of the column must not be duplicated. The second column lists the menus that comprise each meal, one in each column. However, all diets must consist of the same number of menu items. If the length of each diet is not the same, a dummy menu item such as "empty" can be used and is provided by the sample menu data.

### C.2.2   Evaluator

Table 16: Part of sample criteria

| On | Condition | Value |
|---|---|---|
| Energy | <= | 1540 |
| Protein | >= | 20 |
| Total Dietary | >= | 11 |
| Total Dietary | <= | 20 |
| Vitamin B1 (Thiamine) | > | 0.4 |
| Sodium | < | 1600 |

- load_sample_criteria
  Load the sample criteria from the file. The sample criteria correspond to the standard nutrition criteria of South Korea. The details of the sample criteria are described in Table 16. The first column should be "On," and this indicates the nutrient to be compared. The second column is "Condition." This corresponds to the condition to pass the criterion and must be one of four conditions: ">"(pass if the amount of nutrition is greater than the baseline value), "<"(pass if the amount of nutrition is less than the baseline value), ">="(pass if the amount of nutrition is greater than or equal to the baseline value), and "<="(pass if the amount of nutrition is less than or equal to the baseline value). The last column is the baseline value.

- menu_test_nutrition
  Test the menu's nutrition using the input criterion object. If the menu passes the criterion, then the criterion is returned as "True."

- diet_test_nutrition
  Test the diet's nutrition using the input criterion object. This function applies the criterion to each menu list of the diet plan. The function returns the pairs of identifiers of the diet and the result (i.e., "Pass") from the diet.

- menu_test_ingredient
  Test if the menu includes the input ingredient. If the menu is included, then the function returns "True."

- diet_test_ingredient
  Test if the diet includes the input ingredient. This function applies the criterion to each menu list in the diet plan. The function returns pairs of identifiers of the diet and the result (i.e., "Pass") from the diet.

### C.2.3   Visualizer

- bar_menu_test_nutrition
  Plot a bar graph to illustrate the proportion of the diet plan that has passed the input nutrition criteria. Figure 8 shows an example of the execution result. The x-axis corresponds to the applied criterion, and the y-axis is the ratio of diet plans that passed the criterion. The ratio of the blue section of each bar refers to the proportion of diets that passed the criterion.

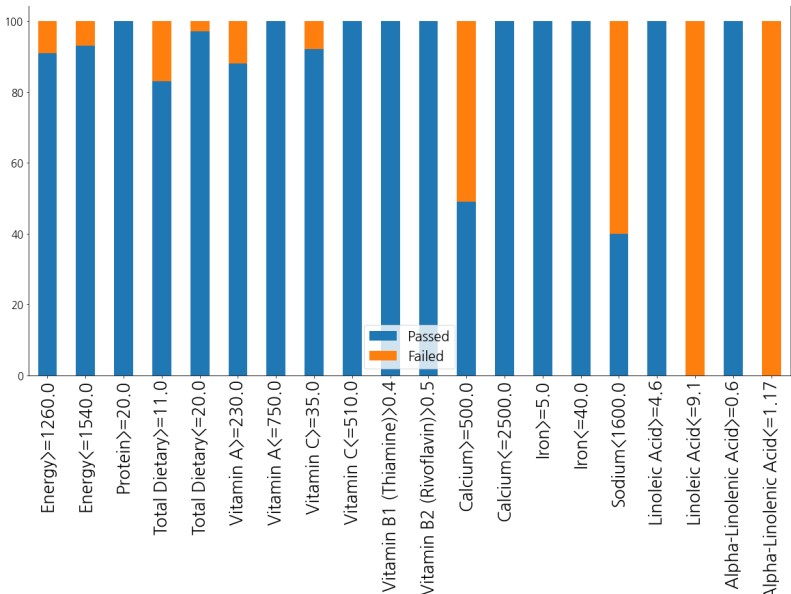

Figure 8: Execution result of bar_menu_test_nutrition

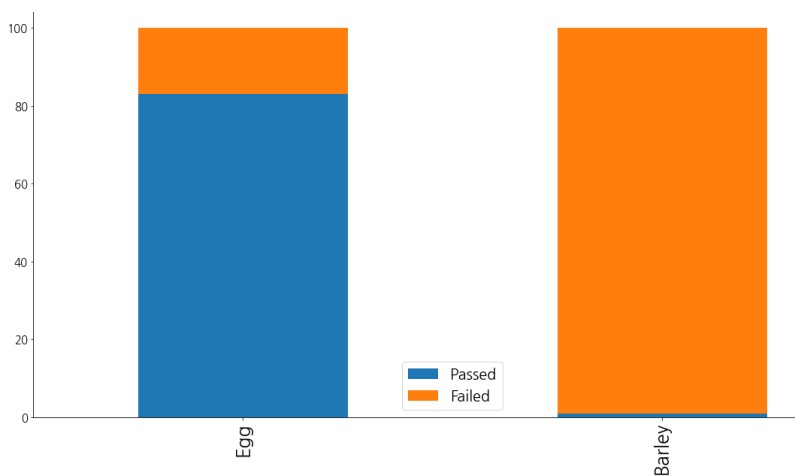

Figure 9: Execution result of bar_menu_test_ingredient

- bar_menu_test_ingredient
  Plot a bar graph to illustrate the proportion of the diet plan that includes the input ingredient. Figure 9 shows an example of the execution result. The x-axis corresponds to the reference ingredients, and the y-axis corresponds to the ratio of the diet plan that passed the criterion. The ratio of blue sections of each bar refers to the proportion of diets that passed the criterion.

- heatmap_menu_test_nutrition
  Plot a heatmap to illustrate which diet has passed the input nutrition criteria. Figure 10 shows an example of an execution result. The x-axis is the applied criteria, and the y-axis is the identifier of the diet (key in the diet plan dictionary). A bright cell indicates that the diet passed the criteria, and a dark cell indicates the diet failed to pass the criteria.

- heatmap_menu_test_ingredient
  Plot a heatmap to illustrate which diet includes the input ingredients. Figure

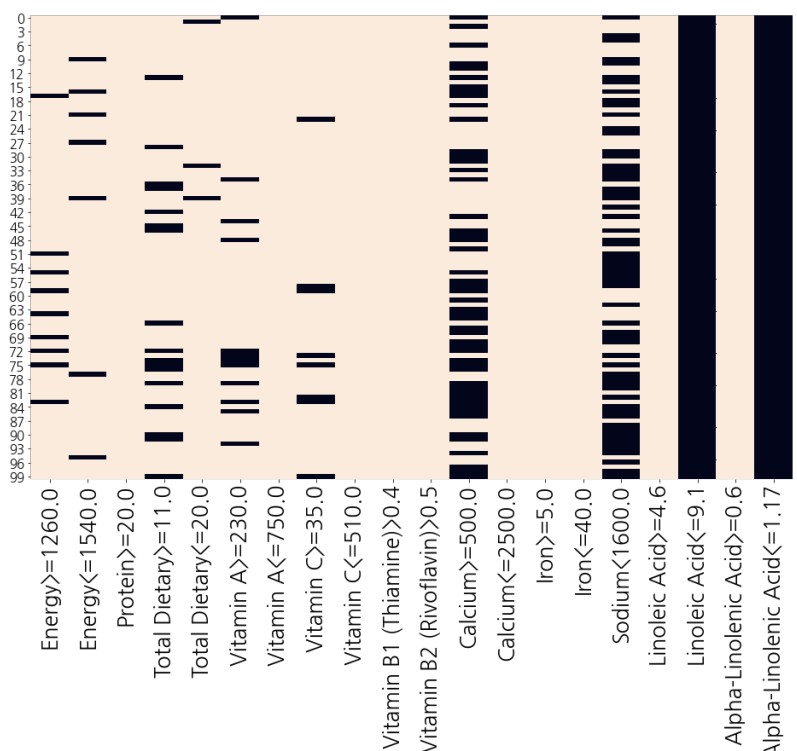

Figure 10: Execution result of heatmap_menu_test_nutrition

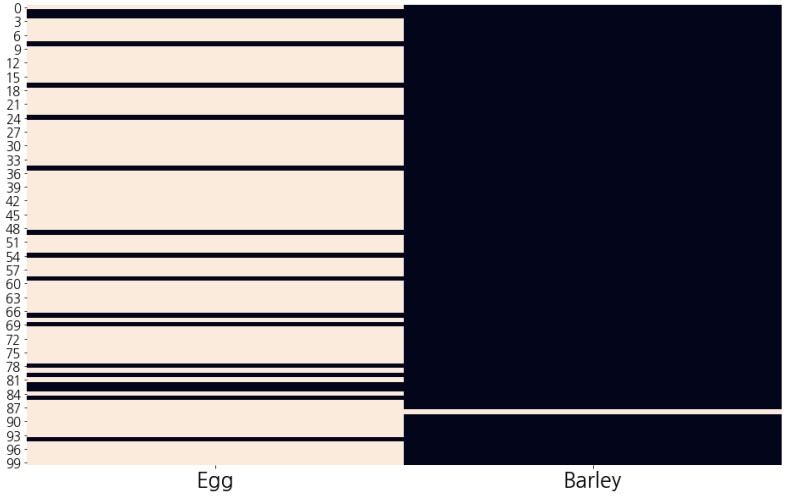

Figure 11: Execution result of heatmap_menu_test_ingredient

11 shows an example of the execution result. The x-axis corresponds to the test ingredient, and the y-axis corresponds to the identifier of the diet, the key in the diet plan dictionary. A bright cell indicates that the diet includes the ingredient, and a dark cell indicates that the diet does not include the ingredient.

- diet_ingredient_freq
  Plot a bar graph to illustrate how many times each ingredient has been used in the entire diet. Figure 12 shows an example of an execution result.

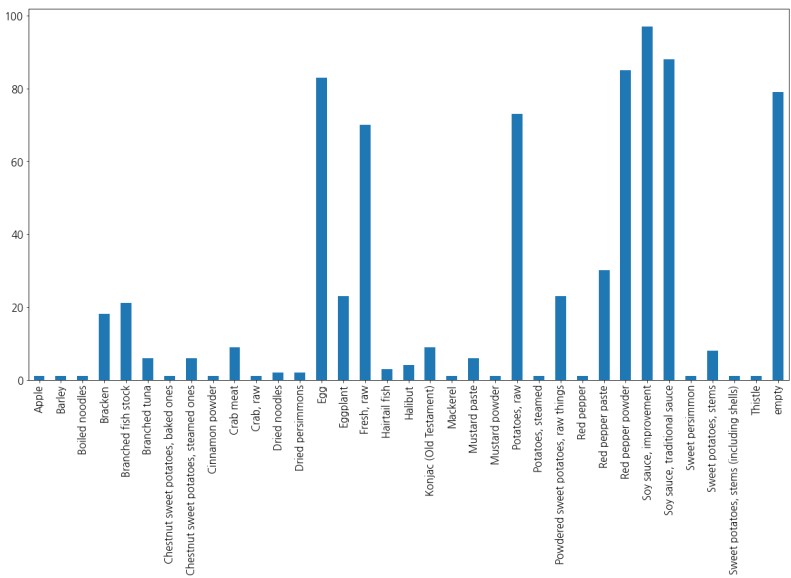

Figure 12: Execution result of diet_ingredient_freq

# D   Use of the dietkit package with MIND

## D.1   Example use cases

### D.1.1   Test diet for food allergic patients

Food allergies are one of the most important diet-related issues. These can sometimes be fatal. Therefore, when planning a diet for patients with food allergies, menus should be carefully checked for the presence of allergens. The evaluator function in dietkit provides filters for ingredients. Therefore, this function can simply detect the menu items that can cause allergy problems. The package is also helpful in finding the appropriate replacement menus. The same evaluator function can be used to find menus that have the same main ingredients but do not contain allergens. For example, 'mushroom egg stew' is inappropriate for patients with an egg allergy. In this case, the package can be used to search for menus containing mushrooms, the main ingredient. Then, the user can exclude menus with eggs from the searched list. Through this process, the user can find candidates for alternative menus.

### D.1.2   Design supplementary diets

Many people receive some of their daily diets from the diet provision service of affiliated institutions such as a school or firm. In this case, nutritionally balancing the remainder of the daily diet at home is advisable. For example, if vitamin C is lacking in the lunch provided in school, consumption of fruits at dinner to supplement the missing nutrition would be desirable. The dietkit package and MIND dataset can be used to find answers regarding the nutrients to be replenished in this situation. The nutrition test function of our evaluator function can be used to determine any nutritional deficiencies in the provided diets by applying recommended daily nutritional criteria.

## D.2   Limitations and responsibly use

The diets provided by MIND are healthy "reference diets" for an unspecified majority of the population. Therefore, there is no guarantee that a user will have a positive response to the provided diets for their meals. Although our diets have been evaluated and confirmed by the related government organization and the affiliated professional dietitians, there is further need to consider health and preference issues. These include food allergies and a low-salt

diet preference. We do not consider these user-specific factors because our dataset was not created for a particular target group or for a customized purpose. Therefore, in order to use the diets provided in this MIND dataset, we recommend that the user inspect the ingredients and nutrition using the analysis functions provided in the dietkit package.

# E    Dataset distribution

## E.1    Access to the MIND dataset

The dataset can be accessed with dietkit in our GitHub repository: 'github.com/pki663/dietkit'. The dataset can be accessed from here: 'github.com/pki663/dietkit/tree/master/sample/data'. These resources are also distributed through the Python package index package manager system (pip): 'pypi.org/project/dietkit'.

## E.2    Dataset maintenance plan and future work

We will accommodate the suggestions and contributions for our dataset from the users through GitHub issues/commits, and all amendments will be explicitly recorded as commit commands. In addition, we plan to update the data approximately every year. For example, the National Standard Food Components, which is the original source of the ingredient data, is updated every year by the Rural Development Administration of the Korean government. Finally, the current data in the MIND are collected from Korean institutions and lack many of the dietary and food data on Western meals. We plan to supplement this dataset using additional appropriate data sources.

More specifically, we already started the projects to expand our dataset into multinational and multicultural contexts beyond Korea. In July, 2021, we received funding of USD 0.8 million from the National Research Foundation Korea (Grant Number: 2021R1I1A4A01049121) to use and improve the ORxML framework in order to expand our dataset. In this project, we will add diet sequence data from Western countries and other Asian countries, e.g., diets from North America, Europe, China, and Japan, to enhance the coverage and utility of our dataset. For this task, we will recruit domestic and international experts on specific national/cultural diets and collaborate with international restaurants in South Korea. Furthermore, with the goals of recruiting many experts and collaborating with various restaurants, we will continuously request additional funding from the South Korean government. Finally, as one of the authors is affiliated with a medical school in South Korea, we will continuously seek opportunities to work with more physicians to use our dataset for clinical studies of dietary healthcare research and services.

# F    About license and rights

Table 17: License information of the MIND dataset

| Data | Author Indication | Redistribution | Commerical | Derivative Works |
|------|-------------------|----------------|------------|------------------|
| **Ingredients** | Required | Allowed | Can be used for commerical | Free |
| **Menu** | Required | Allowed | Only for non-commerical | Free |
| **Diet** | Required | Allowed | Only for non-commerical | Free |

As described earlier, the MIND dataset has three sub-datasets with different license information. The ingredient dataset is distributed under the Korean Open Government License Type 1. In this license, a user can use the data freely and without fee, regardless of its commercial use, and can change or modify the data to create secondary works. However, the Rural Development Administration of the Korean government, which is the source of the data, must be indicated. The menu dataset is permitted to be used under these conditions: (1) the Center for Children's Food Service Management in South Korea should be indicated as the

author, (2) redistribution of the menu data and its derivative works are allowed, the menu dataset should be used only for non-commercial purposes, (3) the 'CC BY-NC 4.0' license is applied to the diet dataset, (4) when using the diet data, the authors of this paper should be indicated as the attribution parties, (5) the redistribution of diet data and its derivative works are allowed, and (6) the data should be used only for non-commercial purposes.