# OpenReview forum: "MIND dataset for diet planning and dietary healthcare with machine learning: Dataset creation using combinatorial optimization and controllable generation with domain experts"
_NeurIPS.cc/2021/Track/Datasets_and_Benchmarks/Round2 — NeurIPS 2021 Datasets and Benchmarks Track (Round 2)_

### Official Review · Reviewer_ttNw · 2021-09-20
**A well-thought solution to an important problem, with some clarity issues**

**Rating:** 6
**Confidence:** 3
**Correctness:** No issues other than those noted in "…

**Strengths:**

1. *Significance*. The authors are tackling an important problem. Both on ground-level advisory and more research-oriented settings in industry, the lack of high-quality data on diets and nutrition is indeed a major impediment to progress. To the best of my knowledge, this has not been tackled at this scale. The authors propose a hierarchical approach through multi-level relationships, and approach the problem through synthetic data generation, which solves the two primary hurdles---i.e. the unstructured nature of vast swaths of observational datasets out there, as well as the fact that no single consistent dataset is large enough for more in-depth research.

2. *Transparency*. On a high level, the 6-step structure proposed is clear. There appears to be no black magic and the authors make it known that what is being done is simply smartly combining known methods in operations research and machine learning with domain experts, to achieve the desired result. This is an advantage, as it is clear what goes into generating this dataset. Moreover, the process can be replicated (cf. "controlled generation") by other users that may desire a slightly different dataset.

3. *Human-in-the-Loop*. The use of domain experts in the loop is well-considered, especially at a time when the machine learning community as a whole is warming up to the notion that we can be smart about injecting human knowledge into models, instead of "learning from scratch".

**Weaknesses:**

1. *Definitions*. There needs to be a clear definition of the various hierarchical terms being used. Understanding the terminology took several reads, especially going back to Figure 1 repeatedly. The impression given is as follows:

- "Nutrients": The lowest level, i.e. carbs, proteins, fat, etc.

- "Ingredients": Each ingredient contains a combination of nutrients. There needs to be an example of what an ingredient is.

- "Menu": Each menu contains a combination of ingredients. This is the most confusing term! It appears what is being referred to is a "food item". If this is an idiosyncrasy of the field, this still needs to be defined, with examples of what it can be.

- "Diet": Each diet contains a combination of menus. Again, this needs to be defined more clearly, as "diet" can mean any number of things in vernacular. Does this simply refer to what we take to be a "meal"?

- Sequence: Each sequence contains an ordered combination of diets. There does not appear to be a specific term that is used consistently throughout for this. However, it is present as the red box in Figure 1, so perhaps this can also be defined.

Here is a suggestion: Somewhere in the introduction, it would be much clearer if there were an explicit definition of all of these terms, so that readers unfamiliar with the hierarchy of diet-related concepts would be initiated appropriately. This can also be a text-box. It just needs to be up front. (At the moment some of these only make more sense after reading Appendix B).

2. *Vague Terms*. Some terms would benefit from more concrete illustration (either through concrete examples or concrete criteria) in the main manuscript:

- "Nutrient requirements": It appears this refers to the combinatorial optimization problem. What are the constraints, and are the constraints specified externally (i.e. this is simply RDI), or are some constraints introduced by the authors? This needs to be clearer.

- "Compositional quality": It appears that this refers to literally anything that humans may desire (i.e. "soft criteria") that are not captured by the combinatorial optimization problem (which could conceivably yield very odd meal recommendations). Is this correct? This needs to be clearer, with better examples.

3. *Clarity of Evaluation*. At the moment the evaluation section discusses evaluating RDI, percent mis-positioned, and Chi-square. Two questions:

- How is the RDI score (which is defined as "nutritional excellence") computed? How is the mis-positioning determined (i.e. relative to what correct ordering)? There are details in the Appendix, but at least one or two lines of explanation in the main manuscript would be good.

- Some of these measures are the input to the method. For example, RDI is the input to the OR component, which explains why that component does best on RDI score. Is correct positioning implicitly part of what the expert component encourages when they edit the plans? Same with Chi-square. This is important to briefly discuss, because that would give some context as to why the numbers are the way they are.

**Additional Feedback:**

Minor: The typesetting used appears to be incorrect. Is the correct latex template being used?

**Clarity:**

Apart from the specific clarity concerns above, in general the paper is well-organized on a high level. Minor comments:

- "RDI is a shortcut for recommended daily intake" --> shorthand?

- In Figure 1, it may be easier to understand if these were bolded and put first: "Food Item Level", "Menu Item Level", etc., and these are put second (or even removed): "diet problem", "diet planning", and "diet planning with ML", which are too vague.

**Documentation:**

There is sufficient detail of these considerations in Appendices C, D, and E.

**Ethics:**

There is sufficient detail in Appendix C.2.

**Relation To Prior Work:**

To the best of my knowledge, the relationship to prior work is covered satisfactorily.

**Summary And Contributions:**

This paper proposes a multi-level synthetic dataset of information useful for diet-planning research.

In order to generate such a dataset, the authors propose a six-step methodology, incorporating (1) a component driven by operations research for the first two, i.e. in formulating and filtering for solutions in the combinatorial optimization problem; (2) a component driven by intervention by domain experts in the middle two, i.e. in accounting for implicit and human factors for "compositional quality"; and (3) a component driven by machine learning for the last two, i.e. in learning the patterns in the previous two components to generate synthetic diets, as well as explaining latent elements that are not explicit to experts.

The full dataset is presented in three databases, each containing synthetic examples illustrating the relationships between consecutive levels in the full hierarchy required for diet-planning---relating (a) nutrients to ingredients, (b) ingredients to menus, and (c) menus to diets. A discussion is provided on the evaluation of the proposed method of generation, as well as the final output dataset; these include (i) conformity to nutritional requirements, (ii) compositional quality of diets, and (iii) homogeneity of ingredient usage. It is shown that the machine learning module partially recovered some of the nutritional quality "lost" by the expert component, and it also outperforms even the expert

---

> ### Author Response · Authors · 2021-09-28
> **Response to Comment 4.6**
>
> Comment 4.6: Rating: 6: Marginally above acceptance threshold, Confidence: 3: The reviewer is fairly confident that the evaluation is correct
>
> Response: Once again, thank you very much for your positive comments and valuable suggestions. We sincerely hope that we have successfully clarified our contribution and enhanced the presentation of our work by addressing your comments. We hope that these adjustments increased your confidence in our work and have provided you with a more positive view of our work. If this is the case, may we humbly and respectfully ask that you elevate your ratings of our work with your increased confidence? Once again, thank you very much for your positive comments and valuable suggestions. We are pleased to be engaged with you as our reviewer and appreciate your incisive comments, which helped us improve our work significantly. Please let us know if you have any other concerns, questions, or suggestions. We would love to incorporate your additional feedback into our work.

---

> ### Author Response · Authors · 2021-09-28
> **Response to Comment 4.5**
>
> Comment 4.5: Additional Feedback: Minor: The typesetting used appears to be incorrect. Is the correct latex template being used?
>
> Response: Thank you again for your careful review and thoughtful comment. We checked our overleaf file again, but there was no compile issue on the file. If there is the same problem in the updated version submitted, please kindly let us know.

---

> ### Author Response · Authors · 2021-09-28
> **Response to Comment 4.4**
>
> Comment 4.4:
> (a) Correctness: No issues other than those noted in “clarity of evaluation” above.
>
> (b) Clarity: Apart from the specific clarity concerns above, in general the paper is well-organized on a high level. Minor comments: “RDI is a shortcut for recommended daily intake” --> shorthand? In Figure 1, it may be easier to understand if these were bolded and put first: “Food Item Level”, “Menu Item Level”, etc., and these are put second (or even removed): “diet problem”, “diet planning”, and “diet planning with ML”, which are too vague.
>
> (c) Relation To Prior Work: To the best of my knowledge, the relationship to prior work is covered satisfactorily.
>
> (d) Documentation: There is sufficient detail of these considerations in Appendices C, D, and E.
>
> (e) Ethics: There is sufficient detail in Appendix C.2.
>
> Response: Once again, thank you very much for your careful review and incisive comments.
>
> (a) We hope this comment has been addressed with our responses to your Comments 4.3(c). Thank you again for your constructive comments and valuable suggestions.
>
> (b) You are correct. Thank you very much for your careful reading and precise comment. We have edited the typo from “shortcut” to “shorthand”. Thank you for your next comment as well. Taking your advice into account, we revised Figure 1 such that the data (“Food Item Level”, “Menu Item Level”, and “Diet Sequence Level”) are put first and bolded, while the problems (e.g., “diet problem”, “diet planning”, and “diet planning with ML”) are put second. Meanwhile, we believe that the ambiguity you mentioned regarding the definition of problems is probably due to the fact that the problem concept first appeared in Section 1, but their concrete definitions were presented in Section 2. Therefore, we added some sentences that briefly introduce and compare the problems in Section 1 to ensure readability and clarity. Thank you very much for your valuable suggestions to enhance the presentation of our work.
>
> (c) Thank you very much for your positive feedback. Although you thankfully mentioned that you are satisfied with the relation to prior work, please note that we have elaborated on our literature review in Sections 1 and 2. For example, we have clarified how existing literature differs from our work. In summary, all previous studies on diet planning consider the “ingredient and menu level” information whereas “diet level” planning should involve the compositional patterns of menus in diets. In addition, existing studies on dietary healthcare with machine learning (e.g., Zeevi et al., 2015 published in Cell) consider only the ingredient and menu levels. Diet planning is a highly important problem that requires solutions with machine learning but cannot be addressed as such due to the lack of datasets. We clarified that this work originally solves this concern, and our dataset is the first high-quality resource to promote the research field of “diet planning with machine learning”. Accordingly, we have edited Figure 1 in the Introduction to serve as a visual taxonomy or landscape that shows the spectrum from existing works (e.g., diet problem and diet planning with OR) to ours (i.e., diet planning and dietary healthcare with ML). Finally, we have added clear definitions in the Introduction with the revised Figure 1 as well as elaborated the definitions in Section 5 based on your Comment 4.3. Please note that we have highlighted the revised or added sentences in green font.
>
> (d) Thank you very much for your positive feedback. Nonetheless, please note that in this revision process, we have tried hard to further improve the documentation of our dataset creation and management processes. For example, we have elaborated the validation of our work evaluated by many professional dietitians in Section 4. For another example, as we have described in Section 5 and Supplementary material D.2, our dataset will be managed and updated regularly through our research on diet planning and dietary healthcare with ML (e.g., extending the size and information of our dataset in accordance with the updates of related public databases. Our next objective is to extend the high-quality 1,500 diet data to 3,000 data using the ORxML framework.
>
> (e) Yes, there are no ethical issues. Furthermore, as shown in the Supplementary material E, we have prepared all the license issues for the MIND dataset to ensure that there would be no credit or authority issues.

---

> ### Author Response · Authors · 2021-09-28
> **Response to Comment 4.3: Weaknesses(c)**
>
> (c) “RDI score” is calculated by overall nutrition of the diet and determined by how many of the nutrition criteria are satisfied. As mentioned above, the used criteria are referenced from the 2020 Dietary Reference Intakes for Koreans. The overall nutrition of the diet is calculated by summing up the nutrition of each consisting of ingredients.
>
> A menu is considered “mispositioned” when it is a side dish but located in the position of the main dish, or vice versa. In our sample diet data, main dishes should be located on the 3rd, 10th, and 17th, and side dishes are on the 4th, 11th, and 18th position. Also, the menu data indicate whether a menu is the main dish or side dish. This indication was made by the expert dietitians. As such, when a menu is in a completely wrong position (e.g. side dish in the 4th position), we counted it and calculated that total ratio of mispositioned menus in diets.
>
> “Chi-square” is an indicator of how similar the pattern of ingredient usage is between the generated diets and real diets. Since the usage of ingredients has a strong influence on the properties of the diet, like taste, color, and texture, this is a valid indicator to measure the compositional quality of diets.
>
> Finally, as you mentioned, RDI is the input to the OR module, therefore this module does best on RDI score. Also, correct positioning was one of the objectives when the expert module edits the OR-generated diets. Please see Appendix A.2 for further details on how the ORxML framework addressed the requirements and the difficulties of diet planning.
>
> We added all the above information in green font to Section 4. Please note that we have highlighted the revised or added sentences in green font. Once again, thank you very much for your comments that helped us significantly improve the clarity of our work.

---

> ### Author Response · Authors · 2021-09-28
> **Response to Comment 4.3: Weaknesses(b)**
>
> (b) “Nutrient requirements” is defined by the external literature. In this work, we referenced the 2020 Dietary Reference Intakes for Koreans. (Please see Table 5 in the Supplementary material). This reference is provided by the reliable institution, the Ministry of Health and Welfare of the Korean government. The reason we chose this reference is that our diet dataset consists mainly of Korean menus, and this document is a nutritional standard published by one of the most reliable organizations about Korean menus. As requested, we have clearly stated this reference information in the revised Section 4. Please note that we have highlighted the revised or added sentences in green font.
>
> Meanwhile, “Compositional quality” is, as you mentioned, difficult to be captured by the combinatorial optimization problem and even by humans explicitly. This factor is dependent on the contexts and implicit in nature; therefore, even experts feel difficulties to define it. Thus, we tried our best to define it as clearly as possible through the collaboration with many professional dietitians. Please see the Supplementary material A.3 (Diet planning and diet data generation) for the details. In addition, please refer to Appendix A.4 (Explicit and implicit criteria of diet planning) for further details of the compositional requirements of diet planning and the related literature. As requested, we have clearly stated this indication in the revised Section 4.

---

> ### Author Response · Authors · 2021-09-28
> **Response to Comment 4.3: Weaknesses(a)**
>
> Comment 4.3: Weaknesses:
> (a) Definitions. There needs to be a clear definition of the various hierarchical terms being used. Understanding the terminology took several reads, especially going back to Figure 1 repeatedly. The impression given is as follows: “Nutrients”: The lowest level, i.e. carbs, proteins, fat, etc. “Ingredients”: Each ingredient contains a combination of nutrients. There needs to be an example of what an ingredient is. “Menu”: Each menu contains a combination of ingredients. This is the most confusing term! It appears what is being referred to is a “food item”. If this is an idiosyncrasy of the field, this still needs to be defined, with examples of what it can be. “Diet”: Each diet contains a combination of menus. Again, this needs to be defined more clearly, as “diet” can mean any number of things in vernacular. Does this simply refer to what we take to be a “meal”? Sequence: Each sequence contains an ordered combination of diets. There does not appear to be a specific term that is used consistently throughout for this. However, it is present as the red box in Figure 1, so perhaps this can also be defined. Here is a suggestion: Somewhere in the introduction, it would be much clearer if there were an explicit definition of all of these terms, so that readers unfamiliar with the hierarchy of diet-related concepts would be initiated appropriately. This can also be a text-box. It just needs to be up front. (At the moment some of these only make more sense after reading Appendix B).
>
> (b) Vague Terms. Some terms would benefit from more concrete illustration (either through concrete examples or concrete criteria) in the main manuscript: “Nutrient requirements”: It appears this refers to the combinatorial optimization problem. What are the constraints, and are the constraints specified externally (i.e. this is simply RDI), or are some constraints introduced by the authors? This needs to be clearer. “Compositional quality”: It appears that this refers to literally anything that humans may desire (i.e. “soft criteria”) that are not captured by the combinatorial optimization problem (which could conceivably yield very odd meal recommendations). Is this correct? This needs to be clearer, with better examples.
>
> (c) Clarity of Evaluation. At the moment the evaluation section discusses evaluating RDI, percent mis-positioned, and Chi-square. Two questions: How is the RDI score (which is defined as “nutritional excellence”) computed? How is the mis-positioning determined (i.e. relative to what correct ordering)? There are details in the Appendix, but at least one or two lines of explanation in the main manuscript would be good. Some of these measures are the input to the method. For example, RDI is the input to the OR component, which explains why that component does best on RDI score. Is correct positioning implicitly part of what the expert component encourages when they edit the plans? Same with Chi-square. This is important to briefly discuss, because that would give some context as to why the numbers are the way they are.
>
> Response: First of all, thank you very much for your kind and thoughtful comments.
>
> (a) Thank you for valuable suggestions. As you point out, we agree on the need for “a clear definition of hierarchical terms”. Thus, we have added definitions of diet, menu, ingredient, and nutrition in Sections 1 and 5. In addition, the hierarchical relationship between them has been described in Sections 1 and 5, as well as the Supplement material. Meanwhile, “diet” and “meal” are strictly different terms from a nutritional or healthcare point of view. The reference [SmartDiet: A personal diet consultant for healthy meal planning, A Definition of “Regular Meals” Driven by Dietary Quality Supports a Pragmatic Schedule] indicates that the term “meal” implies eaten or consumed foods in general, whereas the term “diet” is used to indicate the combination of foods planned for a specific purpose such as nutritional satisfaction, allergen avoidance, or weight control. Finally, we have clarified why and how a diet should be defined as a sequence in detail in Sections 1 and 3. Please note that we have highlighted the revised or added sentences in green font.

---

> ### Author Response · Authors · 2021-09-28
> **Response to Comment 4.2: Strengths(c)**
>
> (c) Yes, thank you very much for highlighting this important point. Please let us describe our thoughts that concur with your thoughts as follows. Despite its dramatic success in many fields, from academia to industry, machine learning still suffers from critical shortcomings. In particular, a huge amount of data needs to be used when training a machine learning model so that it can subsequently attain an acceptable level of model performance. In particular, we found that there is a large problem of data insufficiency when we need to develop a machine to support “professional tasks” that are difficult to perform or describe, even among experts. The quality of some professional tasks, such as diet planning and drug discovery cannot be easily trusted because even experts—including scientists and physicians—are not skilled enough to optimally execute these tasks. The challenge is attributed to the inherent complexity of the task that involves both explicit and implicit criteria. Given this challenge, ML models trained on available but poor-quality records of the task cannot be fully utilized because the model will not perform well. Moreover, the development of a simulation environment to generate realistic data of these tasks can be almost impossible because of task complexity. Ironically, the value of learning and automating professional tasks is high given the difficulty and importance of these tasks, requiring assistance from machines.
>
> This challenge motivated us to devise the ORxML framework. As described in Section 3, our proposed ORxML framework addresses this challenge using the “human-in-the-loop” as you highlighted upon. Specifically, the ORxML framework integrates three modules: 1) the OR module to generate initial data satisfying the explicit criteria of a professional diet planning task, 2) professional dietitians, who evaluate and adjust the initial data, considering the implicit criteria of the task (i.e., criteria that cannot be modeled), and 3) the ML module to control the diet generation and tune the quality and completeness of the data. We believe our ORxML framework used for the diet data creation will contribute to the researchers and practitioners who deal with real-world applications of machine learning for professional tasks.
>
> Thanks to your Comment 4.2(c), we were able to elaborate our paper significantly. Please see the revised conclusion Section 7 and Appendix A.7, with changes shown in green font, for further details. By addressing your comment, we feel that the contribution of our work has been further clarified. Thank you.

---

> ### Author Response · Authors · 2021-09-28
> **Response to Comment 4.2: Strengths(a)(b)**
>
> Comment 4.2: Strengths:
>
> (a) Significance. The authors are tackling an important problem. Both on ground-level advisory and more research-oriented settings in industry, the lack of high-quality data on diets and nutrition is indeed a major impediment to progress. To the best of my knowledge, this has not been tackled at this scale. The authors propose a hierarchical approach through multi-level relationships, and approach the problem through synthetic data generation, which solves the two primary hurdles---i.e. the unstructured nature of vast swaths of observational datasets out there, as well as the fact that no single consistent dataset is large enough for more in-depth research.
>
> (b) Transparency. On a high level, the 6-step structure proposed is clear. There appears to be no black magic and the authors make it known that what is being done is simply smartly combining known methods in operations research and machine learning with domain experts, to achieve the desired result. This is an advantage, as it is clear what goes into generating this dataset. Moreover, the process can be replicated (cf. “controlled generation”) by other users that may desire a slightly different dataset.
>
> (c) Human-in-the-Loop. The use of domain experts in the loop is well-considered, especially at a time when the machine learning community as a whole is warming up to the notion that we can be smart about injecting human knowledge into models, instead of “learning from scratch”.
>
> Response: Once again, thank you very much for highlighting our unique contributions.
>
> (a) As we responded to your Comment 4.1, the task of diet planning is actually the problem that should be solved with ML (to consider both nutrient and composition requirements) beyond operations research (which is mainly able to consider nutrient requirements). However, this problem could not be addressed in this way due to the lack of datasets for this data-driven way. Our dataset is the first high-quality resource for this research direction. As diet planning—a basic and regular human activity—is important to all individuals, from children to seniors and from healthy people to patients. Related to this, we believe our paper can become a seminal work that can help promote the research field of “diet planning and dietary healthcare with machine learning.”
>
> (b) Yes, as we responded to your Comment 4.1, we believe the ORxML framework can further encourage research in the area of diet planning. Furthermore, beyond creating any other contexts of generating diet data (e.g., diet sequence data in Western countries and other Asian countries), this framework can be used to create the data of difficult professional design tasks (e.g., data of fashion item sequence to fulfill explicit seasonal and implicit preference requirements). As highlighted by you in your Comment 1.3(b) and Comment 4.2(c) from Reviewer 4, the unique advantage of this framework is to combine the capabilities of OR (for explicit requirements of data of difficult professional design tasks) experts (for implicit requirements), and ML (for both) to create a high-quality synthetic dataset for machine learning research and development. We sincerely hope that this human-in-the-loop framework will be used by other researchers for other contexts of creating diet data and the data of difficult professional design tasks.
> Thanks to your Comments 4.2(a)(b), we were able to elaborate our paper significantly. Please see the revised Section 1 and Appendix A.7, with changes shown in green font, for further details. By addressing your comment, we feel that the contribution of our work has been further clarified. Thank you.

---

> ### Author Response · Authors · 2021-09-28
> **Response to Comment 4.1**
>
> Dear Reviewer 4, Thank you for your careful reading of our work and positive feedback. Thank you very much for your kind words “A well-thought solution to an important problem”. Please kindly allow us to further highlight the importance of our contributions as follows. We trust that the following paragraphs can help discuss our response to your comments.
>
> As highlighted by you and all the other reviewers, our first original contribution is to create the “first large-scale and high-quality” dataset for “diet planning with machine learning (ML)”. We believe that this contribution is significant and meaningful because diet planning is mainly considered an operations research (OR) problem (particularly a combinatorial optimization), which aims to fulfill nutrient requirements. However, this approach is ineffective in practice due to its limited capability to consider the implicit patterns in diets that professional dietitians believe as the most important (see Section 1 and Appendix A.4). In our paper, we originally clarify that the diet planning problem should actually be considered as a sequence generation to accommodate these implicit patterns (i.e., to fulfill the “compositional” requirements of diet planning) and tackle the concern on the lack of high-quality dataset to promote “diet planning with ML”.
>
> Specifically, diet planning—a basic and regular human activity—is important to all individuals, from children to seniors and from healthy people to patients; therefore, the history of this research stream is in fact quite long. Around 1945, diet planning was first defined as a linear programming problem to identify optimal quantities of food items. However, humans do not consume a specific quantity of each food unit (i.e., combination of ingredients cooked) but rather the end-product (i.e., food “menu”) as an entire unit. Thus, around 1990, diet planning has expanded into a mixed-integer programming problem to identify optimal combinations of menu items. However, all previous studies only considered the “ingredient and menu level” information whereas “diet level” planning involves the aforementioned compositional patterns of menus. Compositional patterns are implicit depending upon their contexts, and thus require data-driven machine learning approaches. Surprisingly, apart from our work, no other study seems to focus on real diet-level planning, for which we believe the main reason is the lack of high-quality diet dataset to research data-driven machine learning approaches. To the best of our knowledge, existing studies on the application of machine learning to dietary healthcare (e.g., Zeevi et al., 2015 published in Cell) consider only the ingredient and menu levels.
>
> In summary, diet planning is a highly important problem that needs machine learning solutions but could not be addressed as such due to the lack of datasets. In this paper, our dataset is the first high-quality resource to promote the research of “diet planning with machine learning”. Thanks to the allowance of one additional page, we could include machine learning use cases of diet planning and dietary healthcare with our dataset in Section 6; some of the cases are ongoing research of the authors with other colleagues. Through research and development of such use cases, we sincerely hope that the proposed dataset can be widely disseminated for diet planning and dietary healthcare with machine learning.
>
> Furthermore, we believe that our dataset creation framework, OR–Xperts–ML (ORxML), is also unique and relevant to the dataset creation efforts for machine learning research (e.g., the NeurIPS 2021 Datasets and Benchmarks Track). Our second contribution is to devise the ORxML framework to systematically create a large-scale high-quality synthetic diet dataset. As described in our paper, given the high complexity of diet planning (see Section 1 and Appendix A.4), real data are not high quality despite being created by professional dietitians. Thus, we had to create a novel high-quality dataset and devised the ORxML framework that integrates the capabilities of three modules: the OR module (i.e., a combinatorial optimization model) generates synthetic diets to satisfy explicit nutrient requirements; the expert module evaluates and adjusts the initial data in terms of implicit composition requirements (i.e., criteria that cannot be specified in the combinatorial optimization model); and the ML module automatically augments the data to ensure the composition compliance with nutrition enhancement. A series of experiments demonstrate the significance of the three modules and validate the quality of our dataset (see Section 4 and Appendix A.6). Finally, please note that this framework can be used in any other contexts of creating diet data and those of difficult professional tasks as well.
>
> For further details on this revision to clarify the abovementioned contribution points, please see the following point-to-point responses to your comments.

---

> ### Author Response · Authors · 2021-09-28
> **Comment 4.1**
>
> Comment 4.1: “A well-thought solution to an important problem, with some clarity issues” - Summary And Contributions: This paper proposes a multi-level synthetic dataset of information useful for diet-planning research. In order to generate such a dataset, the authors propose a six-step methodology, incorporating (1) a component driven by operations research for the first two, i.e. in formulating and filtering for solutions in the combinatorial optimization problem; (2) a component driven by intervention by domain experts in the middle two, i.e. in accounting for implicit and human factors for “compositional quality”; and (3) a component driven by machine learning for the last two, i.e. in learning the patterns in the previous two components to generate synthetic diets, as well as explaining latent elements that are not explicit to experts. The full dataset is presented in three databases, each containing synthetic examples illustrating the relationships between consecutive levels in the full hierarchy required for diet-planning---relating (a) nutrients to ingredients, (b) ingredients to menus, and (c) menus to diets. A discussion is provided on the evaluation of the proposed method of generation, as well as the final output dataset; these include (i) conformity to nutritional requirements, (ii) compositional quality of diets, and (iii) homogeneity of ingredient usage. It is shown that the machine learning module partially recovered some of the nutritional quality “lost” by the expert component, and it also outperforms even the expert

---

> ### Author Response · Authors · 2021-09-28
> **Dear Reviewer 4 (ttNw), Thank you very much for your positive comments and valuable suggestions. Please see our responses to your comments as follows.**
>
> Dear Reviewer 4 (ttNw), Thank you very much for your positive comments and valuable suggestions. Please see the following point-to-point responses to your comments. In addition, could you please kindly check and refer to the  submitted Supplementary Material for our full response letter to all comments from you? We prepared such a document to address the limit of characters in this openreview platform. Thank you very much for your understanding.
>
> We are pleased to be engaged with you as our reviewer and appreciate your incisive comments, which helped us improve our work significantly. Please let us know if you have any other concerns, questions, or suggestions. We would love to incorporate your additional feedback into our work. Through this discussion period, we will continuously update our submission and response as appropriate.

---

### Official Review · Reviewer_exb7 · 2021-09-20
**Useful diet planning dataset**

**Rating:** 7
**Confidence:** 4

**Strengths:**

The significance of the work described in this paper is large since diet planning is a very important task in the domain of food and nutrition sciences. So having such a developed resource would lead to further developments of (semi)automatic methods for different aspects of software development in diet planning applications. Hence this work is important for the broader research community. The proposed resource is public and is easily accessible by using a Python library (dietkit). Given that this dataset is synthetic there are no open ethical issues.


**Weaknesses:**

The limitations of the proposed dataset have not been explicitly discussed in the paper. In addition, one would expect a more elaborate discussion of the potential use cases and the advantages this new dataset would bring to the applications that would use the data. This is somewhat given in the Supplementary Material, but having a paragraph or two in the main text would significantly improve the paper.

**Additional Feedback:**

- Incorporate a discussion about potential use cases in the main text
- Discuss the limitations of the use of the proposed dataset in the main text

**Clarity:**

In my opinion, the paper is well written and contains enough information for researchers to start using the presented dataset as well as understand the process of dataset construction.

**Correctness:**

In my opinion, the claims made in the paper are correct and the dataset was constructed in a sound way taking into account all relevant data sources and expert knowledge.

**Documentation:**

The authors have provided sufficient details on data collection and organization, availability and maintenance, and ethical and responsible use. They have also included documentation and intended uses of the proposed dataset both in the main paper and the appendix. The dataset is publically available at Zenodo and has a DOI (10.5281/zenodo.5302044) and the dietkit Python library is available at GitHub and PyPI.

**Ethics:**

Given that this dataset is synthetic I believe that there are no open ethical issues.

**Relation To Prior Work:**

The authors have discussed all relevant prior work in the context of diet planning theory and resources in general, as well as discussed the importance to combine existing operations research approaches

**Summary And Contributions:**

In this paper, the authors present a novel dataset for diet planning and healthcare research. The authors describe the process of dataset creation by combining approaches from operations research, machine learning and knowledge from domain experts, as well as evaluation of the generated resource. The authors claim that their resource is the first resource for diet planning of that type. The contributions of the paper include: the K-MIND dataset and a diet planning methodology that combines knowledge from operations research models, machine learning and domain experts

---

> ### Author Response · Authors · 2021-09-28
> **Response to Comment 3.6**
>
> Comment 3.6: Rating: 7: Good paper, accept, Confidence: 4: The reviewer is confident but not absolutely certain that the evaluation is correct
>
> Response: Once again, thank you very much for your positive comments and valuable suggestions. We sincerely hope that we have successfully clarified our contribution and enhanced the presentation of our work by addressing your comments. We hope that these adjustments increased your confidence in our work and have provided you with a more positive view of our work. If this is the case, may we humbly and respectfully ask that you elevate your ratings of our work with your increased confidence? Once again, thank you very much for your positive comments and valuable suggestions. We are pleased to be engaged with you as our reviewer and appreciate your incisive comments, which helped us improve our work significantly. Please let us know if you have any other concerns, questions, or suggestions. We would love to incorporate your additional feedback into our work.

---

> ### Author Response · Authors · 2021-09-28
> **Response to Comment 3.5**
>
> Comment 3.5: Additional Feedback: (a) Incorporate a discussion about potential use cases in the main text (b) Discuss the limitations of the use of the proposed dataset in the main text
>
> Response: We believe this comment has been addressed with our responses to your Comments 3.3(a)(b). Thank you again for your valuable suggestion.

---

> ### Author Response · Authors · 2021-09-28
> **Response to Comment 3.4**
>
> Comment 3.4:
> (a) Correctness: In my opinion, the claims made in the paper are correct and the dataset was constructed in a sound way taking into account all relevant data sources and expert knowledge. (b) Clarity: In my opinion, the paper is well written and contains enough information for researchers to start using the presented dataset as well as understand the process of dataset construction.
>
> (c) Relation To Prior Work: The authors have discussed all relevant prior work in the context of diet planning theory and resources in general, as well as discussed the importance to combine existing operations research approaches
>
> (d) Documentation: The authors have provided sufficient details on data collection and organization, availability and maintenance, and ethical and responsible use. They have also included documentation and intended uses of the proposed dataset both in the main paper and the appendix. The dataset is publically available at Zenodo and has a DOI (10.5281/zenodo.5302044) and the dietkit Python library is available at GitHub and PyPI.
>
> (e) Ethics: Given that this dataset is synthetic I believe that there are no open ethical issues.
>
> Response: Thank you for your positive comment and suggestion.
>
> (a) (b) Although we are thankful that you are already satisfied with the correctness and clarity of our paper, please note that in this revision, we attempted to further clarify the dataset creation and management processes by addressing the reviewers’ comments. For example, we have further explained how our ORxML framework is being used now and in the future to expand the MIND dataset into multinational and multicultural contexts. Finally, please note that we used a copyediting service for the updated manuscript to check English grammar and ensure readability.
>
> (c) Thank you very much for your positive feedback. Although you thankfully mentioned that you are satisfied with the relation to prior work, please note that we have elaborated on our literature review in Sections 1 and 2. For example, we have clarified how existing literature differs from our work. In summary, all previous studies on diet planning consider the “ingredient and menu level” information whereas “diet level” planning should involve the compositional patterns of menus in diets. In addition, existing studies on dietary healthcare with machine learning (e.g., Zeevi et al., 2015 published in Cell) consider only the ingredient and menu levels. Diet planning is a highly important problem that requires solutions with machine learning but cannot be addressed as such due to the lack of datasets. We clarified that this work originally solves this concern, and our dataset is the first high-quality resource to promote the research field of “diet planning with machine learning”. Accordingly, we have edited Figure 1 in the Introduction to serve as a visual taxonomy or landscape that shows the spectrum from existing works (e.g., diet problem and diet planning with OR) to ours (i.e., diet planning and dietary healthcare with ML). Finally, we have added clear definitions in the Introduction with the revised Figure 1 as well as elaborated the definitions in Section 5.
>
> (d) Thank you very much for your positive feedback. Nonetheless, please note that in this revision process, we have tried hard to further improve the documentation of our dataset creation and management processes. For example, as we have described in Section 5 and Supplementary material D.2, our dataset will be managed and updated regularly through our research on diet planning and dietary healthcare with ML. For example, we will extend the size and information of our dataset in accordance with the updates of related public databases (e.g., ingredient-nutrition databases of the Rural Development Administration of the Korean government). As for our abovementioned new project granted from the National Research Foundation Korea (Grant Number: 2021R1I1A4A01049121), our next objective is to extend the high-quality 1,500 diet data to 3,000 data using the ORxML framework.
>
> (e) Yes, there is no ethical issues. Furthermore, as shown in the Supplementary material E, we have prepared all the license issues for the MIND dataset to ensure that there would be no credit or authority issues.

---

> ### Author Response · Authors · 2021-09-28
> **Comment 3.3: Weaknesses(a)**
>
> Comment 3.3: Weaknesses:
> (a) The limitations of the proposed dataset have not been explicitly discussed in the paper.
>
> Response:
> (a) Likewise, there was a paragraph describing the limitations of our work in the Supplementary material of our draft paper. In the updated version newly submitted, we explicitly discuss the limitations of our work and corresponding future research issues in the revised Section 6.
> We believe that the contribution and positioning of our work in the fields of machine learning, diet planning, and dietary healthcare research have been strengthened and clarified significantly by addressing your comment. Thank you very much.

---

> ### Author Response · Authors · 2021-09-28
> **Comment 3.3: Weaknesses(b)**
>
> Comment 3.3: Weaknesses:
> (b) In addition, one would expect a more elaborate discussion of the potential use cases and the advantages this new dataset would bring to the applications that would use the data. This is somewhat given in the Supplementary Material, but having a paragraph or two in the main text would significantly improve the paper.
>
> (b) We were happy to see this positive comment as we actually wished to discuss the applications of our work in the main text but could not do so due to page limitations. Thanks to the allowance of one additional page, we could include the applications of our work (e.g., machine learning use cases of diet planning and dietary healthcare) in Section 6; in addition, we added new use cases with real experiment results with the proposed MIND dataset in Section 6 and Appendix A.7. Please see the revised Section 6 and Appendix A.7. Some of the cases are ongoing research of the authors as exemplified below. Based on the exemplified use cases, we sincerely hope that the proposed dataset can be widely disseminated for diet planning and dietary healthcare with machine learning.
>
> For example, we are currently conducting a clinical study for children with atopic dermatitis (AD) and food allergy (FA) who must restrict allergenic foods which may lead to fatal anaphylaxis. The parental burden of these patients is very high because the children are limited to select social activities (e.g., camp and party) and restaurants due to the accidental exposure of allergenic food (Stensgaard et al., 2017 published in Clinical & Experimental Allergy). The children should eat at home or pack a lunch box. Thus, precise diet planning and dietary healthcare are necessary to manage the growth and health levels of the children. The authors are working together to develop an AI service application for this purpose. In fact, this work on creating a novel high-quality diet dataset was initiated for our clinical study for children with AD and FA; as real diet data in practice were not high quality enough to be used to train a machine for the AI service application.
>
> We are glad to tell you that recently we received a conditional approval from the IRB (Institutional Review Boards) for our clinical study on September 15th 2021 (IRB number: KUGH IRB No. 2021-09-019). To the best of our knowledge, this is the first clinical study to use and test the utility of an AI for diet-level planning and healthcare for patients. In this clinical study, the quantifiable health measures you inquired upon include the Food Allergy Quality of Life-Parental burden, Food Allergy Quality Of Life Questionnaire-Parent Form, Food Allergy Independent Measure, nutrient assessment using 24-hour dietary recall, and food frequency questionnaires, growth status, and other life satisfaction indices. These measures are validated measures in the clinical studies on patients with restricting foods (please see the references in Appendix A.7). Using these measures, our clinical study will evaluate the utility of our machine for diet-level planning and healthcare in the AI service user group and the control group. We believe there will be a significant difference between the two groups and expect to start this clinical study on October or November 2021.
>
> Please note that, as we have highlighted several times earlier, a high-quality large-scale diet dataset is the basis for such clinical studies on diet planning and dietary healthcare with ML that could not be conducted due to its nonexistence. Our MIND dataset would be dedicated to diet planning and dietary healthcare not only for the contexts of allergic diseases but also for other contexts of chronic diseases such as diabetes mellitus, hypertension, obesity, celiac disease, gastrointestinal caner, liver cirrhosis, and chronic kidney failure. Although there exist digital healthcare services for obesity, existing services only support the assessment of what patients eat and recommend menus (not a way tailored to the individual needs). In the contexts of diabetes and mellitus, the proportion of carbohydrate, protein and fat must be considered carefully in diet planning, while the compositional compliance is still important. For patients with chronic kidney diseases, diets should be composed of menus with less potassium, phosphorus, and calcium. As such, the needs and requirements of diet planning and dietary healthcare for chronic diseases are all different, and it is always important to design compositionally quality diets that the patients would take. The MIND dataset is the first high-quality resource for this research direction and the ORxML framework can further encourage this research. Physicians and dietitians have been educated patient dietary management at the nutrition and menu levels, and our work can contribute to expanding patient dietary management at the diet level which patients actually need.

---

> ### Author Response · Authors · 2021-09-28
> **Response to Comment 3.2: Strengths**
>
> Comment 3.2: Strengths:
> (a) The significance of the work described in this paper is large since diet planning is a very important task in the domain of food and nutrition sciences. So having such a developed resource would lead to further developments of (semi)automatic methods for different aspects of software development in diet planning applications. Hence this work is important for the broader research community. The proposed resource is public and is easily accessible by using a Python library (dietkit).
>
> (b) Given that this dataset is synthetic there are no open ethical issues.
>
> Response: Once again, thank you very much for highlighting our two unique contributions.
>
> (a) As we responded to your Comment 3.1(a), the task of diet planning is actually the problem that should be solved with ML (to consider both nutrient and composition requirements) beyond operations research (which is mainly able to consider nutrient requirements). However, this problem could not be addressed in this way due to the lack of datasets for this data-driven way. Our dataset is the first high-quality resource for this research direction. As diet planning—a basic and regular human activity—is important to all individuals, from children to seniors and from healthy people to patients. Related to this, we believe our paper can become a seminal work that can help promote the research field of “diet planning and dietary healthcare with machine learning.”
>
> (b) Yes, there is no ethical issues. Meanwhile, please note that the ORxML framework we devised can be used to create any other contexts of generating diet data (e.g., diet sequence data in Western countries and other Asian countries) as well as the data of difficult professional design tasks (e.g., data of fashion item sequence to fulfill explicit seasonal and implicit preference requirements). As we responded to your Comment 3.1(b), we believe this point is our another contribution and strength for the fields of machine learning, diet planning, and dietary healthcare research.

---

> ### Author Response · Authors · 2021-09-28
> **Response to Comment 3.1(b)**
>
> (b) Furthermore, we believe that our dataset creation framework, OR–Xperts–ML (ORxML), is also unique and relevant to the dataset creation efforts for machine learning research (e.g., the NeurIPS 2021 Datasets and Benchmarks Track). Our second contribution is to devise the ORxML framework to systematically create a large-scale high-quality synthetic diet dataset. As described in our paper, given the high complexity of diet planning (see Section 1 and Appendix A.4), real data are not high quality despite being created by professional dietitians. Thus, we had to create a novel high-quality dataset and devised the ORxML framework that integrates the capabilities of three modules: the OR module (i.e., a combinatorial optimization model) generates synthetic diets to satisfy explicit nutrient requirements; the expert module evaluates and adjusts the initial data in terms of implicit composition requirements (i.e., criteria that cannot be specified in the combinatorial optimization model); and the ML module automatically augments the data to ensure the composition compliance with nutrition enhancement. A series of experiments demonstrate the significance of the three modules and validate the quality of our dataset (see Section 4 and Appendix A.6). Finally, please note that this framework can be used in any other contexts of creating diet data and those of difficult professional tasks as well.
> For further details on this revision to clarify the abovementioned contribution points, please see the following point-to-point responses to your comments. We thank you again and sincerely hope that you will be satisfied with the responses and updated version of the paper.

---

> ### Author Response · Authors · 2021-09-28
> **Response to Comment 3.1(a)**
>
> Comment 3.1:
>
> (a) “Useful diet planning dataset” - Summary And Contributions: In this paper, the authors present a novel dataset for diet planning and healthcare research.
>
> (b) The authors describe the process of dataset creation by combining approaches from operations research, machine learning and knowledge from domain experts, as well as evaluation of the generated resource.
> The authors claim that their resource is the first resource for diet planning of that type. The contributions of the paper include: the K-MIND dataset and a diet planning methodology that combines knowledge from operations research models, machine learning and domain experts
>
> Response: Dear Reviewer 3, Thank you for your positive comments. Please kindly allow us to further highlight the importance of our contributions as follows. We trust that the following paragraphs can help discuss our response to your comments.
>
> (a) As highlighted by you and all the other reviewers, our first original contribution is to create the “first large-scale and high-quality” dataset for “diet planning with machine learning (ML)”. We believe that this contribution is significant and meaningful because diet planning is mainly considered an operations research (OR) problem (particularly a combinatorial optimization), which aims to fulfill nutrient requirements. However, this approach is ineffective in practice due to its limited capability to consider the implicit patterns in diets that professional dietitians believe as the most important (see Section 1 and Appendix A.4). In our paper, we originally clarify that the diet planning problem should actually be considered as a sequence generation to accommodate these implicit patterns (i.e., to fulfill the “compositional” requirements of diet planning) and tackle the concern on the lack of high-quality dataset to promote “diet planning with ML”.
>
> Specifically, diet planning—a basic and regular human activity—is important to all individuals, from children to seniors and from healthy people to patients; therefore, the history of this research stream is in fact quite long. Around 1945, diet planning was first defined as a linear programming problem to identify optimal quantities of food items. However, humans do not consume a specific quantity of each food unit (i.e., combination of ingredients cooked) but rather the end-product (i.e., food “menu”) as an entire unit. Thus, around 1990, diet planning has expanded into a mixed-integer programming problem to identify optimal combinations of menu items. However, all previous studies only considered the “ingredient and menu level” information whereas “diet level” planning involves the aforementioned compositional patterns of menus. Compositional patterns are implicit depending upon their contexts, and thus require data-driven machine learning approaches. Surprisingly, apart from our work, no other study seems to focus on real diet-level planning, for which we believe the main reason is the lack of high-quality diet dataset to research data-driven machine learning approaches. To the best of our knowledge, existing studies on the application of machine learning to dietary healthcare (e.g., Zeevi et al., 2015 published in Cell) consider only the ingredient and menu levels.
>
> In summary, diet planning is a highly important problem that needs machine learning solutions but could not be addressed as such due to the lack of datasets. In this paper, our dataset is the first high-quality resource to promote the research of “diet planning with machine learning”. Thanks to the allowance of one additional page, we could include machine learning use cases of diet planning and dietary healthcare with our dataset in Section 6; some of the cases are ongoing research of the authors with other colleagues. Through research and development of such use cases, we sincerely hope that the proposed dataset can be widely disseminated for diet planning and dietary healthcare with machine learning.

---

> ### Author Response · Authors · 2021-09-28
> **Dear Reviewer 3 (exb7), Thank you very much for your positive comments and valuable suggestions. Please see our responses to your comments as follows.**
>
> Dear Reviewer 3 (exb7), Thank you very much for your positive comments and valuable suggestions. Please see the following point-to-point responses to your comments. In addition, could you please kindly check and refer to the  submitted Supplementary Material for our full response letter to all comments from you? We prepared such a document to address the limit of characters in this openreview platform. Thank you very much for your understanding.
>
> We are pleased to be engaged with you as our reviewer and appreciate your incisive comments, which helped us improve our work significantly. Please let us know if you have any other concerns, questions, or suggestions. We would love to incorporate your additional feedback into our work. Through this discussion period, we will continuously update our submission and response as appropriate.

---

### Official Review · Reviewer_X1Na · 2021-09-20
**New and interesting contribution but the manuscript needs writing and presentation improvement**

**Rating:** 7
**Confidence:** 3

**Strengths:**

This dataset appears to be the first of its kind in this area. However, I am not an expert and might be missing out on related work. The authors have identified several important applications of various aspects of this work in the Appendix which I believe make this work useful beyond Korean food. The ML techniques applied here seem to improve performance on some of the metrics.

**Weaknesses:**

Introduction -
Related work - Almost no discussion of related works.
Considering each diet as a collection of 19 menu items appears to be a restrictive formulation. As authors pointed out, they include dummy menus to fix that problem with diets with a fewer number of menu items. Nevertheless, I think adopting a data structure like a tree or graph might be a more generic and flexible approach here.

**Additional Feedback:**

I am willing to bump up the rating if authors can address some of the writing and clarity-related concerns.

**Clarity:**

The terms introduced in L155-163 need to be thoroughly revised so that their English definition is clear to the reader before we jump into mathematical symbols associated with the optimization.

**Correctness:**

"Incidentally, a planner’s knowledge and belief rarely changes; therefore, the system of diet planning can be regarded as static and
explicit." This claim can be a controversial one. Since the whole point of integrating data driven techniques is to model the dynamic nature of diet planning.

**Documentation:**

Documentation of the two code repositories is reasonably well done. Authors also include datasheets for dataset which is a good step given the heavy data dependence of this work.

**Ethics:**

I do not see any significant ethical issues here.

**Relation To Prior Work:**

This paper lacks discussion with some of the prior work. Although the authors have cited many papers in the relevant areas. A proper taxonomy or a landscape of existing works and how they are different than this work should have been mentioned.

**Summary And Contributions:**

This work introduces the K-MIND dataset that has granularity at three different levels - Diet, Menu, and Nutrition.
It would be easier to navigate the paper if the authors can give a clear definition of all three key components.
Systematically breaks down food consumption under different abstractions through their proposed schema - diet, menu, ingredients and nutrition.

---

> ### Author Response · Authors · 2021-09-28
> **Response to Comment 2.5**
>
> Comment 2.5: Additional Feedback: I am willing to bump up the rating if authors can address some of the writing and clarity-related concerns. Rating: 5: Marginally below acceptance threshold, Confidence: 3: The reviewer is fairly confident that the evaluation is correct
>
> Response: Once again, thank you again for your constructive feedback and valuable suggestions. We sincerely hope that we have successfully clarified our contribution and enhanced the presentation of our work by addressing your comments. We hope that these adjustments increased your confidence in our work and have provided you with a more positive view of our work. If this is the case, may we humbly and respectfully ask that you elevate your ratings of our work with your increased confidence? Once again, thank you very much for your positive comments and valuable suggestions. We are pleased to be engaged with you as our reviewer and appreciate your incisive comments, which helped us improve our work significantly. Please let us know if you have any other concerns, questions, or suggestions. We would love to incorporate your additional feedback into our work.

---

> ### Author Response · Authors · 2021-09-28
> **Response to Comment 2.4**
>
> Comment 2.4: (a) Correctness: “Incidentally, a planner’s knowledge and belief rarely changes; therefore, the system of diet planning can be regarded as static and explicit.” This claim can be a controversial one. Since the whole point of integrating data driven techniques is to model the dynamic nature of diet planning. (b) Clarity: The terms introduced in L155-163 need to be thoroughly revised so that their English definition is clear to the reader before we jump into mathematical symbols associated with the optimization. (c) Relation To Prior Work: This paper lacks discussion with some of the prior work. Although the authors have cited many papers in the relevant areas. A proper taxonomy or a landscape of existing works and how they are different than this work should have been mentioned. (d) Documentation: Documentation of the two code repositories is reasonably well done. Authors also include datasheets for dataset which is a good step given the heavy data dependence of this work. (e) Ethics: I do not see any significant ethical issues here.
>
> Response: Thank you very much for your comment and suggestion. (a) We particularly appreciate this comment because our original contribution is to create a high-quality dataset to consider the dynamic nature of diet planning with data-driven machine learning techniques. Please note that the original sentence, “Incidentally, a planner’s knowledge and belief rarely change; therefore, the system of diet planning can be regarded as static and explicit.” was used to pinpoint the limitation of traditional approaches based on operations research that relies on the planner’s explicit knowledge and belief in modeling. In this revision, we deleted this phrase to avoid unnecessary confusion.
>
> (b) We thank you again for your sharp review regarding the clarity of our paper. In response to your comment, we clarified the definitions of diet and menu in Sections 1 and 5, as well as added example sentences helping to distinguish these concepts. In addition, the notations used in the optimization model in Section 2 were also modified more appropriately to prevent readers being confused with mathematical symbols.
>
> (c) As we responded to your Comment 2.2(a), for readers and users who are not familiar with diet planning or dietary healthcare research, we have elaborated on our literature review in Section 2 and Appendices A.1–A.4. Specifically, as requested, we clarify how existing literature differs from our work. In summary, all previous studies on diet planning consider the “ingredient and menu level” information whereas “diet level” planning should involve the compositional patterns of menus in diets. In addition, existing studies on dietary healthcare with machine learning (e.g., Zeevi et al., 2015 published in Cell) consider only the ingredient and menu levels. Diet planning is a highly important problem that requires solutions with machine learning but cannot be addressed as such due to the lack of datasets. We solve this concern in our paper, and our dataset is the first high-quality resource to promote the research field of “diet planning with machine learning”. Furthermore, Figure 1 in the Introduction has been edited to serve as a visual taxonomy or landscape that shows the spectrum from existing works (e.g., diet problem and diet planning with OR) to ours (i.e., diet planning and dietary healthcare with ML).
>
> By addressing your Comments 2.4(a)(b)(c), we believe that the presentation of our work has been significantly enhanced. Furthermore, please note that we used a copy-editing service for the updated manuscript to ensure readability. We sincerely thank you for providing valuable comments that enabled us to enhance the presentation of our work.
>
> (d) Thank you very much for your positive feedback. Nonetheless, please note that in this revision, we attempted to further improve our documentation. For example, we tried hard to further clarify the dataset creation and management processes by addressing the reviewers’ comments, including your Comment 2.3. In addition, we have further explained how our ORxML framework is being used now and in the future to expand the MIND dataset into multinational and multicultural contexts. Please see the revised Supplementary material D.2.
>
> (e) Yes, we also see no ethical concerns at this point. Furthermore, as shown in the Supplementary material, we have prepared all the license issues for the MIND dataset to ensure that no credit or authority issues would occur.

---

> ### Author Response · Authors · 2021-09-28
> **Response to Comment 2.3: Weaknesses**
>
> Comment 2.3: Weaknesses: (a) Introduction - Related work - Almost no discussion of related works. (b) Considering each diet as a collection of 19 menu items appears to be a restrictive formulation. As authors pointed out, they include dummy menus to fix that problem with diets with a fewer number of menu items. (c) Nevertheless, I think adopting a data structure like a tree or graph might be a more generic and flexible approach here.
>
> Response: Thank you very much for your constructive feedback and valuable suggestion. (a) As we responded to your Comment 2.2(a), for readers and users who are not familiar with diet planning or dietary healthcare research, we have elaborated on our literature review in Sections 1 and 2. In summary, all previous studies on diet planning consider the “ingredient and menu level” information whereas “diet level” planning should involve the compositional patterns of menus in diets. In addition, existing studies on dietary healthcare with machine learning (e.g., Zeevi et al., 2015 published in Cell) consider only the ingredient and menu levels. Diet planning is a highly important problem that requires solutions with machine learning but cannot be addressed as such due to the lack of datasets. We solve this concern in our paper, and our dataset is the first high-quality resource to promote the research field of “diet planning with machine learning”.
>
> (b) The diet sequence of 19 or less menu items is the standardized form of daily diets for children that have been used by the Center for Children’s Food Service Management in South Korea and the Ministry of Food and Drug Safety. Given that our research objective is to create a “standard” high-quality daily diet dataset, we applied this standardized form. On this basis, we have clarified this point in the revised Appendix A.2. Please see the added sentence in green font. In addition, we have added further details of diet data form in the Introduction and Section 5 to ensure clarity of our dataset. Meanwhile, please note that our work can be used as a reference in creating different forms of diet datasets (e.g., diet sequence of nine menu items including morning snacks, lunch, and afternoon snacks, with the assumption that the children eat breakfast and dinner at home). Our ORxML framework can also be used for this purpose.
>
> (c) Thank you for your sharp review and insightful comment. We agree with you that a tree or graph structure can be adopted as a diet data structure as well. Meanwhile, the mission of ML module in our ORxML framework was to ensure composition compliance with nutrition enhancement in diet generation. Thus, we thought one of the best ML methods for this mission could be the use of neural machine translation (NMT) for composition compliance (i.e., learning and complying sequential patterns in diet generation) and the use of reinforcement learning (RL) for nutrition enhancement (i.e., controlling the nutritional quality by providing rewards when specific nutrients are satisfied). While many of the NMT techniques deal with sequence data, RL controls that the agent (i.e., the generative model) is able to maximize the rewards based on sequential trial and error. Thus, in our diet data synthesis with the NMT-and-RL-based machine, it was effective to define diet as a sequence.
>
> Based on your advice, we have added a sentence in Section 3 that explicitly states the context why and how the ML module of our ORxML framework is based on NMT and RL. Please refer to the added sentence in green font. While a methodological contribution of our ORxML framework lies in the combination of NMT and RL for high-quality diet generation, we think the ML module can be designed in a different way to learn diet data in a graph or tree structure. We sincerely hope that future studies will take different approaches and expand the research field of “diet planning and dietary healthcare with machine learning.” We believe the proposed MIND dataset and the ORxML framework will be a good reference for such work. We stated this future research issues in the revised conclusion Section 6.

---

> ### Author Response · Authors · 2021-09-28
> **Response to Comment 2.2: Strengths(c.2)**
>
> (c.2) Second, as we responded to your Comment 2.2(b), we elaborated the applications of our work (e.g., machine learning use cases of diet planning and dietary healthcare with our dataset) in Section 6 and Appendix A.7. The exemplified applications can improve various metrics of dietary healthcare. For example, please kindly let us introduce our ongoing clinical study as follows. As we described in Appendix A.7 in the draft version, we are currently conducting a clinical study for children with atopic dermatitis (AD) and food allergy (FA) who must restrict allergenic foods which may lead to fatal anaphylaxis. The parental burden of these patients is very high because the children are limited to select social activities (e.g., camp and party) and restaurants due to the accidental exposure of allergenic food (Stensgaard et al., 2017 published in Clinical & Experimental Allergy). The children should eat at home or pack a lunch box. Thus, precise diet planning and dietary healthcare are necessary to manage the growth and health levels of the children. The authors are working together to develop an AI service application for this purpose. In fact, this work on creating a novel high-quality diet dataset was initiated for our clinical study for children with AD and FA; as real diet data in practice were not high quality enough to be used to train a machine for the AI service application.
>
> We are glad to tell you that recently we received a conditional approval from the IRB (Institutional Review Boards) for our clinical study on September 15th 2021 (IRB number: KUGH IRB No. 2021-09-019). To the best of our knowledge, this is the first clinical study to use and test the utility of an AI for diet-level planning and healthcare for patients. In this clinical study, the quantifiable “health metrics” you inquired upon include the Food Allergy Quality of Life-Parental burden, Food Allergy Quality Of Life Questionnaire-Parent Form, Food Allergy Independent Measure, nutrient assessment using 24-hour dietary recall, and food frequency questionnaires, growth status, and other life satisfaction indices. These metrics are validated measures in the clinical studies on patients with restricting foods (please see the references in Appendix A.7). Using these metrics, our clinical study will evaluate the utility of our machine for diet-level planning and healthcare in the AI service user group and the control group. We believe there will be a significant difference between the two groups and expect to start this clinical study on October or November 2021.
>
> Please note that, as we have highlighted several times earlier, a high-quality large-scale diet dataset is the basis for such clinical studies on diet planning and dietary healthcare with ML that could not be conducted due to its nonexistence. Our MIND dataset would be dedicated to diet planning and dietary healthcare not only for the contexts of allergic diseases but also for other contexts of chronic diseases such as diabetes mellitus, hypertension, obesity, celiac disease, gastrointestinal caner, liver cirrhosis, and chronic kidney failure. Although there exist digital healthcare services for obesity, existing services only support the assessment of what patients eat and recommend menus (not a way tailored to the individual needs). In the contexts of diabetes and mellitus, the proportion of carbohydrate, protein and fat must be considered carefully in diet planning, while the compositional compliance is still important. For patients with chronic kidney diseases, diets should be composed of menus with less potassium, phosphorus, and calcium. As such, the needs and requirements of diet planning and dietary healthcare for chronic diseases are all different, and it is always important to design compositionally quality diets that the patients would take. The MIND dataset is the first high-quality resource for this research direction and the ORxML framework can further encourage this research. Physicians and dietitians have been educated patient dietary management at the nutrition and menu levels, and our work can contribute to expanding patient dietary management at the diet level which patients actually need.
>
> Following your comment, we found that our draft could not highlight enough the use cases of our dataset for diet planning and dietary healthcare. Thanks to the allowance of adding one additional page, we were able to include and elaborate such use cases in Section 6 as well as in Appendix A.7. Please see the revised Section 4 and Appendix A.7, with the related revision shown in green font, for further details. By addressing your comment, we feel that the contribution of our work has become significantly stronger. Thank you.

---

> ### Author Response · Authors · 2021-09-28
> **Response to Comment 2.2: Strengths(c.1)**
>
> (c) We interpret your words “ML techniques” and “metrics” in two ways. (c.1) The first comprises the “ML techniques in the ORxML framework” for dataset creation to satisfy the required “metrics of diet planning” and (c.2) second are “ML techniques for healthcare applications” of our work to improve the “metrics of dietary healthcare”. Both points are discussed as follows.
>
> (c.1) First, as stated in our response to your Comment 2.1(a), ML techniques were applied in creating the MIND dataset using the ORxML framework. The purposes were to: (a) ensure composition compliance by learning the data patterns constructed by the OR and experts; (b) enhance nutrition by approximating an optimal policy to maximize the nutrient rewards; and (c) automatically augment the data by executing an optimal policy and generate synthetic diets. The domain users (i.e., dietitians) validated the quality of our dataset. As a result, we received approval from the government organizations responsible for the food quality and provision in South Korea (e.g., the Ministry of Food and Drug Safety and the Rural Development Administration) to distribute our MIND dataset. As the first step, the MIND dataset will be used by the professional dietitians and nutrition researchers in the Center for Children’s Food Service Management in South Korea from October 2021 (the fourth quarter); this organization is responsible for distributing high-quality diet references to all public daycare centers in South Korea. In addition, we are currently working with the Ministry of Food and Drug Safety for the extended use of the MIND dataset (e.g., developing an AI service application for diet planning).

---

> ### Author Response · Authors · 2021-09-28
> **Response to Comment 2.2: Strengths(a)(b)**
>
> Comment 2.2: Strengths: (a) This dataset appears to be the first of its kind in this area. However, I am not an expert and might be missing out on related work. (b) The authors have identified several important applications of various aspects of this work in the Appendix which I believe make this work useful beyond Korean food. (c) The ML techniques applied here seem to improve performance on some of the metrics.
>
> Response: Thank you very much for your positive comments. (a) For readers and users who are not familiar with diet planning or dietary healthcare research, we have elaborated on our literature review in Sections 1 and 2. In summary, as stated in our response to your Comment 2.1(a), all previous studies on diet planning consider the “ingredient and menu level” information whereas “diet level” planning should involve the compositional patterns of menus in diets. In addition, existing studies on dietary healthcare with machine learning (e.g., Zeevi et al., 2015 published in Cell) consider only the ingredient and menu levels. Diet planning is a highly important problem that requires solutions with machine learning but cannot be addressed as such due to the lack of datasets. We solve this concern in our paper, and our dataset is the first high-quality resource to promote the research field of “diet planning with machine learning”.
>
> (b) We were happy to see this positive comment as we actually wished to discuss the applications of our work in the main text but could not do so due to page limitations. Thanks to the allowance of one additional page, we could include the applications of our work (e.g., machine learning use cases of diet planning and dietary healthcare with our dataset) in Section 6; in addition, we added new use cases with real experiment results with the proposed MIND dataset in Section 6 and Appendix A.7. Please see the revised Section 6 and Appendix A.7. Some of the cases are ongoing research of the authors. Based on the exemplified use cases, we sincerely hope that the proposed dataset can be widely disseminated for diet planning and dietary healthcare with machine learning.
>
> Furthermore, we particularly appreciate this comment because we already started R&D projects to expand our dataset into multinational and multicultural contexts beyond Korea. Specifically, on July 2021, we received funding of USD 0.8 million from the National Research Foundation Korea (Grant Number: 2021R1I1A4A01049121) to use and improve the ORxML framework in order to expand our dataset. In this project, we will add diet sequence data from Western countries and other Asian countries (e.g., diets in North America, Europe, China, and Japan) to enhance the coverage and utility of our dataset. For this task, we will recruit domestic and international experts on specific national/cultural diets and collaborate with international restaurants in South Korea. Furthermore, with the goals of recruiting many experts and collaborating with various restaurants, we requested an additional funding of USD 1 million to the Ministry of Science and ICT of the South Korean government on September 2021. Finally, one of the authors is affiliated in a medical school in South Korea, and we are seeking opportunities to work with more physicians to use our dataset for clinical studies of dietary healthcare research and services.
>
> As such, we are highly committed to conducting research on diet planning and dietary healthcare with ML and will continuously expand our dataset for this research direction. As we have described in Section 5 and Supplementary material D.2, our dataset will be managed and updated regularly through our research on diet planning and dietary healthcare with ML. For example, we will extend the size and information of our dataset in accordance with the updates of related public databases (e.g., ingredient-nutrition databases of the Rural Development Administration of the Korean government). As for our abovementioned new project granted from the National Research Foundation Korea (Grant Number: 2021R1I1A4A01049121), our next objective is to extend the high-quality 1,500 diet data to 3,000 data using the ORxML framework.
>
> Based on your comment, we have changed the title of our paper to “MIND dataset for diet planning and dietary healthcare with machine learning: Dataset creation using combinatorial optimization and controllable generation with domain experts”. We deleted “K-” to reflect the abovementioned plan and to represent the general and broad impacts of our work. We thank you very much for inspiring and motivating us to clarify the contributions of this paper.

---

> ### Author Response · Authors · 2021-09-28
> **Response to Comment 2.1(a)**
>
> (a) Thank you very much for your words “new and interesting contribution” and “the dataset has granularity at three different levels”. Please kindly allow us to further highlight the importance of our contributions as follows. We trust that the following paragraphs can help discuss our response to your comments. As highlighted by you and all the other reviewers, our first original contribution is to create the “first large-scale and high-quality” dataset for “diet planning with machine learning (ML)”.
>
> We believe that this contribution is significant and meaningful because diet planning is mainly considered an operations research (OR) problem (particularly a combinatorial optimization), which aims to fulfill nutrient requirements. However, this approach is ineffective in practice due to its limited capability to consider the implicit patterns in diets that professional dietitians believe as the most important (see Section 1 and Appendix A.4). In our paper, we originally clarify that the diet planning problem should actually be considered as a sequence generation to accommodate these implicit patterns (i.e., to fulfill the “compositional” requirements of diet planning) and tackle the concern on the lack of high-quality dataset to promote “diet planning with ML”.
>
> Specifically, diet planning—a basic and regular human activity—is important to all individuals, from children to seniors and from healthy people to patients; therefore, the history of this research stream is in fact quite long. Around 1945, diet planning was first defined as a linear programming problem to identify optimal quantities of food items. However, humans do not consume a specific quantity of each food unit (i.e., combination of ingredients cooked) but rather the end-product (i.e., food “menu”) as an entire unit. Thus, around 1990, diet planning has expanded into a mixed-integer programming problem to identify optimal combinations of menu items. However, all previous studies only considered the “ingredient and menu level” information whereas “diet level” planning involves the aforementioned compositional patterns of menus. Compositional patterns are implicit depending upon their contexts, and thus require data-driven machine learning approaches. Surprisingly, apart from our work, no other study seems to focus on real diet-level planning, for which we believe the main reason is the lack of high-quality diet dataset to research data-driven machine learning approaches. To the best of our knowledge, existing studies on the application of machine learning to dietary healthcare (e.g., Zeevi et al., 2015 published in Cell) consider only the ingredient and menu levels.
>
> In summary, diet planning is a highly important problem that needs machine learning solutions but could not be addressed as such due to the lack of datasets. In this paper, our dataset is the first high-quality resource to promote the research of “diet planning with machine learning”. Thanks to the allowance of one additional page, we could include machine learning use cases of diet planning and dietary healthcare with our dataset in Section 6; some of the cases are ongoing research of the authors with other colleagues. Through research and development of such use cases, we sincerely hope that the proposed dataset can be widely disseminated for diet planning and dietary healthcare with machine learning.
>
> Furthermore, we believe that our dataset creation framework, OR–Xperts–ML (ORxML), is also unique and relevant to the dataset creation efforts for machine learning research (e.g., the NeurIPS 2021 Datasets and Benchmarks Track). Our second contribution is to devise the ORxML framework to systematically create a large-scale high-quality synthetic diet dataset. As described in our paper, given the high complexity of diet planning (see Section 1 and Appendix A.4), real data are not high quality despite being created by professional dietitians. Thus, we had to create a novel high-quality dataset and devised the ORxML framework that integrates the capabilities of three modules: the OR module (i.e., a combinatorial optimization model) generates synthetic diets to satisfy explicit nutrient requirements; the expert module evaluates and adjusts the initial data in terms of implicit composition requirements (i.e., criteria that cannot be specified in the combinatorial optimization model); and the ML module automatically augments the data to ensure the composition compliance with nutrition enhancement. A series of experiments demonstrate the significance of the three modules and validate the quality of our dataset (see Section 4 and Appendix A.6). Finally, please note that this framework can be used in any other contexts of creating diet data and those of difficult professional tasks as well.
>
> For further details on this revision to clarify the abovementioned contribution points, please see the following point-to-point responses to your comments. We thank you again.

---

> ### Author Response · Authors · 2021-09-28
> **Response to Comment 2.1(b)**
>
> Comment 2.1: (a) “New and interesting contribution but the manuscript needs writing and presentation improvement” - Summary And Contributions: This work introduces the K-MIND dataset that has granularity at three different levels - Diet, Menu, and Nutrition. (b) It would be easier to navigate the paper if the authors can give a clear definition of all three key components. Systematically breaks down food consumption under different abstractions through their proposed schema - diet, menu, ingredients and nutrition.
>
> Response: Dear Reviewer 2, Thank you for your comment and suggestion. Please let us respond to point (b) first and then (a).
>
> (b) We particularly appreciate this comment given that the users of our proposed dataset may not be familiar with the definition of diet, menu, ingredient, and nutrition. For this reason, we have added clear definitions in the Introduction with the revised Figure 1 as well as elaborated the definitions in Section 5. Please check the texts in green font. We thank you very much for enabling us to significantly improve the clarity of our dataset and writing based on your comment.

---

> ### Author Response · Authors · 2021-09-28
> **Dear Reviewer 2 (X1Na), Thank you very much for your positive comments and valuable suggestions. Please see our responses to your comments as follows.**
>
> Dear Reviewer 2 (X1Na), Thank you very much for your positive comments and valuable suggestions. Please see the following point-to-point responses to your comments. In addition, could you please kindly check and refer to the  submitted Supplementary Material for our full response letter to all comments from you? We prepared such a document to address the limit of characters in this openreview platform. Thank you very much for your understanding.
>
> We are pleased to be engaged with you as our reviewer and appreciate your incisive comments, which helped us improve our work significantly. Please let us know if you have any other concerns, questions, or suggestions. We would love to incorporate your additional feedback into our work. Through this discussion period, we will continuously update our submission and response as appropriate.

---

> ### Comment · Reviewer_X1Na · 2021-09-29
> **Response to the rebuttal**
>
> I appreciate the author's diligence in providing a strong rebuttal. Some of the concerns have been addressed in the revised draft and hence I have bumped up the rating.
>
> I would appreciate it if authors can add a small discussion on few papers that I think might warrant some discussion -
>
> - Veselkov, K., Gonzalez, G., Aljifri, S., Galea, D., Mirnezami, R., Youssef, J., Bronstein, M. and Laponogov, I., 2019. HyperFoods: Machine intelligent mapping of cancer-beating molecules in foods. Scientific reports, 9(1), pp.1-12.
> - Cammarota, G., Ianiro, G., Ahern, A., Carbone, C., Temko, A., Claesson, M.J., Gasbarrini, A. and Tortora, G., 2020. Gut microbiome, big data and machine learning to promote precision medicine for cancer. Nature reviews gastroenterology & hepatology, 17(10), pp.635-648.
> - Zhao, S., Mao, X., Lin, H., Yin, H. and Xu, P., 2020. Machine Learning Prediction for 50 Anti-Cancer Food Molecules from 968 Anti-Cancer Drugs. International Journal of Intelligence Science, 10(1), pp.1-8.
> - Min, W., Liu, C. and Jiang, S., 2021. Towards Building a Food Knowledge Graph for Internet of Food. arXiv preprint arXiv:2107.05869.
> - Park, D., Kim, K., Kim, S., Spranger, M. and Kang, J., 2021. FlavorGraph: a large-scale food-chemical graph for generating food representations and recommending food pairings. Scientific reports, 11(1), pp.1-13.
> - Yang, Z.F., Xiao, R., Luo, F.J., Lin, Q.L., Ouyang, D., Dong, J. and Zeng, W.B., 2020. Food bioactive small molecule databases: Deep boosting for the study of food molecular behaviors. Innovative Food Science & Emerging Technologies, 66, p.102499.

---

> > ### Author Response · Authors · 2021-09-30
> > **"Toward precision diet for healthcare" : Thank you very much for your positive response to our revision and rebuttal.**
> >
> > Dear Reviewer 2 (X1Na), Thank you very much for your positive response to our revision and rebuttal. We are glad to hear that we have successfully clarified our contribution and enhanced the presentation of our work by addressing your comments. In addition, we appreciate your additional feedback with very interesting references. We had read one of the references already but could not think of explicitly connecting it to our work. However, after reading the other five references all together, we found that it would be great to add a subsection entitled “Toward precision diet for healthcare” in our paper that discusses the potential synergies of our work with existing studies on precision nutrition and food science. Please see this newly added subsection in Appendix A.7. In the revised Section 6, we have also elaborated the limitation of our work considering the necessity of integrating our MIND dataset with existing databases covering the molecules and compounds. Finally, we have elaborated the subsection “Task 4: Clinical studies with the MIND data set” in Appendix A.7 from the perspective of gut microbiome.
> >
> > Please check the newly updated paper. We thank you very much for inspiring and motivating us to further enhance the contribution and implications of our work. We hope that this revision increased your confidence in our work and have provided you with a more positive view of our work. Once again, thank you very much for your positive comments and valuable suggestions so far. We are pleased to be engaged with you as our reviewer and appreciate your incisive comments, which helped us improve our work significantly. Please let us know if you have any other concerns, questions, or suggestions. We would love to incorporate your additional feedback into our work continuously after this rebuttal and revision process.

---

### Official Review · Reviewer_RZVR · 2021-09-22
**First large-scale dataset on diet planning**

**Rating:** 6
**Confidence:** 3
**Correctness:** The procedure for creating the datase…
**Clarity:** The paper is well written.

**Strengths:**

1. K–MIND dataset is the largest dataset that is ever introduced for the task of diet planning.
2. ORxML framework that is suggested in this dataset can further encourage research in the area of diet planning.

**Weaknesses:**

1. The first limitation of this work is that it only includes the Korean diet. The proposed ORxML framework is indeed more general-purpose, but the K–MIND dataset cannot be used for diet planning in any other part of the world.
2. It is great that the authors take into account the dietitians' edits in the third step of the dataset creation process, but K–MIND is a synthetic dataset. The authors have evaluated the generated diets in terms of RDI score, but it doesn't consider the human element of the diet. It is essential to make sure that the generated diets are actually desirable for humans. Otherwise, it might be a very healthy and nutritionally perfect diet, but people might not want to take it.

**Additional Feedback:**

It would be great if the authors could run a clinical study to further evaluated the suggested diets. It is beneficial to see how participants would take the diest and hear their feedback. In addition, showing quantifiable health measures from the clinical study would make this work significantly stronger.

**Documentation:**

The authors have very well documented the dataset and the data creation process.

**Ethics:**

There are no ethical concerns at this point.

**Relation To Prior Work:**

The authors have discussed the diet planning problem and previous approaches using machine learning to tackle the problem. However, a detailed comparison between the available datasets on the task of dier planning is missing.

**Summary And Contributions:**

The authors introduced the Korean Menus–Ingredients–Nutrients–Diets (K–MIND) dataset for diet planning containing 1,140 diets, 3,238 menus, 3,036 ingredients. For creating the dataset, they used combinatorial optimization techniques in operations research (OR) to generate menus that satisfy the nutritional requirements. Then, dietitians edit the solutions to make them more desirable. Finally, an ML module is trained on the edited diet data. The ML module can be used to generate as many diets as necessary. They call this framework ORxML and have released its python package.

---

> ### Author Response · Authors · 2021-09-28
> **Response to Comment 1.6**
>
> Comment 1.6: Rating: 6: Marginally above acceptance threshold, Confidence: 3: The reviewer is fairly confident that the evaluation is correct
>
> Response: Once again, thank you very much for your positive comments and valuable suggestions. We sincerely hope that we have successfully clarified our contribution and enhanced the presentation of our work by addressing your comments. We hope that these adjustments increased your confidence in our work and have provided you with a more positive view of our work. If this is the case, may we humbly and respectfully ask that you elevate your ratings of our work with your increased confidence? Once again, thank you very much for your positive comments and valuable suggestions. We are pleased to be engaged with you as our reviewer and appreciate your incisive comments, which helped us improve our work significantly. Please let us know if you have any other concerns, questions, or suggestions. We would love to incorporate your additional feedback into our work.

---

> ### Author Response · Authors · 2021-09-28
> **Response to Comment 1.5: Additional Feedback(b)**
>
> (b) Furthermore, we believe the other part of your comment can be addressed by our ongoing clinical study as follows. As we described in Appendix A.7 in the draft version, we are currently conducting a clinical study for children with atopic dermatitis (AD) and food allergy (FA) who must restrict allergenic foods which may lead to fatal anaphylaxis. The parental burden of these patients is very high because the children are limited to select social activities (e.g., camp and party) and restaurants due to the accidental exposure of allergenic food (Stensgaard et al., 2017 published in Clinical & Experimental Allergy). The children should eat at home or pack a lunch box. Thus, precise diet planning and dietary healthcare are necessary to manage the growth and health levels of the children. The authors are working together to develop an AI service application for this purpose. In fact, this work on creating a novel high-quality diet dataset was initiated for our clinical study for children with AD and FA; as real diet data in practice were not high quality enough to be used to train a machine for the AI service application.
>
> We are glad to tell you that recently we received a conditional approval from the IRB (Institutional Review Boards) for our clinical study on September 15th 2021 (IRB number: KUGH IRB No. 2021-09-019). To the best of our knowledge, this is the first clinical study to use and test the utility of an AI for diet-level planning and healthcare for patients. In this clinical study, the quantifiable health measures you inquired upon include the Food Allergy Quality of Life-Parental burden, Food Allergy Quality Of Life Questionnaire-Parent Form, Food Allergy Independent Measure, nutrient assessment using 24-hour dietary recall, and food frequency questionnaires, growth status, and other life satisfaction indices. These measures are validated measures in the clinical studies on patients with restricting foods (please see the references in Appendix A.7). Using these measures, our clinical study will evaluate the utility of our machine for diet-level planning and healthcare in the AI service user group and the control group. We believe there will be a significant difference between the two groups and expect to start this clinical study on October or November 2021.
>
> Please note that, as we have highlighted several times earlier, a high-quality large-scale diet dataset is the basis for such clinical studies on diet planning and dietary healthcare with ML that could not be conducted due to its nonexistence. Our MIND dataset would be dedicated to diet planning and dietary healthcare not only for the contexts of allergic diseases but also for other contexts of chronic diseases such as diabetes mellitus, hypertension, obesity, celiac disease, gastrointestinal caner, liver cirrhosis, and chronic kidney failure. Although there exist digital healthcare services for obesity, existing services only support the assessment of what patients eat and recommend menus (not a way tailored to the individual needs). In the contexts of diabetes and mellitus, the proportion of carbohydrate, protein and fat must be considered carefully in diet planning, while the compositional compliance is still important. For patients with chronic kidney diseases, diets should be composed of menus with less potassium, phosphorus, and calcium. As such, the needs and requirements of diet planning and dietary healthcare for chronic diseases are all different, and it is always important to design compositionally quality diets that the patients would take. The MIND dataset is the first high-quality resource for this research direction and the ORxML framework can further encourage this research. Physicians and dietitians have been educated patient dietary management at the nutrition and menu levels, and our work can contribute to expanding patient dietary management at the diet level which patients actually need.
>
> Following your comment, we found that our draft could not highlight enough the use cases of our dataset for diet planning and dietary healthcare. Thanks to the allowance of adding one additional page, we were able to include and elaborate such use cases in Section 6 as well as in Appendix A.7. Please see the revised Section 4 and Appendix A.7, with the related revision shown in green font, for further details. By addressing your comment, we feel that the contribution of our work has become significantly stronger. Thank you.

---

> ### Author Response · Authors · 2021-09-28
> **Response to Comment 1.5: Additional Feedback(a)**
>
> Comment 1.5: Additional Feedback: It would be great if the authors could run a clinical study to further evaluated the suggested diets.
>
> (a) It is beneficial to see how participants would take the diest and hear their feedback.
>
> (b) In addition, showing quantifiable health measures from the clinical study would make this work significantly stronger.
>
> Response: Thank you for your positive comment and valuable suggestion.
>
> (a) We think a part of your comment can be addressed by our response to your Comment 1.3(b). Specifically, although the expert survey was not a clinical study, the healthcare and nutrition domain users (i.e., dietitians) of our dataset validated the dataset quality. As a result, the MIND dataset will be used by the professional dietitians and nutrition researchers in the Center for Children’s Food Service Management in South Korea from October 2021 (the fourth quarter); this organization is responsible for distributing high-quality diet references to all public daycare centers in South Korea. In addition, we are currently working with the Ministry of Food and Drug Safety for the extended use of the MIND dataset (e.g., developing an AI service application for diet planning).

---

> ### Author Response · Authors · 2021-09-28
> **Response to Comment 1.4**
>
> Comment 1.4: (a) Correctness: The procedure for creating the dataset seems logit to me. (b) Clarity: The paper is well written. (c) Relation To Prior Work: The authors have discussed the diet planning problem and previous approaches using machine learning to tackle the problem. However, a detailed comparison between the available datasets on the task of dier planning is missing. (d) Documentation: The authors have very well documented the dataset and the data creation process. (e) Ethics: There are no ethical concerns at this point.
>
> Response: Thank you for your positive comment and suggestion. (a) (b) Although we are thankful that you are already satisfied with the correctness and clarity of our paper, please note that we used a copyediting service for the updated manuscript to check English grammar and ensure readability.
>
> (c) In response to our response to your Comment 1.1(a), to the best of our knowledge, no work on diet planning with ML has been conducted, except ours (this work and our previous work on the ML module for ORxML framework—Lee et al., 2021 published in Proceedings of the 27th ACM SIGKDD). Previous studies on diet planning have focused on OR-based modeling approaches (not data-driven ML approaches). As for your words “previous approaches using machine learning”, if you meant the application of ML to “dietary healthcare”, there are such studies in the medical and nutrition domains (e.g., Zeevi et al., 2015 published in Cell). However, although these studies use the term “diet”, these only considered the ingredient and menu levels (not a diet level). As we have mentioned earlier, “diet-level” planning is a very important yet highly difficult problem that must be solved with ML but could not be addressed in this way due to the lack of datasets for this data-driven approach. Our dataset is the first high-quality resource to promote the research field of “diet planning and dietary healthcare with ML.” Nonetheless, we think our writing of the literature review section in our draft paper could be improved. Thus, in this revision process we have tried hard to improve the literature review in the introduction section and Section 2.
>
> (d) Thank you very much for your positive feedback. Nonetheless, please note that, in this revision process, we tried hard to further clarify the dataset creation and evaluation processes by addressing the reviewers’ comments, including your Comment 1.3(b). In addition, by addressing your Comment 1.3(a), we have further explained how our ORxML framework is being used now and in the future to expand the MIND dataset into multinational and multicultural contexts.
>
> (e) Yes, there are no ethical concerns at this point. Furthermore, as shown in the Supplementary material E, we have prepared all the license issues for the MIND dataset to ensure that there would be no credit or authority issues.

---

> ### Author Response · Authors · 2021-09-28
> **Response to Comment 1.3: Weaknesses(b)**
>
> (b) Indeed, we also particularly appreciate this comment, because our experiments already included “the human evaluation of our dataset” that you inquired upon. We conducted a survey of 51 professional dietitians to further evaluate whether the generated diets are actually desirable for humans. Please see Section 4 and Appendix A.6 for details. In summary, the quality of the generated diets was validated by the experts. As a result, we received approval from the government organizations responsible for the food quality and provision in South Korea (e.g., the Ministry of Food and Drug Safety and the Rural Development Administration) to distribute our MIND dataset. As the first step, the MIND dataset will be used by the professional dietitians and nutrition researchers in the Center for Children’s Food Service Management in South Korea from October 2021 (the fourth quarter); this organization is responsible for distributing high-quality diet references to all public daycare centers in South Korea. In addition, we are currently working with the Ministry of Food and Drug Safety for the extended use of the MIND dataset (e.g., developing an AI service application for diet planning). Furthermore, the findings from this survey of 51 professional dietitians indicate the following interesting points of our work. First, the experts considered composition compliance as the most important factor in evaluating diets, despite being incapable of accurately evaluating the nutritional quality of diets. Second, the lack of capability confirms the motivation of this work to create and publish the MIND dataset using combinatorial optimization and controllable generation. This is because there is no high-quality diet-level dataset existing at present, even among professionals, such as dietitians and physicians, due to the extremely high complexity of diet planning. This situation confirms the necessity of the OR and ML modules in our ORxML framework. Third, the importance of composition compliance confirms the irreplaceable role of domain experts (professional dietitians) for diet planning and data generation in considering the implicit requirements. Again, this confirms the necessity of the expert module in our ORxML framework. Following your comment, we found that our draft could not highlight the importance of “the human evaluation of our dataset” that you inquired upon. Thus, we elaborated the related discussion in our paper. Please see the last paragraph in the revised Section 4 with the related revision shown in green font; please see Appendix A.6 for details of the human evaluation of our dataset. Again, we thank you for inspiring and motivating us to clarify the contribution of our work. Finally, in relation to the above response (a), the human evaluation results will always be conducted when we expand the dataset into multinational and multicultural contexts. Please note that we have also highlighted this point in Supplementary material D.2.

---

> ### Author Response · Authors · 2021-09-28
> **Response to Comment 1.3: Weaknesses(a)**
>
> Comment 1.3: Weaknesses: (a) The first limitation of this work is that it only includes the Korean diet. The proposed ORxML framework is indeed more general-purpose, but the K–MIND dataset cannot be used for diet planning in any other part of the world. (b) It is great that the authors take into account the dietitians' edits in the third step of the dataset creation process, but K–MIND is a synthetic dataset. The authors have evaluated the generated diets in terms of RDI score, but it doesn't consider the human element of the diet. It is essential to make sure that the generated diets are actually desirable for humans. Otherwise, it might be a very healthy and nutritionally perfect diet, but people might not want to take it.
>
> Response: Thank you for your constructive comments and suggestions. (a) We particularly appreciate this comment because, in fact, we already started the R&D projects to expand our dataset into multinational and multicultural contexts beyond Korea. Specifically, on July 2021, we received funding of USD 0.8 million from the National Research Foundation Korea (Grant Number: 2021R1I1A4A01049121) to use and improve the ORxML framework in order to expand our dataset. In this project, we will add diet sequence data from Western countries and other Asian countries (e.g., diets in North America, Europe, China, and Japan) to enhance the coverage and utility of our dataset. For this task, we will recruit domestic and international experts on specific national/cultural diets and collaborate with international restaurants in South Korea. Furthermore, with the goals of recruiting many experts and collaborating with various restaurants, we requested an additional funding of USD 1 million to the Ministry of Science and ICT of the South Korean government on September 2021. Finally, one of the authors is affiliated in a medical school in South Korea, and we are seeking opportunities to work with more physicians to use our dataset for clinical studies of dietary healthcare research and services. As such, we are highly committed to conducting research on diet planning and dietary healthcare with ML and will continuously expand our dataset for this research direction. As we have described in Section 5 and Supplementary material D.2, our dataset will be managed and updated regularly through our research on diet planning and dietary healthcare with ML. For example, we will extend the size and information of our dataset in accordance with the updates of related public databases (e.g., ingredient-nutrition databases of the Rural Development Administration of the Korean government). As for our abovementioned new project granted from the National Research Foundation Korea (Grant Number: 2021R1I1A4A01049121), our next objective is to extend the high-quality 1,500 diet data to 3,000 data using the ORxML framework. Based on your comment, we have changed the title of our paper to “MIND dataset for diet planning and dietary healthcare with machine learning: Dataset creation using combinatorial optimization and controllable generation with domain experts.” We deleted “K-” to reflect the abovementioned plan and to represent the anticipated general and broader impacts of our work. In addition, we elaborated Supplementary material D.2. Once again, we express our deepest gratitude for inspiring and motivating us to clarify the contribution of our work.

---

> ### Author Response · Authors · 2021-09-28
> **Response to Comment 1.2: Strengths**
>
> Comment 1.2: Strengths: (a) K–MIND dataset is the largest dataset that is ever introduced for the task of diet planning. (b) ORxML framework that is suggested in this dataset can further encourage research in the area of diet planning.
>
> Response: Once again, thank you very much for highlighting our two unique contributions. (a) As we responded to your Comment 1.1(a), the task of diet planning is actually the problem that should be solved with ML (to consider both nutrient and composition requirements) beyond operations research (which is mainly able to consider nutrient requirements). However, this problem could not be addressed in this way due to the lack of datasets for this data-driven way. Our dataset is the first high-quality resource for this research direction. As diet planning—a basic and regular human activity—is important to all individuals, from children to seniors and from healthy people to patients. Related to this, we believe our paper can become a seminal work that can help promote the research field of “diet planning and dietary healthcare with machine learning.”
>
> (b) Yes, as we responded to your Comment 1.1(b), we believe the ORxML framework can further encourage research in the area of diet planning. Furthermore, beyond creating any other contexts of generating diet data (e.g., diet sequence data in Western countries and other Asian countries), this framework can be used to create the data of difficult professional design tasks (e.g., data of fashion item sequence to fulfill explicit seasonal and implicit preference requirements). As highlighted by you in your Comment 1.3(b) and Comment 4.2(c) from Reviewer 4, the unique advantage of this framework is to combine the capabilities of OR (for explicit requirements of data of difficult professional design tasks) experts (for implicit requirements), and ML (for both) to create a high-quality synthetic dataset for machine learning research and development. We sincerely hope that this human-in-the-loop framework will be used by other researchers for other contexts of creating diet data and the data of difficult professional design tasks. Thanks to your comments, we were able to elaborate our paper significantly. Please see the revised Section 1 and Appendix A.7, with changes shown in green font, for further details. By addressing your comment, we feel that the contribution of our work has been further clarified. Thank you.

---

> ### Author Response · Authors · 2021-09-28
> **Response to Comment 1.1(b)**
>
> (b) Furthermore, we believe our dataset creation framework (ORxML) is unique and relevant to the dataset creation efforts for ML research (e.g., the NeurIPS 2021 Datasets and Benchmarks Track). Our second contribution is to devise the ORxML framework to systematically create a large-scale high-quality synthetic diet dataset. As described in our paper, given the high complexity of diet planning (see Section 1 and Appendix A.4), real diet data in practice are often not high quality even though such data were generated by professional dietitians. Thus, we had to create a novel high-quality dataset and devised an OR–Xperts–ML (ORxML) framework for diet planning and dietary healthcare with ML. This framework integrates the capabilities of OR, expert, and ML modules, wherein the OR module (i.e., a combinatorial optimization model) generates synthetic diets to satisfy explicit nutrient requirements. In addition, the expert module evaluates and adjusts the initial data in terms of implicit composition requirements (i.e., the criteria that cannot be specified in the combinatorial optimization model), while the ML module automatically augments the data that ensure the composition compliance with nutrition enhancement. A series of experiments demonstrate the significance of the three modules and validate the quality of our dataset (see Section 4 and Appendix A.6). Finally, please note that this framework can be used in any other contexts of creating diet data and those of difficult professional tasks as well.
>
> For further details on this revision job and to clarify the abovementioned contribution points, please see the following point-by-point responses to your comments. Once again, we thank you for this opportunity to clarify some issues. We sincerely hope that you will be satisfied with our responses and with the updated version of our paper.

---

> ### Author Response · Authors · 2021-09-28
> **Response to Comment 1.1(a)**
>
> Comment 1.1: (a) “First large-scale dataset on diet planning” - Summary And Contributions: The authors introduced the Korean Menus–Ingredients–Nutrients–Diets (K–MIND) dataset for diet planning containing 1,140 diets, 3,238 menus, 3,036 ingredients.
> (b) For creating the dataset, they used combinatorial optimization techniques in operations research (OR) to generate menus that satisfy the nutritional requirements. Then, dietitians edit the solutions to make them more desirable. Finally, an ML module is trained on the edited diet data. The ML module can be used to generate as many diets as necessary. They call this framework ORxML and have released its python package.
>
> Response: Dear Reviewer 1, thank you very much for your positive comment.
>
> (a) As you have pointed out along with the other reviewers, our first original contribution is the creation of the “first large-scale and high-quality” dataset for “diet planning with machine learning (ML).” Please kindly allow us to further highlight the importance of this contribution as follows. We trust that the following paragraphs can help discuss our response to your comments.
>
> We believe this contribution is significant and meaningful, because diet planning has been mainly considered an operations research (OR) problem in academia (particularly a combinatorial optimization problem)—one that mainly aims to fulfill the nutrient requirements of diets. However, this approach has been ineffective in actual practice due to its limited capability to consider the implicit patterns in diets, which are considered most important by professional dietitians (see Section 1 and Appendix A.4). In our paper, we originally clarified that the diet planning problem should actually be considered a sequence generation problem in order to accommodate the implicit patterns in diet sequences (i.e., to fulfill the “compositional” requirements of diet planning) and to effectively tackle the issue that there is no high-quality dataset to promote “diet planning with ML.”
>
> Specifically, diet planning—a basic and regular human activity—is important to all individuals—from children to seniors and from healthy people to patients. As such, the history of diet planning research is quite long. Around 1945, diet planning was first defined as a linear programming problem to identify optimal quantities of food items. However, humans do not consume a specific quantity of each food unit (i.e., combination of ingredients cooked) but rather the end-product (i.e., food “menu”) as a whole unit. Thus, around 1990, the problem was expanded to a mixed-integer programming problem to identify optimal combinations of menu items. Meanwhile, all these previous studies considered the “ingredient and menu-level” information, whereas “diet-level” planning should involve the aforementioned compositional patterns of menus in diets. The compositional patterns are implicit depending upon the contexts, which means that it should be addressed by data-driven machine learning (ML) approaches. Surprisingly, there seems no study on real diet-level planning except ours (this work and our previous work on the ML module for ORxML framework—Lee et al., 2021 published in Proceedings of the 27th ACM SIGKDD), and we think the main reason is the lack of high-quality diet dataset with which to investigate data-driven ML approaches. To the best of our knowledge, existing studies on the application of ML to dietary healthcare (e.g., Zeevi et al., 2015 published in Cell) also considered the ingredient and menu levels only.
>
> In summary, diet planning is a very important problem that, we believe, must be solved with ML but could not be addressed in this way due to the lack of datasets for this data-driven approach. Our dataset is the first high-quality resource to promote the research field of “diet planning with ML.” Thanks to the allowance of adding one additional page, we were able to include the ML use cases of diet planning and dietary healthcare with our dataset in Section 6; some of the cases are ongoing research of the authors with other colleagues. Through research and development of such use cases, we sincerely hope that the proposed dataset will be widely disseminated for diet planning and dietary healthcare with ML.

---

> ### Author Response · Authors · 2021-09-28
> **Dear Reviewer 1 (RZVR), Thank you very much for your positive comments and valuable suggestions. Please see our responses to your comments as follows.**
>
> Dear Reviewer 1 (RZVR), Thank you very much for your positive comments and valuable suggestions. Please see the following point-to-point responses to your comments. In addition, could you please kindly check and refer to the  submitted Supplementary Material for our full response letter to all comments from you? We prepared such a document to address the limit of characters in this openreview platform. Thank you very much for your understanding.
>
> We are pleased to be engaged with you as our reviewer and appreciate your incisive comments, which helped us improve our work significantly. Please let us know if you have any other concerns, questions, or suggestions. We would love to incorporate your additional feedback into our work. Through this discussion period, we will continuously update our submission and response as appropriate.

---

### Author Response · Authors · 2021-09-29
**Dear Reviewers, Thank you very much for your positive comments and valuable suggestions. Please see our responses to your comments as follows.**

We sincerely thank the reviewers and area chairs for their helpful and insightful comments and suggestions, which helped us improve the paper significantly. Herein, we respond to each comment.

The updated paper and point-to-point responses to the comments are submitted as requested. Please see our responses to the comments for details. In summary, given the comments from reviewers, we have revised our paper by (1) clarifying and highlighting our contributions in the introduction and discussion sections; (2) improving the introduction and literature review sections to reflect related studies clearly; (3) elaborating the method, experiment, and dataset sections to enhance the detail and transparency; and (4) improving the discussion and conclusion sections to enhance the implications of our work. We have also revised the title to “MIND dataset for diet planning and dietary healthcare with machine learning: Dataset creation using combinatorial optimization and controllable generation with domain experts” to clarify the contribution of our work to “diet planning and dietary healthcare with machine learning”. According to the guideline, we added one additional page in the manuscript to incorporate the reviewer comments; this change significantly helped us highlight the contribution of our work as well as enhance the clarity of writing. Thank you.

Through this revision process, we believe that the academic contribution and positioning of our work in the fields of machine learning, diet planning, and dietary healthcare research have been strengthened and clarified significantly based on the reviewer comments. We thank the reviewers and area chairs for the valuable suggestions. We look forward to hearing from you.

---

### Decision · Program_Chairs · 2021-10-09

**Decision:**

Accept

**Comment:**

Dear authors,

Thank you for submitting your paper and addressing the issues raised by the reviewers.

The paper introduces a novel dataset on diet planning and a tool that generated the data. The dataset was generated synthetically but edited and improved by human domain experts and Machine Learning.

The reviewers commend the importance of the data, use of human experts to improve the data quality, general clarity of the paper and easy to use python package.

The main weakness is the potentially limited use of the data and its limitation to Korean food. In the appendix A7 several additional use-cases are discussed.

The reviewer scores are 7,6,6 and 7 which is sufficient for accepting the paper.